# Young People's Burden: Requirement of Negative CO₂ Emissions

**James Hansen,[1] Makiko Sato,[1] Pushker Kharecha,[1] Karina von Schuckmann,[2] David J. Beerling,[3] Junji Cao,[4] Shaun Marcott,[5] Valerie Masson-Delmotte,[6] Michael J. Prather,[7] Eelco J. Rohling,[8,9] Jeremy Shakun,[10] Pete Smith,[11] Andrew Lacis,[12] Gary Russell,[12] Reto Ruedy[12,13]**

[1]Climate Science, Awareness and Solutions, Columbia University Earth Institute, New York, NY 10115 [2]Mercator Ocean, 10 Rue Hermes, 31520 Ramonville St Agne, France [3]Leverhulme Centre for Climate Change Mitigation, University of Sheffield, Sheffield S10 2TN, UK [4]Key Lab of Aerosol Chemistry and Physics, SKLLQG, Institute of Earth Environment, Xi'an 710061, China [5]Department of Geoscience, 1215 W. Dayton St., Weeks Hall, University of Wisconsin-Madison, Madison, WI 53706 [6]Institut Pierre Simon Laplace, Laboratoire des Sciences du Climat et de l'Environnement (CEA-CNRS-UVSQ) Université Paris Saclay, Gif-sur-Yvette, France [7]Earth System Science Department, University of California at Irvine, CA [8]Research School of Earth Sciences, The Australian National University, Canberra, 2601, Australia [9]Ocean and Earth Science, University of Southampton, National Oceanography Centre, Southampton, SO14 3ZH, UK [10]Department of Earth and Environmental Sciences, Boston College, Chestnut Hill, MA 02467 [11]Institute of Biological and Environmental Sciences, University of Aberdeen, 23 St Machar Drive, AB24 3UU, UK, [12]NASA Goddard Institute for Space Studies, New York, NY 10025, [13]SciSpace LLC, 2880 Broadway, New York, NY 10025

**E-mail**: jeh1@columbia.edu

**Keywords**: climate change, carbon budget, intergenerational justice

## Abstract

Global temperature is a fundamental climate metric highly correlated with sea level, which implies that keeping shorelines near their present location requires keeping global temperature within or close to its preindustrial Holocene range. However, global temperature excluding short-term variability now exceeds +1°C relative to the 1880-1920 mean and annual 2016 global temperature was almost +1.3°C. We show that global temperature has risen well out of the Holocene range and Earth is now as warm as it was during the prior (Eemian) interglacial period, when sea level reached 6-9 meters higher than today. Further, Earth is out of energy balance with present atmospheric composition, implying that more warming is in the pipeline, and we show that the growth rate of greenhouse gas climate forcing has accelerated markedly in the past decade. The rapidity of ice sheet and sea level response to global temperature is difficult to predict, but is dependent on the magnitude of warming. Targets for limiting global warming thus, at minimum, should aim to avoid leaving global temperature at Eemian or higher levels for centuries. Such targets now require "negative emissions", i.e., extraction of CO₂ from the air. If phasedown of fossil fuel emissions begins soon, improved agricultural and forestry practices, including reforestation and steps to improve soil fertility and increase its carbon content, may provide much of the necessary CO₂ extraction. In that case, the magnitude and duration of global temperature excursion above the natural range of the current interglacial (Holocene) could be limited and irreversible climate impacts could be minimized. In contrast, continued high fossil fuel emissions today place a burden on young people to undertake massive technological CO₂ extraction if they are to limit climate change and its consequences. Proposed methods of extraction such as bioenergy with carbon capture and storage (BECCS) or air capture of CO₂ have minimal estimated costs of 89-535 trillion dollars this century and also have large risks and uncertain feasibility. Continued high fossil fuel emissions unarguably sentences young people to either a massive, implausible cleanup or growing deleterious climate impacts or both.

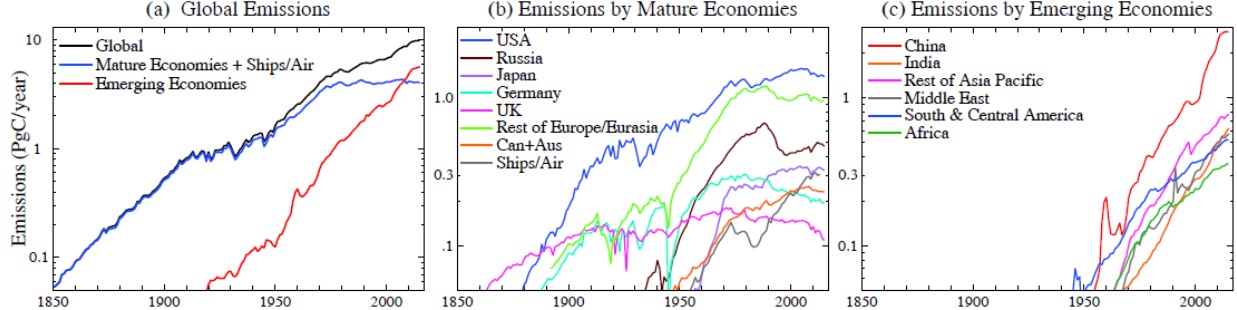

**Figure 1.** Fossil fuel (and cement manufacture) $CO_2$ emissions (note log scale) based on Boden et al. (2016) with BP data used to infer 2014-2015 estimates. Europe/Eurasia is Turkey plus the Boden et al. categories Western Europe and Centrally Planned Europe. Asia Pacific is sum of Centrally Planned Asia, Far East and Oceania. Middle East is Boden et al. Middle East less Turkey. Russia is Russian Federation since 1992 and 0.6 of USSR in 1850-1991. Ships/Air is sum of bunker fuels of all nations. Can+Aus is the sum of emissions from Canada and Australia.

# 1 Introduction

The United Nations 1992 Framework Convention on Climate Change (UNFCCC, 1992) stated its objective as "*...stabilization of GHG concentrations in the atmosphere at a level that would prevent dangerous anthropogenic interference with the climate system.*" The 15th Conference of the Parties (Copenhagen Accord, 2009) concluded that this objective required a goal to "*...reduce global emissions so as to hold the increase of global temperature below 2°C....*" The 21st Conference of the Parties (Paris Agreement, 2015), currently ratified by 148 nations, aims to strengthen the global response to the climate change threat by "[h]olding the increase in the global average temperature to well below 2°C above the pre-industrial levels and pursuing efforts to limit the temperature increase to 1.5°C above the pre-industrial levels."

Global surface temperature has many merits as the principal metric for climate change, but additional metrics, such as atmospheric $CO_2$ amount and Earth's energy imbalance, help refine targets for avoiding dangerous human-made climate change. Paleoclimate data and observations of Earth's present energy imbalance led Hansen et al. (2008, 2013a, 2016) to recommend reducing $CO_2$ to less than 350 ppm, with the understanding that this target must be adjusted as $CO_2$ declines and empirical data accumulates. The 350 ppm $CO_2$ target is moderately stricter than the 1.5°C warming target. The near planetary energy balance anticipated at 350 ppm $CO_2$ implies a global temperature close to recent values, i.e., about +1°C relative to preindustrial.

We advocate pursuit of this goal within a century to limit the period with global temperature above that of the current interglacial period, the Holocene.[1] Limiting the period and magnitude of temperature excursion above the Holocene range is crucial to avoid strong stimulation of slow feedbacks. Slow feedbacks include ice sheet disintegration and thus sea level rise, which is probably the most threatening climate impact, and release of greenhouse gases (GHGs) via such mechanisms as thawing tundra and loss of soil carbon. Holocene climate stability allowed sea level to be stable for the past several millennia (Kopp et al., 2016) as civilizations developed. But there is now a danger that temperature rises so far above the Holocene range that slow feedbacks are activated to a degree that continuing climate change will be out of humanity's control. Both the 1.5°C and 350 ppm targets require rapid phasedown of fossil fuel emissions.

---

[1] By Holocene we refer to the pre-industrial portion of the present interglacial period. As we will show, the rapid warming of the past century has brought temperature above the range in the prior 11,700 years of the Holocene.

Today, global fossil fuel emissions continue at rates that make these targets increasingly improbable (Fig. 1 and Appendix A1). On a per capita historical basis the U.S. is 10 times more accountable than China and 25 times more accountable than India for the increase of atmospheric $CO_2$ above its preindustrial level (Hansen and Sato, 2016). In response, a lawsuit [Juliana et al. vs United States, 2016, hereafter J et al. vs US, 2016] was filed against the United States asking the U.S. District Court, District of Oregon, to require the U.S. government to produce a plan to rapidly reduce emissions. The suit requests that the plan reduce emissions at the 6%/year rate that Hansen et al. (2013a) estimated as the requirement for lowering atmospheric $CO_2$ to a level of 350 ppm. At a hearing in Eugene Oregon on 9 March 2016 the United States and three interveners (American Petroleum Institute, National Association of Manufacturers, and the American Fuels and Petrochemical Association) asked the Court to dismiss the case, in part based on the argument that the requested rate of fossil fuel emissions reduction was beyond the court's authority. Magistrate Judge Coffin stated that he found "the remedies aspect of the plaintiff's complaint [to be] *troublesome*", in part because it involves "a separation of powers issue." But he also noted that some of the alleged climate change consequences, if accurate, could be considered "beyond the pale", and he rejected the motion to dismiss the case. Judge Coffin's ruling was certified, as required, by a second judge (Aiken, 2016) on 9 September 2016, and, barring a settlement that would be overseen by the court, the case is expected to proceed to trial in late 2017. It can be anticipated that the plausibility of achieving the emission reductions needed to stabilize climate will be a central issue at the remedy stage of the trial.

Urgency of initiating emissions reductions is well recognized (IPCC, 2013, 2014; Huntingford et al., 2012; Friedlingstein et al., 2014; Rogelj et al., 2016a) and was stressed in the paper (Hansen et al., 2013a) used in support of the lawsuit J et al. vs US (2016). It is also recognized that the goal to keep global warming less than 1.5°C likely requires negative net $CO_2$ emissions later this century if high global emissions continue in the near-term (Fuss et al., 2014; Anderson, 2015; Rogelj et al., 2015; Sanderson et al., 2016). The Intergovernmental Panel on Climate Change (IPCC) reports (IPCC 2013, 2014) do not address environmental and ecological feasibility and impacts of large-scale $CO_2$ removal, but recent studies (Smith et al., 2016; Williamson 2016) are taking up this crucial issue and raising the question of whether large-scale negative emissions are even feasible.

Our aim is to contribute to understanding of the required rate of $CO_2$ emissions reduction via an approach that is transparent to non-scientists. We consider potential drawdown of atmospheric $CO_2$ by reforestation and afforestation, the potential for improved agricultural practices to store more soil carbon, and potential reductions of non-$CO_2$ GHGs that could reduce human-made climate forcing[2]. Quantitative examination reveals the merits of these actions to partly offset demands on fossil fuel $CO_2$ emission phasedown, but also their limitations, thus clarifying the urgency of government actions to rapidly advance the transition to carbon-free energies to meet the climate stabilization targets they have set.

We first describe the status of global temperature change and then summarize the principal climate forcings that drive long term climate change. We show that observed global warming is consistent with knowledge of changing climate forcings, Earth's measured energy imbalance,

---

[2] A climate forcing is an imposed change of Earth's energy balance, measured in W/m². For example, Earth absorbs about 240 W/m² of solar energy, so if the sun's brightness increases 1% it is a forcing of +2.4 W/m².

and the canonical estimate of climate sensitivity[3], i.e., about 3°C global warming[4] for doubled atmospheric $CO_2$. For clarity we make global temperature calculations with our simple climate
model, which we show (Appendix A2) has a transient climate sensitivity near the midpoint of the sensitivity of models illustrated in Fig. 10.20a of IPCC (2013). The standard climate sensitivity and climate model do not include effects of "slow" climate feedbacks such as change of ice sheet size. There is increasing evidence that some slow feedbacks can be triggered within decades, so they must be given major consideration in establishing the dangerous level of human-made
climate interference. We thus incorporate consideration of slow feedbacks in our analysis and discussion, even though precise specification of their magnitude and time scales is not possible. We present updates of GHG observations and find a notable acceleration during the past decade of the growth rate of GHG climate forcing. For future fossil fuel emissions we consider both the Representative Concentration Pathways (RCP) scenarios used in Climate Model Intercomparison
Project 5 (CMIP5) IPCC studies, and simple emission growth rate changes that help evaluate the plausibility of needed emission changes. We use a Green's function calculation of global temperature with canonical climate sensitivity for each emissions scenario, which yields the amount of $CO_2$ that must be extracted from the air – effectively the climate debt – to return atmospheric $CO_2$ to less than 350 ppm or limit global warming to less than 1.5°C above
preindustrial levels. We discuss alternative extraction technologies and their estimated costs, and finally we consider the potential alleviation of $CO_2$ extraction requirements that might be obtained via special efforts to reduce non-$CO_2$ GHGs.

## 2  Global Temperature Change

The framing of human-caused climate change by the Paris Agreement uses global mean surface temperature as the metric for assessing dangerous climate change. We have previously argued the merits of additional metrics, especially Earth's energy imbalance (Hansen et al., 2005; von Schuckmann et al., 2016) and atmospheric $CO_2$ amount (Hansen et al., 2008). Earth's energy imbalance integrates over all climate forcings, known and unknown, and informs us where
climate is heading, because it is this imbalance that drives continued warming. The $CO_2$ metric has merit because $CO_2$ is the dominant control knob on global temperature (Lacis et al., 2010, 2013), including paleo temperature change (cf. Fig. 28 of Hansen et al., 2016). Our present paper uses these alternative metrics to help sharpen determination of the dangerous level of global warming, and to quantify actions that are needed to stabilize climate. We here use global
temperature as the principal metric because several reasons of concern are scaled to global warming (O'Neill et al., 2017), including specifically the potential for slow feedbacks such as ice sheet melt and permafrost thaw. The slow feedbacks, whose time scales depend on how strongly the climate system is being forced, will substantially determine the magnitude of climate impacts and affect how difficult the task of stabilizing climate will be.
Quantitative assessment of both ongoing and paleo temperature change is needed to define acceptable limits on human-made interference with climate, with paleo climate especially helpful

---

[3] Climate sensitivity is the response of global average surface temperature to a standard forcing, with the standard forcing commonly taken to be doubled atmospheric $CO_2$, which is a forcing of about 4 W/m² (Hansen et al., 2005).
[4] IPCC (2013) finds that 2×$CO_2$ equilibrium sensitivity is likely in the range 3 ± 1.5°C, as was estimated by Charney et al. (1979). Median sensitivity in recent model inter-comparisons is 3.2°C (Andrews et al., 2012; Vial et al., 2013).

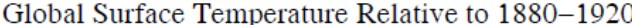

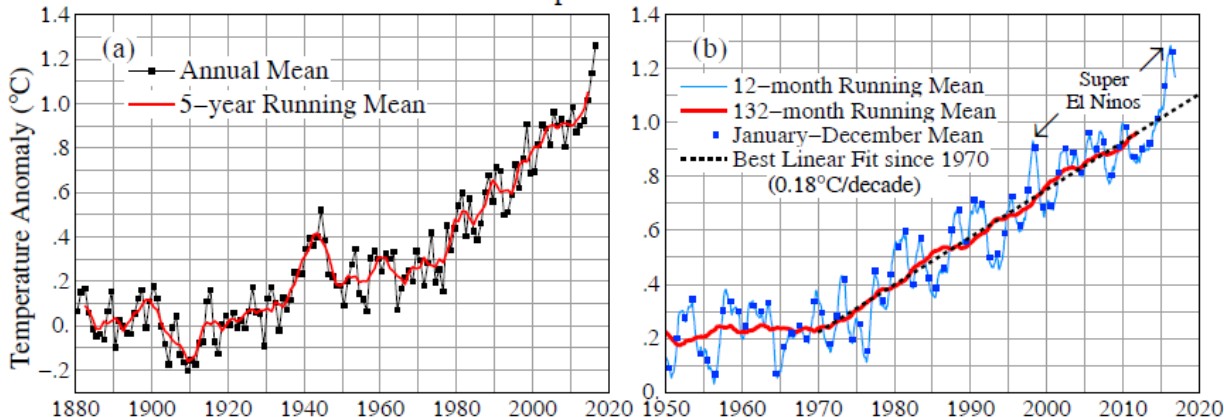

**Figure 2.** Global surface temperature relative to 1880-1920 based on GISTEMP data (Appendix A3). (a) Annual and 5-year means since 1880, (b) 12- and 132-month running means since 1970. Blue squares in (b) are calendar year (Jan-Dec) means used to construct (a). (b) uses data through April 2017.

for characterizing long-term ice sheet and sea level response to temperature change. Thus we examine the modern period with near-global instrumental temperature data in the context of the current and prior (Holocene and Eemian) interglacial periods, for which less precise proxy-based temperatures have recently emerged. The Holocene, over 11,700 years in duration, had relatively stable climate, prior to the remarkable warming in the past half century. The Eemian, which lasted from about 130,000 to 115,000 years ago, was moderately warmer than the Holocene and experienced sea level rise to heights 6-9 m (20-30 feet) greater than today.

## 2.1 Modern Temperature

The several analyses of temperature change since 1880 are in close agreement (Hartmann et al., 2013). Thus we can use the current GISTEMP analysis (see Supporting Information), which is updated monthly and available (http://www.columbia.edu/~mhs119/Temperature/).

The popular measure of global temperature is the annual-mean global-mean value (Fig. 2a), which is publicized at the end of each year. However, as discussed by Hansen et al. (2010), the 12-month running mean global temperature is more informative and removes monthly "noise" from the record just as well as the calendar year average. For example, the 12-month running mean for the past 67 years (Fig. 2b) defines clearly the super-El Niños of 1997-98 and 2015-16 and the 3-year cooling after the Mount Pinatubo volcanic eruption in the early 1990s.

Global temperature in 2014, 2015 and 2016 reached successive record high levels for the period of instrumental data (Fig. 2). Temperature in the latter two years was partially boosted by the 2015-16 El Niño, but the recent warming is sufficient to remove the illusion of a hiatus of global surface warming after the 1997-98 El Niño (Appendix A4).

The present global warming rate, based on a linear fit for 1970-present (dashed line in Fig. 2b) is +0.18°C per decade[5]. The period since 1970 is the time with high growth rate of GHG climate forcing, which has been maintained at approximately +0.4 W/m$^2$/decade (see section 6 below)[6] causing Earth to be substantially out of energy balance (Cheng et al., 2017). The energy

---

[5] Extreme endpoints affect linear trends, but if the 2016 temperature is excluded the calculated trend (0.176°C/decade) still rounds to 0.18°C/decade.

[6] As forcing additions from chlorofluorocarbons (CFCs) and CH$_4$ declined, CO$_2$ growth increased (section 6).

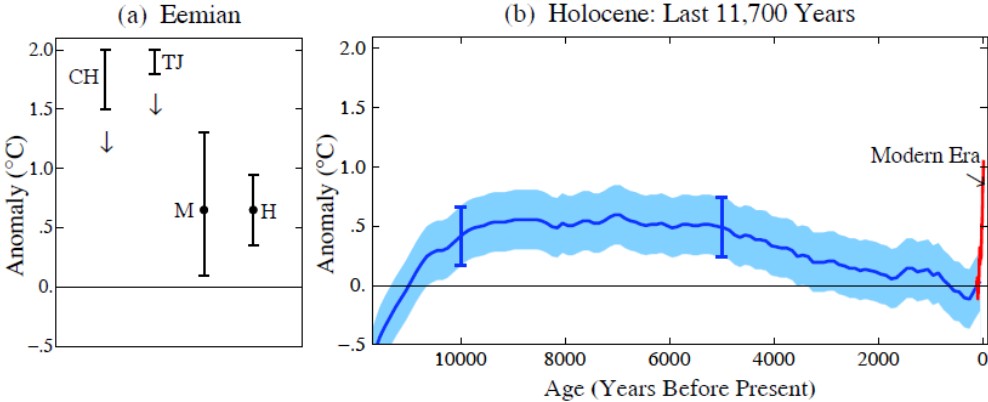

**Figure 3.** Estimated average global temperature for (a) last interglacial (Eemian) period (Clark and Huybers, 2009; Turney and Jones, 2010; McKay et al., 2011; Hoffman et al, 2017), (b) centennially-smoothed Holocene (Marcott et al., 2013) temperature and the 11-year mean of modern data (Fig. 2), as anomalies relative to 1880-1920. Vertical downward arrows indicate likely overestimates (see text).

imbalance drives global warming, so unless and until there is substantial change in the rate of added climate forcing we expect the underlying warming to continue at a comparable rate.

Global temperature defined by the linear fit to temperature since 1970 now exceeds 1°C[7] relative to the 1880-1920 mean (Fig. 2b), where the 1880-1920 mean provides our best estimate of "preindustrial" temperature (Appendix A5). At the rate of 0.18°C/decade the linear trend line of global temperature will reach +1.5°C in about 2040 and +2°C in the late 2060s. However, the warming rate can accelerate or decelerate, depending on policies that affect GHG emissions, developing climate feedbacks, and other factors discussed below.

## 2.2 Temperature during current and prior interglacial periods

Holocene temperature has been reconstructed at centennial-scale resolution from 73 globally distributed proxy temperature records by Marcott et al. (2013). This record shows a decline of 0.6°C from early Holocene maximum temperature to a "Little Ice Age" minimum in the early 1800s [that minimum being better defined by higher resolution data of Abram et al. (2016)]. Concatenation of the modern and Holocene temperature records (Fig. 3; Appendix A5) assumes that 1880-1920 mean temperature is 0.1°C warmer than the Little Ice Age minimum (Abram et al., 2016). The early Holocene maximum in the Marcott et al. (2013) data thus reaches +0.5°C relative to the 1880-1920 mean of modern data. The formal 95% confidence bounds to Holocene temperature (Marcott et al., 2013) are ±0.25°C (blue shading in Fig. 3b), but total uncertainty is larger. Specifically, Liu et al. (2014) points out a bias effect caused by seasonality in the proxy temperature reconstruction. Correction for this bias will tend to push early Holocene temperatures lower, increasing the gap between today's temperature and early Holocene temperature (Marcott and Shakun, 2015).

We emphasize that comparisons of current global temperature with the earlier Holocene must bear in mind the centennial smoothing inherent in the Holocene data (Marcott et al. 2013). Thus the temperature in an anomalous single year such as 2016 is not an appropriate comparison. However, the temperature in 2016 based on the 1970-present linear trend (at least 1°C relative to the 1880-1920 mean) does provide a meaningful comparison. The trend line reduces the effect

---

[7] It is 1.05°C for linear fit to 132-month running mean, but can vary by a few hundredths of a degree depending on the method chosen to remove short-term variability.

of interannual variability, but the more important point is that Earth's energy imbalance assures that this temperature will continue to rise unless and until the global climate forcing begins to decline. In other words, we know that mean temperature over the next several decades will not be lower than 1°C.

We conclude that the modern trend line of global temperature crossed the early Holocene (smoothed) temperature maximum (+0.5°C) in about 1985. This conclusion is supported by the accelerating rate of sea level rise, which approached 3 mm/year at about that date [Hansen et al. (2016) show a relevant concatenation of measurements in their Fig. 29]. Such a high rate of sea level rise, which is 3 meters per millennium, far exceeds the prior rate of sea level rise in the last six millennia of the Holocene (Lambeck et al., 2014). Note that near stability of sea level in the latter half of the Holocene as global temperature fell about 0.5°C, prior to rapid warming of the Modern Era (Fig. 3), is not inconsistent with that global cooling. Hemispheric solar insolation anomalies in the latter half of the Holocene favored ice sheet growth in the Northern Hemisphere and ice sheet decay in Antarctica (Fig. 27a, Hansen et al., 2016), but the Northern Hemisphere did not become cool enough to reestablish ice sheets on North America or Eurasia. There was a small increase of Greenland ice sheet mass (Larsen et al., 2015), but this was presumably at least balanced by Antarctic ice sheet mass loss (Lambeck et al., 2014).

The important point is that global temperature has risen above the centennially-smoothed Holocene range. Global warming is already having substantial adverse climate impacts (IPCC, 2014), including extreme events (NAS, 2016). There is widespread agreement that 2°C warming would commit the world to multi-meter sea level rise (Levermann et al., 2013; Dutton et al., 2015; Clark et al., 2016). Sea level reached 6-9 m higher than today during the Eemian (Dutton et al., 2015), so it is particularly relevant to know how global mean Eemian temperature compares to the preindustrial level and thus to today.

McKay et al. (2011) estimated peak Eemian annual global ocean SST as +0.7°C ± 0.6°C relative to late Holocene temperature, while models, as described by Masson-Delmotte et al. (2013), give more confidence to the lower part of that range. Hoffman et al. (2017) report the maximum Eemian annual global SST as +0.5°C ± 0.3°C relative to 1870-1889, which is +0.65°C relative to 1880-1920. The response of surface air temperature (SAT) over land is twice as large as the SST response to climate forcings in 21st century simulations with models (Collins et al., 2013), in good agreement with observed warming in the industrial era (Appendix A3 this paper, Fig. A3a). The ratio of land SAT change to SST change is reduced only to ~1.8 after 1000 years in climate models (Fig. A6, Appendix A6). This implies that, because land covers ~30% of the globe, SST warmings should be multiplied by 1.24-1.3 to estimate global temperature change. Thus the McKay et al. and Hoffman et al. data are equivalent to a global Eemian temperature of just under +1°C relative to the Holocene. Clark and Huybers (2009) and Turney and Jones (2010) estimated global temperature in the Eemian as 1.5-2°C warmer than the Holocene (Fig. 3), but Bakker and Renssen (2014) point out two biases that may cause this range to be an overestimate. Bakker and Rennsen (2014) use a suite of models to estimate that the assumption that maximum Eemian temperature was synchronous over the planet overestimates Eemian temperature by 0.4 ± 0.3°C – a feature supported by a lack of synchroneity of warmest conditions in assessments with improved synchronization of records (Govin et al., 2015) – and that they also suggest that a possible seasonal bias of proxy temperature could make the total overestimate as large as 1.1 ± 0.4°C. Given uncertainties in the corrections, it becomes a matter of expert judgment. Dutton et al. (2015) conclude that the best estimate for Eemian temperature

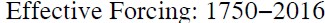

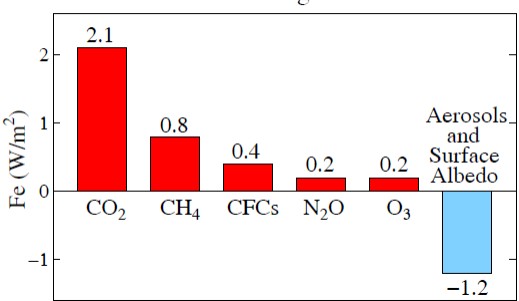

**Figure 4.** Estimated effective climate forcings [update through 2016 of Fig. 28b of Hansen et al. (2005), which are consistent with estimates of Myhre et al. (2013) in the most recent IPCC report (IPCC, 2013). Forcings are based on observations of each gas, except simulated $CH_4$-induced changes of $O_3$ and stratospheric $H_2O$ included in the $CH_4$ forcing. Aerosols and surface albedo change are estimated from historical scenarios of emissions and land use. Oscillatory and intermittent natural forcings (solar irradiance and volcanoes) are excluded. CFCs include not only chlorofluorocarbons, but all Montreal Protocol Trace Gases (MPTGs) and Other Trace Gases (OTGs). Uncertainties (for 5-95% confidence) are 0.6 W/m$^2$ for total GHG forcing and 0.9 W/m$^2$ for aerosol forcing (Myhre et al., 2013).

is +1°C relative to preindustrial. Consistent with these estimates and the discussion of Masson-Delmotte et al. (2013), we assume that maximum Eemian temperature was +1°C relative to preindustrial with an uncertainty of at least 0.5°C.

These considerations raise the question of whether 2°C, or even 1.5°C, is an appropriate target to protect the well-being of young people and future generations. Indeed, Hansen et al. (2008) concluded "*If humanity wishes to preserve a planet similar to that on which civilization developed and to which life on Earth is adapted, ... CO$_2$ will need to be reduced ... to at most 350 ppm, but likely less than that.*" And further "*If the present overshoot of the target CO$_2$ is not brief, there is a possibility of seeding irreversible catastrophic effects.*"

A danger of 1.5°C or 2°C targets is that they are far above the Holocene temperature range. If such temperature levels are allowed to long exist they will spur "slow" amplifying feedbacks (Hansen et al., 2013b; Rohling et al., 2013; Masson-Delmotte et al., 2013), which have potential to run out of humanity's control. The most threatening slow feedback likely is ice sheet melt and consequent significant sea level rise, as occurred in the Eemian, but there are other risks in pushing the climate system far out of its Holocene range. Methane release from thawing permafrost and methane hydrates is another potential feedback, for example, but the magnitude and time scale of this is unclear (O'Connor et al., 2010; Quiquet, 2015).

Here we examine the fossil fuel emission reductions required to restore atmospheric CO$_2$ to 350 ppm or less, so as to keep global temperature close to the Holocene range, in addition to the canonical 1.5°C and 2°C targets. Quantitative investigation requires consideration of Earth's energy imbalance, changing climate forcings, and climate sensitivity.

## 3  Global Climate Forcings and Earth's Energy Imbalance

The dominant human-caused drivers of climate change are changes of atmospheric GHGs and aerosols (Fig. 4). GHGs absorb Earth's infrared (heat) radiation, thus serving as a "blanket" warms Earth's surface by reducing heat radiation to space. Aerosols, fine particles/droplets in the air that cause visible air pollution, both reflect and absorb solar radiation, but reflection of solar energy to space is their dominant effect, so they cause a cooling that partly offsets GHG

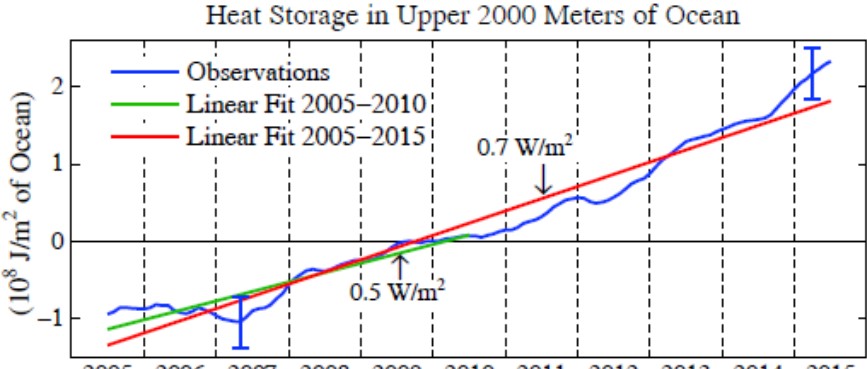

**Figure 5.** Ocean heat uptake in upper 2 km of ocean during 11 years 2005-2015 using analysis method of von Schuckmann and LeTraon (2011). Heat uptake in $W/m^2$ (0.5 and 0.7) refer to global (ocean + land) area, i.e., it is the contribution of the upper ocean to the heat uptake averaged over the entire planet.

warming. Estimated forcings (Fig. 4), an update of Fig. 28b of Hansen et al. (2005), are similar to those of Myhre et al. (2013) in the most recent IPCC report (IPCC, 2013).[8]

  Climate forcings in Fig. 4 are the planetary energy imbalance that would be caused by the preindustrial-to-present change of each atmospheric constituent, if the climate were held fixed at its preindustrial state (Hansen et al., 2005). The $CH_4$ forcing includes its indirect effects, as
increasing atmospheric $CH_4$ causes tropospheric ozone ($O_3$) and stratospheric water vapor to increase (Myhre et al., 2013). Uncertainties, discussed by Myhre et al. (2013), are typically 10-15% for GHG forcings. The aerosol forcing uncertainty, described by a probability distribution function (Boucher et al., 2013), is of order 50%. Our estimate of aerosol plus surface albedo forcing ($-1.2 W/m^2$) differs from the $-1.5 W/m^2$ of Hansen et al. (2005), as discussed below, but
both are within the range of the distribution function of Boucher et al. (2013).

  Positive net forcing (Fig. 4) causes Earth to be temporarily out of energy balance, with more energy coming in than going out, which drives slow global warming. Eventually Earth will become hot enough to restore planetary energy balance. However, because of the ocean's great thermal inertia (heat capacity), full atmosphere-ocean response to the forcing requires a long
time: atmosphere-ocean models suggest that even after 100 years only 60-75% of the surface warming for a given forcing has occurred, the remaining 25-40% still being "in the pipeline" (Hansen et al., 2011; Collins et al., 2013). Moreover, we outline in the next section that global warming can activate "slow" feedbacks, such as changes of ice sheets or melting of methane hydrates, so the time for the system to reach a fully equilibrated state is even longer.

GHGs have been increasing for more than a century and Earth has partially warmed in response. Earth's energy imbalance is the portion of the forcing that has not yet been responded to. This imbalance thus defines additional global warming that will occur without further change of forcings. Earth's energy imbalance can be measured by monitoring ocean subsurface temperatures, because almost all excess energy coming into the planet goes into the ocean (von
Schuckmann et al., 2016). Most of the ocean's heat content change occurs in the upper 2000 m (Levitus et al., 2012), which has been well measured since 2005 when the distribution of Argo

---

[8] Our GHG forcings, calculated with formulae of Hansen et al. (2000), yield a $CO_2$ forcing 6.7% larger than the central IPCC estimate [Table 8.2 of Myhre et al. (2013)] for the $CO_2$ change from 1750-2011. For all well-mixed (long-lived) GHGs we obtain 3.03 $W/m^2$, which is within the IPCC range 2.83 ±0.29 $W/m^2$.

floats achieved good global coverage (von Schuckmann and Le Traon, 2011). Here we update the von Schuckmann and Le Traon analysis with data for 2005-2015 (Fig. 5) finding a decade-average 0.7 $W/m^2$ heat uptake in the upper 2000 m of the ocean. Addition of the smaller terms raises the imbalance to $+0.75 \pm 0.25$ $W/m^2$ averaged over the solar cycle (Appendix A7).

## 4 Climate Sensitivity and Feedbacks

Climate sensitivity has been a fundamental issue at least since the 19[th] century when Tyndall (1861) and Arrhenius (1896) stimulated interest in the effect of $CO_2$ change on climate. Evaluation of climate sensitivity involves the full complexity of the climate system, as all components and processes in the system are free to interact on all time scales. Tyndall and Arrhenius recognized some of the most important climate feedbacks on both fast and slow time scales. The amount of water vapor in the air increases with temperature, which is an amplifying feedback because water vapor is a very effective greenhouse gas; this is a "fast" feedback, because water vapor amount in the air adjusts within days to temperature change. The area covered by glaciers and ice sheets is a prime "slow" feedback; it, too, is an amplifying feedback, because the darker surface exposed by melting ice absorbs more sunlight.

Diminishing climate feedbacks also exist. Cloud-cover changes, e.g., can either amplify or reduce climate change (Boucher at al., 2013). Thus it is not inherent that amplifying feedbacks should be dominant, but climate models and empirical data concur that amplifying feedbacks dominate on both short and long time scales, as we will discuss. Amplifying feedbacks lead to large climate change in response to even weak climate forcings such as ice age cycles caused by small perturbations of Earth's orbit, and still larger climate change occurs on even longer time scales in response to gradual changes in the balance between natural sources and sinks of atmospheric $CO_2$ (Zachos et al., 2001; Royer et al., 2012; Franks et al., 2014).

## 4.1 Fast-Feedback Climate Sensitivity

Doubled atmospheric $CO_2$, a forcing of ~4 $W/m^2$, is a standard forcing in studies of climate sensitivity. Charney et al. (1979) concluded that equilibrium sensitivity, i.e., global warming after a time sufficient for the planet to restore energy balance with space, was $3°C \pm 1.5°C$ for $2\times CO_2$ or $0.75°C$ per $W/m^2$ forcing. The Charney analysis was based on climate models in which ice sheets and all long-lived GHGs (except for the specified $CO_2$ doubling) were fixed. The climate sensitivity thus inferred is the "fast-feedback" climate sensitivity. The central value found in a wide range of modern climate models (Flato et al., 2013) remains $3°C$ for $2\times CO_2$.

The possibility of unknown unknowns in models would keep the uncertainty in the fast-feedback climate sensitivity high, if it were based on models alone, but as discussed by Rohling et al. (2012a), paleoclimate data allow narrowing of the uncertainty. Ice sheet size and the atmospheric amount of long-lived GHGs ($CO_2$, $CH_4$, $N_2O$) under natural conditions change on multi-millennial time scales. These changes are so slow that the climate is in quasi-equilibrium with the changing surface condition and long-lived GHG amounts. Thus these changing boundary conditions, along with knowledge of the associated global temperature change, allow empirical assessment of the fast-feedback climate sensitivity. The central result agrees well with the model-based climate sensitivity estimate of $3°C$ for $2\times CO_2$ (Rohling et al., 2012b), with an uncertainty that is arguably $1°C$ or less (Hansen et al., 2013b).

The ocean has great heat capacity (thermal inertia), so it takes decades to centuries for Earth's surface temperature to achieve most of its fast-feedback response to a change of climate forcing (Hansen et al., 1985). Thus Earth has only partly responded to the human-made increase of GHGs in the air today, the planet must be out of energy balance with the planet gaining energy (via reduced heat radiation to space), and more global warming is "in the pipeline."

A useful check on understanding of ongoing climate change is provided by the consistency of the net climate forcing (Fig. 4), Earth's energy imbalance, observed global warming, and climate sensitivity. Observed warming since 1880-1920 is 1.05°C[9] based on the linear fit to the 132-month running mean (Fig. 2b), which limits bias from short-term oscillations. Global warming between 1700-1800 and 1880-1920 was ~0.1°C (Abram et al., 2016; Hawkins et al., 2017; Marcott et al., 2013), so 1750-2015 warming is ~1.15°C. Taking climate sensitivity as 0.75°C per $W/m^2$ forcing, global warming of 1.15°C implies that 1.55 $W/m^2$ of the total 2.5 $W/m^2$ forcing has been "used up" to cause observed warming. Thus 0.95 $W/m^2$ forcing should remain to be responded to, i.e., the expected planetary energy imbalance is 0.95 $W/m^2$, which is reasonably consistent with the observed $0.75 \pm 0.25$ $W/m^2$. If we instead take the aerosol + surface albedo forcing as $-1.5$ $W/m^2$, as estimated by Hansen et al. (2005), the net climate forcing is 2.2 $W/m^2$ and the forcing not responded to is 0.65 $W/m^2$, which is also within the observational error of Earth's energy imbalance.

## 4.2 Slow Climate Feedbacks

Large glacial-to-interglacial climate oscillations occur on time scales of tens and hundreds of thousands of years, with atmospheric $CO_2$ amount and the size of ice sheets (and thus sea level) changing almost synchronously on these time scales (Masson-Delmotte et al., 2013). It is readily apparent that these climate cycles are due to small changes in Earth's orbit and the tilt of its spin axis, which alter the geographical and seasonal distribution of sunlight striking Earth. The large climate response is a result of two amplifying feedbacks: (1) atmospheric GHGs (mainly $CO_2$ but accompanied by $CH_4$ and $N_2O$), which increase as Earth warms and decrease as it cools (Ciais et al., 2013), thus amplifying the temperature change, and (2) the size of ice sheets, which shrink as Earth warms and grow as it cools, thus changing the amount of absorbed sunlight in the sense that also amplifies the climate change. For example, 20,000 years ago most of Canada and parts of the United States were covered by an ice sheet, and sea level was about 130 m (~400 feet) lower than today. Global warming of ~5°C between the last glacial maximum and the Holocene (Masson-Delmotte et al., 2013) is accounted for almost entirely by radiative forcing caused by decrease in ice sheet area and increase of GHGs (Lorius et al., 1990; Hansen et al., 2007).

      The glacial-interglacial time scale is set by the time scale of the weak orbital forcings. Before addressing the crucial issue of the inherent time scale of slow feedbacks, we need to say more about the two dominant slow feedbacks, described above as ice sheets and GHGs.

      The ice sheet feedback works mainly via the albedo (reflectivity) effect. A shrinking ice sheet exposes darker ground and warming darkens the ice surface by increasing the area and period with wet ice, thus increasing the ice grain size and increasing the surface concentration of

---

[9] The IPCC (2013; p. 37 of Technical Summary) estimate of warming for 1880-2012 is 0.85°C [range 0.65 to 1.06°C]. While within that range, our value is higher because (1) use of 4-year longer period, (2) warming in the past few years eliminates the effect on the 1970-present trend from a seeming 1998-2012 warming hiatus, (3) the GISTEMP analysis has greater coverage of the large Arctic warming than the other analyses [Fig. TS.2, p. 39 of IPCC (2013)].

light-absorbing impurities (Tedesco et al., 2016).  The ice albedo effect is supplemented by a
change of surface albedo in ice-free regions due to vegetation changes.  This vegetation albedo
effect provides a significant amplification of warming as Earth's temperature increases from its
present climate state, because dark forests tend to replace tundra or sparse low-level vegetation
in large areas of Eurasia and North America (Lunt et al., 2010).

The GHG feedback on glacial-interglacial time scales is 75-80 percent from $CO_2$ change;
$N_2O$ and $CH_4$ account for 20-25 percent (Lorius et al., 1990, Hansen et al., 2007, Masson-
Delmotte et al., 2013).  In simple terms, the ocean and land release more of these gases as the
planet becomes warmer.  Mechanisms that control GHG release as Earth warms, and GHG
drawdown as Earth cools, are complex, including many processes that affect the distribution of
carbon, among the ocean, atmosphere, and biosphere (Yu et al., 2016; Ciais et al., 2013 and
references therein).  Release of carbon from methane hydrates and permafrost contributed to
climate change in past warm periods (Zachos et al., 2008; DeConto et al., 2012) and potentially
could have a significant effect in the future (O'Connor et al., 2010; Schädel et al., 2016).

Paleoclimate data help assess the possible time scale for ice sheet change.  Ice sheet size,
judged from sea level, varies almost synchronously with temperature for the temporal resolution
available in paleoclimate records, but Grant et al. (2012) find that sea level change lags
temperature change by 1-4 centuries.  Paleoclimate forcing, however, is both weak and very
slow, changing on millennial time scales.  Hansen (2005, 2007) argues on heuristic grounds that
the much faster and stronger human-made climate forcing projected this century with continued
high fossil fuel emissions, equivalent to doubling atmospheric $CO_2$, would likely lead to
substantial ice sheet collapse and multi-meter sea level rise on the time scale of a century.
Modeling supports this conclusion, as Pollard et al. (2015) found that addition of hydro-
fracturing and cliff failure to their ice sheet model not only increased simulated sea level rise
from 2 m to 17 m in response to 2°C ocean warming, it also accelerated the time for multi-meter
change from several centuries to several decades.  Ice sheet modeling of Applegate et al. (2015)
explicitly shows that the time scale for large ice sheet melt decreases dramatically as the
magnitude of warming increases.  Hansen et al. (2016), based on a combination of climate
modeling, paleo data, and modern observations, conclude that continued high GHG emissions
would likely cause multi-meter sea level rise within 50-150 years.

The GHG feedback plays a leading role in determining the magnitude of paleoclimate
change and there is reason to suspect that it may already be important in modern climate.  Rising
temperatures increase the rate of $CO_2$ and $CH_4$ release from drying soils, thawing permafrost
(Schädel et al., 2016; Schuur et al., 2015) and warming continental shelves (Kvenvolden, 1993;
Judd et al., 2002), and affect the ocean carbon cycle as noted above.  Crowther et al. (2016)
synthesize results of 49 field experiments across North America, Europe and Asia, inferring that
every 1°C global mean soil surface warming can cause a 30 PgC soil carbon loss and suggesting
that continued high fossil fuel emissions might drive 2°C soil warming and a 55 PgC soil carbon
loss by 2050.  Although this analysis admits large uncertainty, such large soil carbon loss could
wreak havoc with efforts to achieve the net soil and biospheric carbon storage that is likely
necessary for climate stabilization, as we discuss in subsequent sections.

Recent changes of GHGs result mainly from industrial and agricultural emissions, but they
also include any existing climate feedback effects.  $CO_2$ and $CH_4$ are the largest forcings (Fig. 4),
so it is especially important to examine their ongoing changes.

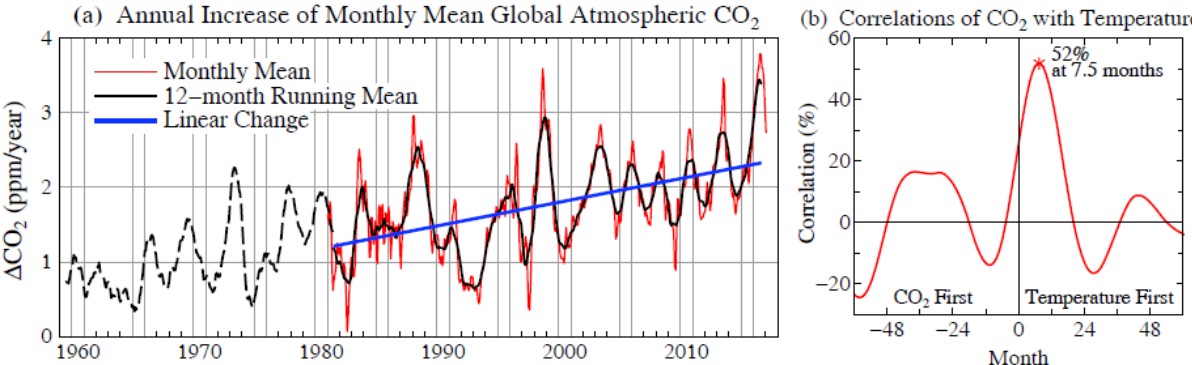

**Figure 6.** (a) Global $CO_2$ annual growth based on NOAA data (http://www.esrl.noaa.gov/gmd/ccgg/trends/). Dashed curve is for a single station (Mauna Loa). Red curve is monthly global mean relative to the same month of prior year; black curve is 12-month running mean of red curve. (b) $CO_2$ growth rate is highly correlated with global temperature, the $CO_2$ change lagging global temperature change by 7-8 months.

# 5 Observed $CO_2$ and $CH_4$ Growth Rates

Annual increase of atmospheric $CO_2$, averaged over a few years, grew from less than 1 ppm/year 50 years ago to more than 2 ppm/year today (Fig. 6), with global mean $CO_2$ now exceeding 400 ppm (Betts et al., 2016). Growth of atmospheric $CO_2$ is about half of fossil fuel $CO_2$ emissions as discussed in Appendix A8 and illustrated in Fig. A8. The large oscillations of annual growth are correlated with global temperature and with the El Niño/La Niña cycle, as discussed in Appendix A8. Recent global temperature anomalies peaked in February 2016, so as expected the $CO_2$ growth rate has been declining for the past several months (Fig. 6a).

Atmospheric $CH_4$ stopped growing between 1998 and 2006, indicating that its sources were nearly in balance with the atmospheric oxidation sink, but growth resumed in the past decade (Fig. 7). $CH_4$ growth averaged 10 ppb/year in 2014-2016, almost as fast as in the 1980s. Likely reasons for the recent increased growth of $CH_4$ are discussed in Appendix A8.

The continued growth of atmospheric $CO_2$ and reaccelerating growth of $CH_4$ raise important questions related to prospects for stabilizing climate. How consistent with reality are scenarios for phasing down climate forcing when tested by observational data? What changes to industrial and agricultural emissions are required to stabilize climate? We address these issues below.

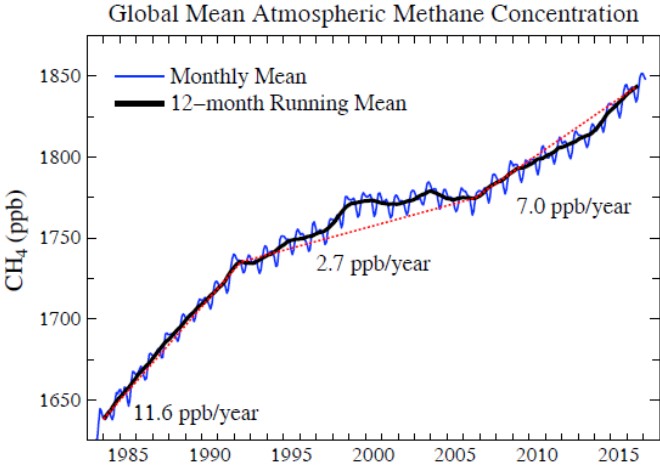

**Figure 7.** Global $CH_4$ from Dlugokencky (2016), NOAA/ESRL (www.esrl.noaa.gov/gmd/ccgg/trends_ch4/). End months for three indicated slopes are January 1984, May 1992, August 2006, and February 2017.

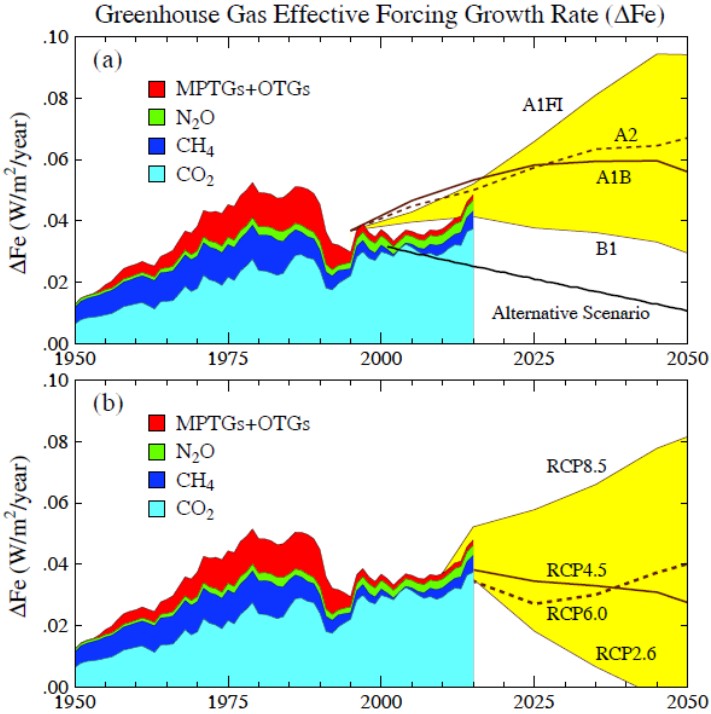

**Figure 8.** GHG climate forcing annual growth rate (ΔFe) with historical data being 5-year running means, except 2015 is a 3-year mean. (a) includes scenarios used in IPCC AR3 and AR4 reports, and (b) has AR5 scenarios. GHG amounts are from NOAA/ESRL Global Monitoring Division. $O_3$ changes are not fully included, as they are not well-measured, but its tropospheric changes are partially included via the effective $CH_4$ forcing. Effective climate forcing (Fe), MPTGs and OTGs are defined in Fig. 4 caption.

## 6 GHG Climate Forcing Growth Rates and Emission Scenarios

Insight is obtained by comparing the growth rate of GHG climate forcing based on observed GHG amounts with past and present GHG scenarios. We examine forcings of IPCC Special Report on Emissions Scenarios (IPCC SRES, 2000) used in the 2001 AR3 and 2007 AR4 reports (Fig. 8a) and Representative Concentration Pathways scenarios (RCP: Moss et al., 2010; Meinshausen et al., 2011a) used in the 2013 IPCC AR5 report (Fig. 8b). We include the "alternative scenario" of Hansen et al. (2000) in which $CO_2$ and $CH_4$ emissions decline such that global temperature stabilizes near the end of the century.[10] We use the same radiation equations for observed GHG amounts and scenarios, so errors in the radiation calculations do not alter the comparison. Equations for GHG forcings are from Table 1 of Hansen et al. (2000) with the $CH_4$ forcing using an efficacy factor 1.4 to include effects of $CH_4$ on tropospheric $O_3$ and stratospheric $H_2O$ (Hansen et al., 2005).

The growth of GHG climate forcing peaked at ~0.05 W/m²/year (5 W/m²/century) in 1978-1988, then falling to a level 10-25% below IPCC SRES (2000) scenarios during the first decade of the 21st century (Fig. 8a). The decline was due to (1) decline of the airborne fraction of $CO_2$ emissions (Fig. A8), (2) slowdown of $CH_4$ growth (Fig. 7), and (3) the Montreal Protocol, which initiated phase-out of the production of gases that destroy stratospheric ozone, primarily chlorofluorocarbons (CFCs).

---

[10]This scenario is discussed by Hansen and Sato (2004). $CH_4$ emissions decline moderately, producing a small negative forcing. $CO_2$ emissions (not captured and sequestered) are assumed to decline until in 2100 fossil fuel emissions just balance uptake of $CO_2$ by the ocean and biosphere. $CO_2$ emissions continue to decline after 2100.

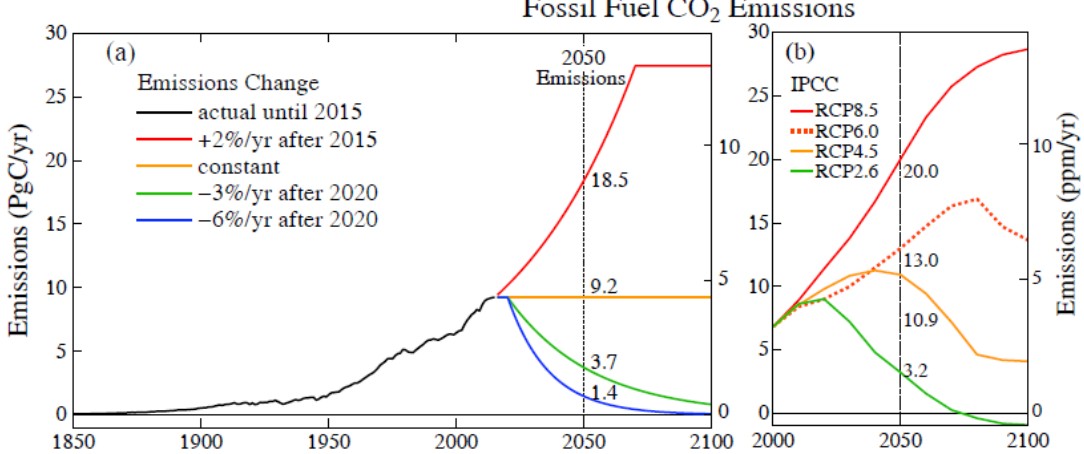

**Figure 9.** Fossil fuel emission scenarios. (a) Scenarios with simple specified rates of emission increase or decrease. (b) IPCC (2013) RCP scenarios. Note: 1 ppm atmospheric $CO_2$ is ~2.12 GtC.

The 2013 IPCC RCP scenarios (Fig. 8b) use observed GHG amounts up to 2005 and diverge thereafter, fanning out into an array of potential futures driven by assumptions about energy demand, fossil fuel prices, and climate policy, chosen to be representative of an extensive literature on possible emissions trajectories (Moss et al., 2010; van Vuuren et al., 2011; Meinshausen et al., 2011a,b). Numbers on the RCP scenarios (8.5, 6.0, 4.5 and 2.6) refer to the GHG climate forcing (W/m$^2$) in 2100.

Scenario RCP2.6 has the world moving into negative growth (net contraction) of GHG forcing 25 years from now (Fig. 8b), through rapid reduction of GHG emissions, along with $CO_2$ capture and storage. Already in 2015 there is a huge gap between reality and RCP2.6. Closing the gap (0.01 W/m$^2$) between actual growth of GHG climate forcing in 2015 and RCP2.6 (Fig. 8b), with $CO_2$ alone, would require extraction from the atmosphere of more than 0.7 ppm of $CO_2$ or 1.5 PgC due to the emissions gap of a single year (2015). We discuss the plausibility and estimated costs of scenarios with $CO_2$ extraction in Section 9.

As a complement to RCP scenarios, we define scenarios with focus on the dominant climate forcing, $CO_2$, with its changes defined simply by percent annual emission decrease or increase. Below (Sec. 10.1 and Appendix A13) we conclude that efforts to limit non-$CO_2$ forcings could keep their growth small or even slightly negative, so a focus on long-lived $CO_2$ is appropriate. Thus for the non-$CO_2$ GHGs we use RCP6.0, a scenario with small changes of these gases. For $CO_2$ we consider rates −6%/year, −3%/year, constant emissions, and +2%/year; emissions stop increasing in the +2%/year case when they reach 25 Gt/year (Fig. 9a). Scenarios with decreasing emissions are preceded by constant emissions for 2015-2020, in recognition that some time is required to achieve policy change and implementation. Note similarity of RCP 2.6 with −3%/year, RCP 4.5 with constant emissions, and RCP 8.5 with +2%/year (Fig. 9).

## 7  Future CO₂ for Assumed Emission Scenarios

We must model Earth's carbon cycle, including ocean uptake of carbon, deforestation, forest regrowth and carbon storage in the soil, for the purpose of simulating future atmospheric $CO_2$ as a function of the fossil fuel emission scenario. Fortunately, the convenient dynamic-sink pulse-response function version of the well-tested Bern carbon cycle model (Joos et al., 1996) does a good job of approximating more detailed models, and it produces a good match to observed

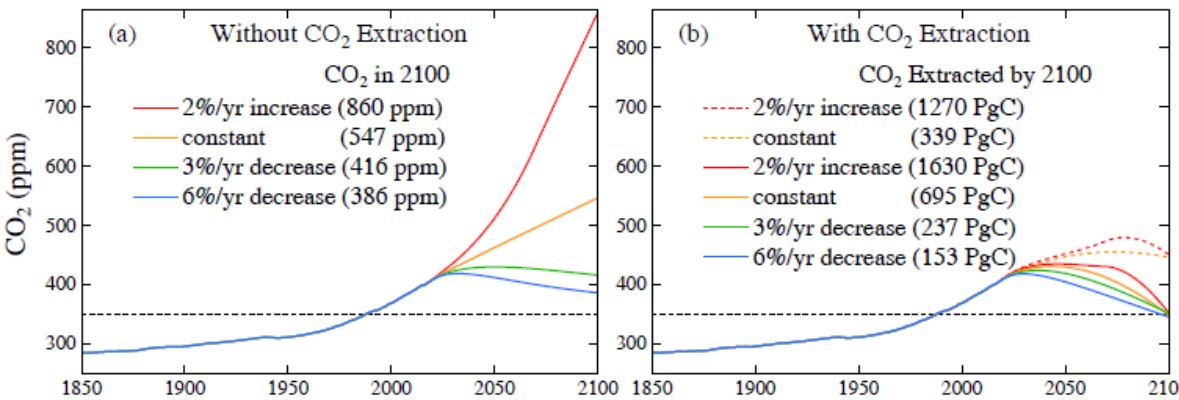

**Figure 10.** (a) Atmospheric $CO_2$ for Fig. 9a emission scenarios. (b) Atmospheric $CO_2$ including effect of $CO_2$ extraction that increases linearly after 2020 (after 2015 in +2%/year case).

industrial-era atmospheric $CO_2$. Thus we use this relatively simple model, described elsewhere (Joos et al., 1996; Kharecha and Hansen, 2008, and references therein), to examine the effect of alternative fossil fuel use scenarios on the growth or decline of atmospheric $CO_2$. Assumptions about emissions in the historical period are given in Appendix A9.

Figure 10a shows the simulated atmospheric $CO_2$ for the baseline emission cases (Fig. 9a). These cases do not include active $CO_2$ removal. Five additional cases including $CO_2$ removal (Fig. 10b) achieve atmospheric $CO_2$ targets of either 350 ppm or 450 ppm in 2100, with cumulative removal amounts listed in parentheses (Fig. 10b). The rate of $CO_2$ extraction in all cases increases linearly from zero in 2010 to the value in 2100 that achieves the atmospheric $CO_2$ target (350 ppm or 450 ppm). The amount of $CO_2$ that must be extracted from the system exceeds the difference between the atmospheric amount without extraction and the target amount, e.g., constant $CO_2$ emissions and no extraction yields 547 ppm for atmospheric $CO_2$ in 2100, but to achieve a target of 350 ppm the required extraction is 328 ppm, not 547 – 350 = 197 ppm. The well-known reason (Cao and Caldeira, 2010) is that ocean outgassing increases and vegetation productivity and ocean $CO_2$ uptake decrease with decreasing atmospheric $CO_2$, as explored in a wide range of Earth System models (Jones et al., 2016).

## 8 Simulations of Global Temperature Change

Analysis of future climate change, and policy options to alter that change, must address various uncertainties. One useful way to treat uncertainty is to use results of many models and construct probability distributions (Collins et al., 2013). Such distributions have been used to estimate the remaining budget for fossil fuel emissions for a specified likelihood of staying under a given global warming limit and to compare alternative policies for limiting climate forcing and global warming (Rogelj et al., 2016a,b).

Our aim here is a fundamental, transparent calculation that clarifies how future warming depends on the rate of fossil fuel emissions. We use best estimates for basic uncertain quantities such as climate sensitivity. If these estimates are accurate, actual temperature should have about equal chances of falling higher or lower than the calculated value. Important uncertainties in projections of future climate change include climate sensitivity, the effects of ocean mixing and dynamics on the climate response function discussed below, and aerosol climate forcing. We provide all defining data so that others can easily repeat calculations with alternative choices.

One clarification is important for our present paper. The climate calculations in this section include only fast-feedbacks, which is also true for most climate simulations by the scientific community for IPCC (2013). This is not a limitation for the past, i.e., for the period 1850-present, because we employ measured GHG changes, which include any GHG change due to slow feedbacks. Also we know that ice sheets did not change significantly in size in that period; there may have been some change in Greenland's albedo and expansion of forests in the Northern Hemisphere (Pearson et al., 2013), but those feedbacks so far have only a small global effect. However, this limitation to fast feedbacks may soon become important; it is only in the past few decades that global temperature rose above the prior Holocene range and only in the past two years that it shot far above that range. This limitation must be borne in mind when we consider the role of slow feedbacks in establishing the dangerous level of warming.

We calculate global temperature change T at time t in response to any climate forcing scenario using the Green's function (Hansen, 2008)

$$T(t) = \int_{1850}^{t} R(t\text{-}t') \, [dF(t')/dt'] \, dt' + Fv \times R(t - 1850) \tag{1}$$

where R(t') is the product of equilibrium global climate sensitivity and the dimensionless climate response function (percent of equilibrium response), dF(t')/dt' is the annual increment of the net forcing, and Fv is the negative of the average volcanic aerosol forcing during the few centuries preceding 1850. Fv ×R(t) is a small correction term that prevents average volcanic aerosol activity from causing a long-term cooling, i.e., it accounts for the fact that the ocean in 1850 was slightly cooled by prior volcanoes. We take $Fv = 0.3 \text{ W/m}^2$, the average stratospheric aerosol forcing for 1850-2015. The assumed-constant pre-1850 volcanic aerosols caused a constant cooling up to 1850, which gradually decreases to zero after 1850 and is replaced by post-1850 time-dependent volcanic cooling; note that T(1850) = 0°C. We use the "intermediate" response function in Fig. 5 of Hansen et al. (2011), which gives good agreement with Earth's measured energy imbalance. The response function is 0.15, 0.55, 0.75 and 1 at years 1, 10, 100 and 2000 with these values connected linearly in log (year). This defined response function allows our results to be exactly reproduced, or altered with alternative choices for climate forcings, climate sensitivity and response function. Forcings that we use are tabulated in Appendix A10.

We use equilibrium fast-feedback climate sensitivity 0.75°C per $W/m^2$ (3°C for $2 \times CO_2$). This is consistent with climate models (Collins et al., 2013: Flato et al., 2013) and paleoclimate evidence (Rohling et al., 2012a; Masson-Delmotte et al., 2013; Bindoff and Stott, 2013). We use RCP6.0 for the non-$CO_2$ GHGs.

We take tropospheric aerosol plus surface albedo forcing as $-1.2 \text{ W/m}^2$ in 2015, presuming the aerosol and albedo contributions to be $-1 \text{ W/m}^2$ and $-0.2 \text{ W/m}^2$, respectively. We assume a small increase this century as global population rises and increasing aerosol emission controls in emerging economies tend to be offset by increasing development elsewhere, so aerosol + surface forcing is $-1.5 \text{ W/m}^2$ in 2100. The temporal shape of the historic aerosol forcing curve (Table A10) is from Hansen et al. (2011), which in turn was based on the Novakov et al. (2003) analysis of how aerosol emissions have changed with technology change.

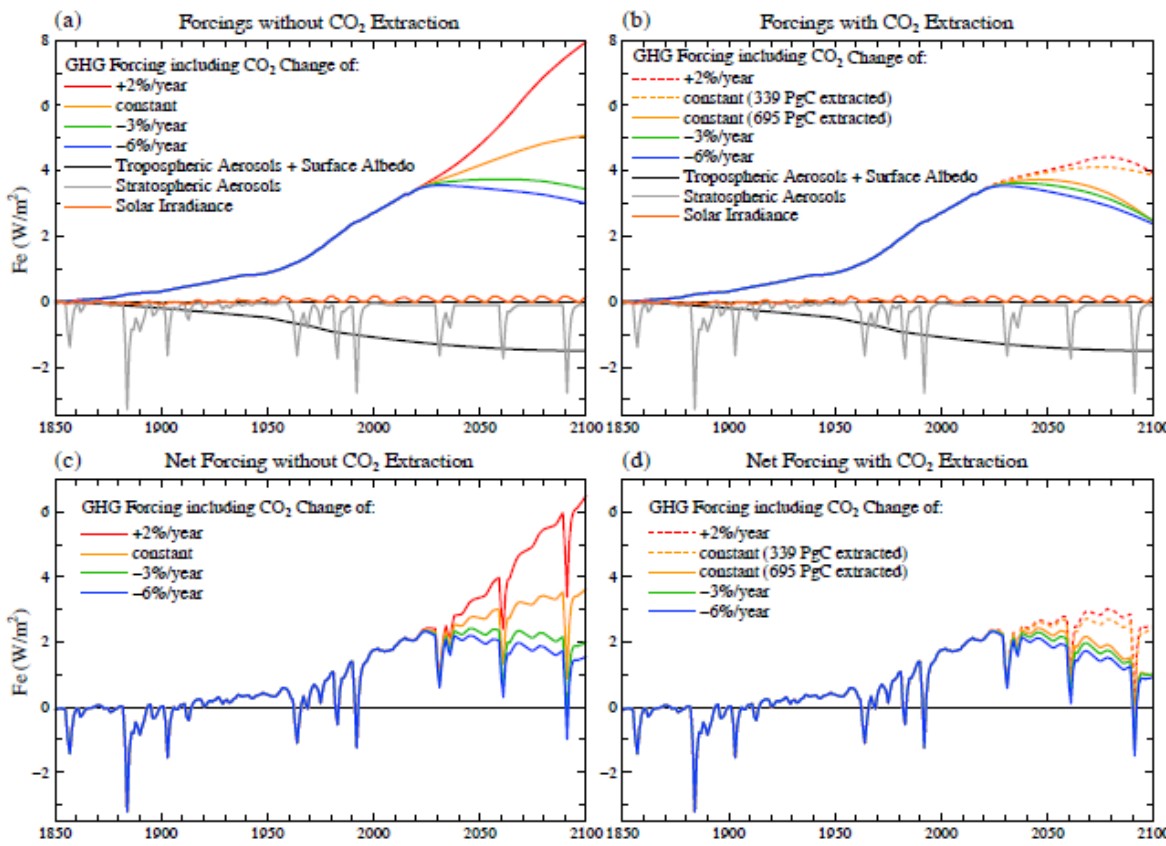

**Figure 11.** Climate forcings used in our climate simulations; Fe is effective forcing, as discussed in connection with Fig. 4. (a) Future GHG forcing uses four alternative fossil fuel emission growth rates. (b) GHG forcings are altered based on $CO_2$ extractions of Fig. 10.

Historic stratospheric aerosol data (Table A10, annual version), an update of Sato et al. (1993), include moderate 21st century aerosol amounts (Bourassa et al., 2012). Future aerosols, for realistic variability, include three volcanic eruptions in the rest of this century with properties of the historic Agung, El Chichon and Pinatubo eruptions, plus a background stratospheric aerosol forcing −0.1 W/m². This leads to mean stratospheric aerosol climate forcing −0.3 W/m² for remainder of the 21st century, similar to the mean stratospheric aerosol forcing for 1850-2015 (Table A10). Reconstruction of historical solar forcing (Coddington et al., 2015; Kopp et al., 2016), based on data in Fig. A11, is extended with an 11-year cycle.

Individual and net climate forcings for the several fossil fuel emission reduction rates are shown in Fig. 11a,c. Scenarios with linearly growing $CO_2$ extraction at rates required to yield 350 or 450 ppm airborne $CO_2$ in 2100 are in Fig. 11b,d. These forcings and the assumed climate response function define expected global temperature for the entire industrial era considered here (Fig. 12). We extended the global temperature calculations from 2100 to 2200 by continuing the %/year change of $CO_2$ emissions. In the cases with $CO_2$ extraction we kept the GHG climate forcing fixed in the 22nd century, which meant that large $CO_2$ extraction continued in cases with continuing high emissions, e.g., the case with constant emissions that required extraction of 695 PgC during 2020-2100 required further extraction of ~900 PgC during 2100-2200. Even the cases with annual emission reductions −6%/year and −3%/year required small extractions to compensate for back-flux of $CO_2$ from the ocean that accumulated there historically.

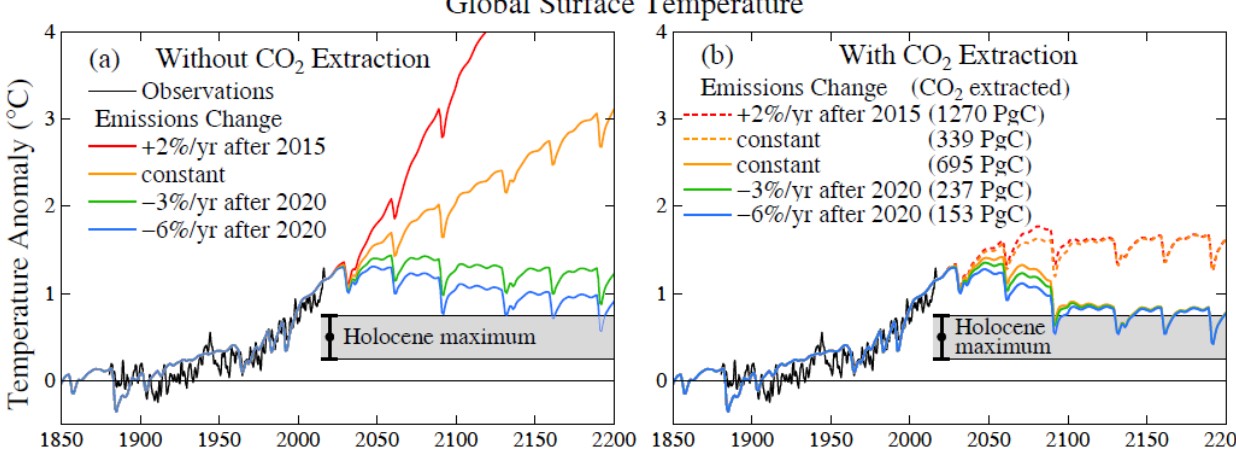

**Figure 12.** Simulated global temperature for Fig. 11 forcings. Observations as in Fig. 2. Temperature zero-point is the 1880-1920 mean temperature for both observations and model. Gray area is 2σ (95% confidence) range for centennially-smoothed Holocene maximum, but there is further uncertainty about the magnitude of the Holocene maximum, as noted in the text and discussed by Liu et al. (2014).

A stark summary of alternative futures emerges from Fig. 12a. If emissions grow 2%/ year, modestly slower than the 2.6%/year growth of 2000-2015, warming reaches ~4°C by 2100. Warming is about 2°C if emissions are constant until 2100. Furthermore, both scenarios launch Earth onto a course of more dramatic change well beyond the initial 2-3°C global warming, because: (1) warming continues beyond 2100 as the planet is still far from equilibrium with the climate forcing, and (2) warming of 2-3°C would unleash strong slow feedbacks, including melting of ice sheets and increases of GHGs.

The most important conclusion from Fig. 12a is the proximity of results for the cases with emission reductions of 6%/year and 3%/year. Although Hansen et al. (2013a) called for emission reduction of 6%/year to restore $CO_2$ to 350 ppm by 2100, that rate of reduction may have been regarded as implausibly steep by a federal court in 2012 when it declined to decide whether the U.S. was violating the public trust by causing or contributing to dangerous climate change (Alec L v. Jackson, 2012). Such a concern is less pressing for emission reductions of 3% per year. Note that reducing global emissions at a rate of 3%/year (or more steeply) maintains global warming at less than 1.5°C above preindustrial temperature.

However, end-of-century temperature still rises 0.5°C or more above the prior Holocene maximum with consequences for slow feedbacks that are difficult to foresee. Desire to minimize sea level rise spurs the need to get global temperature back into the Holocene range. That goal preferably should be achieved on the time scale of a century or less, because paleoclimate evidence indicates that the response time of sea level to climate change is 1-4 centuries (Grant et al., 2012, 2014) for natural climate change, and if anything the response should be faster to a stronger, more rapid human-made climate forcing. The scenarios that reduce $CO_2$ to 350 ppm succeed in getting temperature back close to the Holocene maximum by 2100 (Fig. 12b), but they require extractions of atmospheric $CO_2$ that range from 153 PgC in the scenario with 6%/year emission reductions to 1630 PgC in the scenario with +2%/year emission growth.

Scenarios ranging from constant emissions to +2%/year emissions growth can be made to yield 450 ppm in 2100 via extraction of 339-1270 PgC from the atmosphere (Fig. 10b). However, these scenarios still yield warming more than 1.5°C above the preindustrial level

(more than 1°C above the early Holocene maximum).  Consequences of such warming and the
plausibility of extracting such huge amounts of atmospheric $CO_2$ are considered below.

## 9  CO₂ Extraction: Estimated Cost and Alternatives

Extraction of $CO_2$ from the air, also called negative emissions or carbon dioxide removal (CDR),
is required if large, long-term excursion of global temperature above its Holocene range is to be
averted, as shown above.  In estimating the cost and plausibility of $CO_2$ extraction we distinguish
between (1) carbon extracted from the air by improved agricultural and forestry practices, and
(2) additional "technological extraction" by intensive negative emission technologies.

We assume that improved practices will aim at optimizing agricultural and forest carbon
uptake via relatively natural approaches, compatible with the land delivering a range of
ecosystem services (Smith 2016; Smith et al., 2016).  In contrast, proposed technological
extraction and storage of $CO_2$ generally does not have co-benefits and remains unproven at
relevant scales (NAS, 2015a).  Improved practices have local benefits in agricultural yields and
forest products and services (Smith et al., 2016), which may help minimize net costs. The
Intended Nationally Determined Contributions (INDCs) submitted by 189 countries include
carbon drawdown through land use plans (United Nations, 2016) with aggregate removal rate of
~2 $PgCO_2$/year (~0.55 PgC/year) after 2020.  These targets are not the maximum possible
drawdown, as they are only about a third of amounts Smith (2016) estimated as "realistic".

Developed countries recognize a financial obligation to less developed countries that have
done little to cause climate change (Paris Agreement 2015)[11].  We suggest that at least part of
developed country support should be channeled through agricultural and forestry programs, with
continual evaluation and adjustment to reward and encourage progress (Bustamante et al., 2014).
Efforts to minimize non-$CO_2$ GHGs can be included in the improved practices program.

Here, we do not estimate the cost of $CO_2$ extraction obtained via the "improved agricultural
and forestry practices[12]," because that would be difficult given the range of activities it is likely
to entail, and because it is not necessary for reaching the conclusion that total $CO_2$ extraction
costs will be high due to the remaining requirements for technological extraction.  However, we
do estimate the potential magnitude of $CO_2$ extraction that might be achievable via such
improved practices, as that is needed to quantify the required amount of "technological
extraction" of $CO_2$.  Finally, we compare costs of extraction with estimated costs of mitigation
measures that could limit the magnitude of required extraction, while admitting that there is large
uncertainty in both extraction and mitigation cost estimates.

---

[11] Another conceivable source of financial support for $CO_2$ drawdown might be legal settlements with fossil fuel
companies, analogous to penalties that courts have imposed on tobacco companies, but with the funds directed to
the international "improved practices" programs.

[12] A comment is in order about the relation of "improved agricultural and forestry practices" with an increased role
of biofuels in climate mitigation.  Agriculture, forestry and other land use has potential for important contributions
to climate change mitigation (Smith et al., 2014).  However, first-generation biofuel production and use (which is
usually based on edible portions of feedstocks, such as starch) is not inherently carbon neutral, indeed it is likely
carbon-positive, as has been illustrated in specific quantitative analyses for corn ethanol in the United States
(Searchinger et al., 2008; DeCicco et al., 2016).  The need for caution regarding the role of biofuels in climate
mitigation is discussed by Smith et al. (2014).

## 9.1 Estimated Cost of CO$_2$ Extraction

Hansen et al. (2013) suggested a goal of 100 PgC extraction in the 21$^{st}$ century, which would be almost as large as estimated net emissions from historic deforestation and land use (Ciais et al., 2013). Hansen et al. (2013a) assumed that 100 PgC was about as much as could be achieved via relatively natural reforestation and afforestation (Canadell and Raupach, 2008) and improved agricultural practices that increase soil carbon (Smith, 2016).

Here we first reexamine whether a concerted global effort on carbon storage in forests and soil might have potential to provide a carbon sink substantially larger than 100 PgC this century. Smith et al. (2016) estimate that reforestation and afforestation together have carbon storage potential of about 1.1 PgC/year. However, as forests mature, their uptake of atmospheric carbon decreases (termed "sink saturation"), thereby limiting CO$_2$ drawdown. Taking 50 years as the

average time for tropical, temperate and boreal trees to experience sink saturation yields 55 PgC as the potential storage in forests this century.

Smith (2016) shows that soil carbon sequestration and soil amendment with biochar compare favorably with other negative emission technologies with less impact on land use, water use, nutrients, surface albedo, and energy requirements, but understanding of and literature on

biochar are limited (NAS, 2015a). Smith estimates that soil carbon sequestration has potential to store 0.7 PgC/year. However, as with carbon storage in forest, there is a saturation effect. A commonly used 20-year saturation time (IPCC, 2006) would yield 14 PgC soil carbon storage, while an optimistic 50-year saturation time would yield 35 PgC. Use of biochar to improve soil fertility provides additional carbon storage of up to 0.7-1.8 PgC/year (Woolf et al., 2010; Smith

2016). Larger industrial-scale biochar carbon storage is conceivable, but belongs in the category of intensive negative emission technologies, discussed below, whose environmental impacts and costs require scrutiny. We conclude that 100 PgC is an appropriate ambitious estimate for potential carbon extraction via a concerted global-scale effort to improve agricultural and forestry practices with carbon drawdown as a prime objective.

Intensive negative emission technologies that could yield greater CO$_2$ extraction include (1) burning of biofuels, most commonly at power plants, with capture and sequestration of resulting CO$_2$ (Creutzig et al., 2015), and (2) direct air capture of CO$_2$ and sequestration (Keith, 2009; NAS, 2015a), and (3) grinding and spreading of minerals such as olivine to enhance geological weathering (Taylor et al., 2016). However, energy, land and water requirements of these

technologies impose economic and biophysical limits on CO$_2$ extraction (Smith et al., 2016).

The popular concept of bioenergy with carbon capture and storage (BECCS) requires large areas, high fertilizer and water use, and may compete with other vital land use such as agriculture (Smith, 2016). Costs estimates are ~\$150-350/tC for crop-based BECCS (Smith et al., 2016).

Direct air capture has more limited area and water needs than BECCS and no fertilizer

requirement, but it has high energy use, has not been demonstrated at scale, and cost estimates exceed those of BECCS (Socolow et al., 2011; Smith et al., 2016). Keith et al. (2006) have argued that, with strong research and development support and industrial-scale pilot projects sustained over decades, it may be possible to achieve costs ~\$200/tC, thus comparable to BECCS costs; however other assessments are higher, reaching \$1400-3700/tC (NAS, 2015a).

Enhanced weathering via soil amendment with crushed silicate rock is a candidate negative emission technology that also limits coastal ocean acidification as chemical products liberated by weathering increase land-ocean alkalinity flux (Kohler et al., 2010; Taylor et al., 2016). If two-thirds of global croplands were amended with basalt dust, as much as 1-3 PgC/year might be extracted, depending on application rate (Taylor et al., 2016), but energy costs of mining,

grinding and spreading likely reduce this by 10-25% (Moosdorf et al., 2014).  Such large-scale enhanced weathering is speculative, but potential co-benefits for temperate and tropical agroecosystems could affect its practicality, and may put some enhanced weathering into the category of improved agricultural and forestry practices.  Benefits include crop fertilization that increases yield and reduces use and cost of other fertilizers, increasing crop protection from
insect herbivores and pathogens thus decreasing pesticide use and cost, neutralizing soil acidification to improve yield, and suppression of GHG ($N_2O$ and $CO_2$) emissions from soils (Edwards et al., 2017; Kantola et al., 2017). Against these benefits, we note potential negative impacts of air and water pollution caused by the mining, including downstream environmental consequences if silicates are washed into rivers and the ocean, causing increased turbidity,
sedimentation, and pH, with unknown impacts on biodiversity (Edwards et al., 2017).  Cost of enhanced weathering might be reduced by deployment with reforestation and afforestation and with crops used for BECCS; this could significantly enhance the combined carbon sequestration potential of these methods.

For cost estimates, we first consider restoration of airborne $CO_2$ to 350 ppm in 2100 (Fig.
10b), which would keep global warming below 1.5°C and bring global temperature back close to the Holocene maximum by end-of-century (Fig. 12b).  This scenario keeps the temperature excursion above the Holocene level small enough and brief enough that it has the best chance of avoiding ice sheet instabilities and multi-meter sea level rise (Hansen et al., 2016).  If fossil fuel emission phasedown of 6%/year had begun in 2013, as proposed by Hansen et al. (2013a), this
scenario would have been achieved via the hypothesized 100 PgC carbon extraction from improved agricultural and forestry practices.

We examine here scenarios with 6%/year and 3%/year emission reduction starting in 2021, as well as scenarios with constant emissions and +2%/year emission growth starting in 2016 (Figs. 10b and 12b).  The −6%/year and −3%/year scenarios leave a requirement to extract 153
and 237 PgC from the air during this century.  Constant emission and +2%/year emission scenarios yield extraction requirements of 695 and 1630 PgC to reach 350 ppm $CO_2$ in 2100.

Total $CO_2$ extraction requirements for these scenarios are given in Fig. 10.  Cost estimates here for extraction use amounts 100 Pg less than in Fig. 10 under assumption that 100 PgC can be stored via improved agricultural and forestry practices.  Shortfall of this 100 PgC goal will
increase our estimated costs accordingly, as will the cost of the improved agricultural and forestry program.

Given a $CO_2$ extraction cost of $150-350/tC for intensive negative emission technologies (Fig. 3f of Smith et al., 2016), the 53 PgC additional extraction required for the scenario with 6%/year emission reduction would cost $8-18.5 trillion, thus $100-230 billion per year if spread
uniformly over 80 years.  We cannot rule out possible future reduction in $CO_2$ extraction costs, but given the energy requirements for removal and the already optimistic lower limit on our estimate, we do not speculate further about potential cost reduction.

In contrast, continued high emissions, between constant emissions and +2%/year, would require additional extraction of 595-1530 PgC (Fig. 10b) at a cost $89-535 trillion or $1.1-6.7
trillion per year over 80 years.[13]  Such extraordinary cost, along with the land area, fertilizer and water requirements (Smith et al., 2016) suggest that, rather than the world being able to buy its way out of climate change, continued high emissions would likely force humanity to live with climate change running out of control with all the consequences that would entail.

---

[13] For reference, the United Nations global peacekeeping budget is about $10B/year. National military budgets are larger: the 2015 USA military budget was $596B and the global military budget was $1.77 trillion (SIPRI 2016).

### 9.2 Mitigation Alternative

High costs of $CO_2$ extraction raise the question of how these costs compare to the alternative: taking actions to mitigate climate change by reducing fossil fuel $CO_2$ emissions. The Stern Review (Stern, 2006; Stern and Taylor, 2007) used expert opinion to produce an estimate for the cost of reducing emissions to limit global warming to about 2°C. Their central estimate was 1% of gross domestic product (GDP) per year, thus about $800 billion per year. They argued that

this cost was much less than likely costs of future climate damage if high emissions continue, unless we apply a high "discount rate" to future damage, which has ethical implications in its treatment of today's young people and future generations. However, their estimated uncertainty of the cost is ±3%, i.e., the uncertainty is so large as to encompass GDP gain.

Hsu (2011) and Ackerman and Stanton (2012) argue that economies are more efficient if the

price of fossil fuels better reflects costs to society, and thus GDP gain is likely with an increasing carbon price. Mankiw (2009) similarly suggests that a revenue-neutral carbon tax is economically beneficial. Hansen (2009, 2014) advocates an approach in which a gradually rising carbon fee is collected from the fossil fuel industry with the funds distributed uniformly to citizens. This approach provides incentives to business and the public that drive the economy

toward energy efficiency, conservation, renewable energies and nuclear power. An economic study of this carbon-fee-and-dividend in the United States (Nystrom and Luckow, 2014) supports the conclusion that GDP increases as the fee rises steadily. These studies refute the common argument that environmental protection is damaging to economic prosperity.

We can also compare $CO_2$ extraction cost with the cost of carbon-free energy infrastructure.

Global energy consumption in 2015 was 12.9 Gtoe,[14] with coal providing 30% of global energy and almost 45% of global fossil fuel $CO_2$ emissions (BP, 2016). Most coal use, and its increases, are in Asia, especially China and India. Carbon-free replacement for coal energy is expected to be some combination of renewables (including hydropower) and nuclear power. China is leading the world in installation of wind, solar and nuclear power, with new nuclear power in

2015 approximately matching the sum of new solar and wind power (BP, 2016). For future decarbonization of electricity it is easiest to estimate the cost of the nuclear power component, because nuclear power can replace coal for baseload electricity without the need for energy storage or major change to national electric grids. Recent costs of Chinese and South Korean light water reactors are in the range $2000-3000 per kilowatt (Chinese Academy of Engineering,

2015; Lovering et al., 2016). Although in some countries reactor costs stabilized or declined with repeated construction of the same reactor design, in others costs have risen for a variety of reasons (Lovering et al., 2016). Using $2500 per kilowatt as reactor cost and assuming 85% capacity factor (percent uptime for reactors) yields a cost of $10 trillion to produce 20% of present global energy use (12.9 Gtoe). Note that 20% of current global energy use is a huge

amount (Fig. 13), exceeding the sum of present hydropower (6.8%), nuclear (4.4%), wind (1.4%), solar (0.4%), and other renewable energies (0.9%).

We do not suggest that new nuclear power plants on this scale will or necessarily should be built. Rather we use this calculation to show that mitigation costs are not large in comparison to costs of extracting $CO_2$ from the air. Renewable energy costs have fallen rapidly in the past 2-3

decades with the help of government subsidies, especially renewable portfolio standards that

---

[14] Gtoe is gigatons oil equivalent. 1 Gtoe is 41.868 EJ (Exajoule = $10^{18}$ Joules) or 11,630 TWh (terawatt hours).

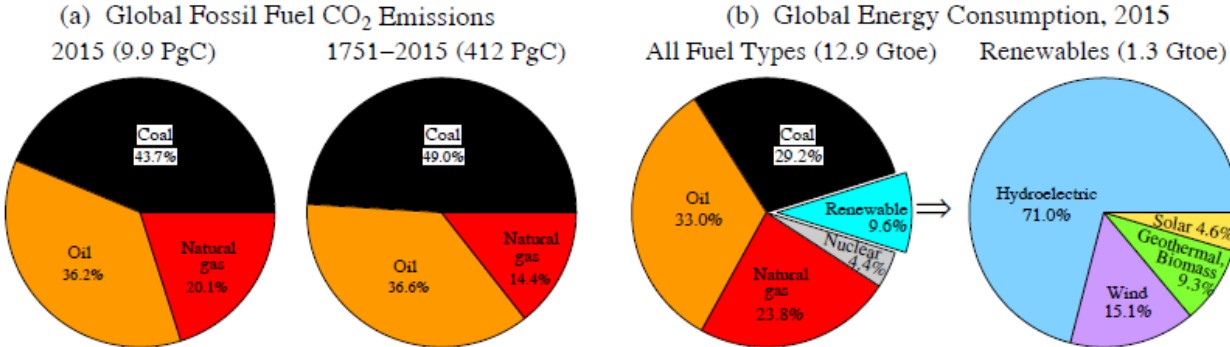

**Figure 13. (a)** Global fossil fuel emissions data from Boden et al. (2017) for 1751-2014 are extended to 2015 using BP (2016) data. **(b)** Global primary energy consumption data from BP (2016); energy accounting method is the substitution method (Macknick, 2011).

require utilities to achieve a specified fraction of their power from renewable sources. Yet fossil fuel use continues to be high, at least in part because fossil fuel prices do not include their full cost to society. Rapid and economic movement to non-fossil energies would be aided by a rising carbon price, with the composition of energy sources determined by competition among all non-fossil energy sources, as well as energy efficiency and conservation. Sweden provides a prime example: it cut per capita emissions by two-thirds since the 1990s while doubling per capita income in a capitalistic framework that embodies free-market principles (Pierrehumbert, 2016).

Mitigation of climate change deserves urgent priority. We disagree with assessments such as "The world will probably have only two choices if it wants to stay below 1.5°C of warming. It must either deploy carbon dioxide removal on an enormous scale or use solar geoengineering" (Parker and Geden, 2016). While we reject 1.5°C as a safe target – it is likely warmer than the Eemian and far above the Holocene range – Figure 12 shows that fossil fuel emission reduction of 3%/year beginning in 2021 yields maximum global warming ~1.5°C for climate sensitivity 3°C for 2×$CO_2$, with neither $CO_2$ removal nor geoengineering. These calculations show that mitigation – reduction of fossil fuel emissions – is very effective. We know no persuasive scientific reason to a priori reject as implausible a rapid phasedown of fossil fuel emissions.

## 10 Non-$CO_2$ GHGs, Aerosols and Purposeful Climate Intervention

### 10.1 Non-$CO_2$ GHGs

The annual increment in GHG climate forcing is growing, not declining. The increase is more than 20% in just the past five years (Fig. 8). Resurgence of $CH_4$ growth is partly responsible, but $CO_2$ is by far the largest contributor to growth of GHG climate forcing (Fig. 8). Nevertheless, given the difficulty and cost of reducing $CO_2$, we must ask about the potential for reducing non-$CO_2$ GHGs. Could realistic reductions of these other gases substantially alter the $CO_2$ abundance required to meet a target climate forcing?

We conclude, as discussed in Appendix A13, that a net decrease of climate forcing by non-$CO_2$ GHGs of perhaps −0.25 W/m$^2$ relative to today is plausible, but we must note that this is a dramatic change from the growing abundances, indeed accelerating growth, of these gases today. Achievement of this suggested negative forcing requires (i) successful completion of planned phase-out of MPTGs (−0.23 W/m$^2$), (ii) absolute reductions of $CH_4$ forcing by 0.12 W/m$^2$ from

its present value, and (iii) $N_2O$ forcing increasing by only 0.1 $W/m^2$. Achieving this net negative forcing of $-0.25$ $W/m^2$ for non-$CO_2$ gases would allow $CO_2$ to be 365 ppm, rather than 350 ppm, while yielding the same total GHG forcing. Absolute reduction of non-$CO_2$ gases is thus helpful,
but does not alter the requirement for rapid fossil fuel emission reductions. Moreover, this is an optimistic scenario that is unlikely to occur in the absence of a reduction of $CO_2$, which is needed to limit global warming and thus avoid amplifying GHG feedbacks.

## 10.2 Aerosols and Purposeful Climate Intervention

Human-made aerosols today are believed to cause a large, albeit poorly measured, negative climate forcing (Fig. 4) of the order of $-1$ $W/m^2$ with uncertainty of at least 0.5 $W/m^2$ (Figure 7.19 Boucher et al., 2013). Fossil fuel burning is only one of several human-caused aerosol sources (Boucher et al., 2013). Given that human population continues to grow, and that human-caused climate effects such as increased desertification can lead to increased aerosols, we do not
anticipate a large reduction in the aerosol cooling effect, even if fossil fuel use declines. Rao et al. (2017) suggest that future aerosol amount will decline due to technological advances and global action to control emissions. We are not confident of such a decline, as past controls have been at least matched by increasing emissions in developing regions, and global population continues to grow. However, to the extent that Rao et al. (2017) projections are borne out, they
will only strengthen the conclusions of our present paper about the threat of climate change for young people and the burden of decreasing GHG amounts in the atmosphere.

    Recognition that aerosols have a cooling effect, combined with the difficulty of restoring $CO_2$ to 350 ppm or less, inevitably raises the issue of purposeful climate intervention, also called geoengineering, and specifically solar radiation management (SRM). The cooling mechanism
receiving greatest attention is injection of $SO_2$ into the stratosphere (Budyko, 1974; Crutzen, 2006), thus creating sulfuric acid aerosols that mimic the effect of volcanic aerosol cooling. That idea and others are discussed in a report of the U.S. National Academy of Sciences (NAS, 2015b) and references therein. We limit our discussion to the following summary comments.

    Such purposeful intervention in nature, an attempt to mitigate effects of one human-made
pollutant with another, raises additional practical and ethical issues. Stratospheric aerosols, e.g., could deplete stratospheric ozone and/or modify climate and precipitation patterns in ways that are difficult to predict with confidence, while doing nothing to alleviate ocean acidification caused by rising $CO_2$; we note that Keith et al. (2016) suggest alternative aerosols that would limit the impact on ozone. However, climate intervention also raises issues of global governance,
and introduces the possibility of sudden global consequences if aerosol injection is interrupted (Boucher at al., 2013). Despite these issues, it is apparent that cooling by aerosols, or other methods that alter the amount of sunlight absorbed by Earth, could be effective more quickly than the difficult process of removing $CO_2$ from the air. Thus we agree with the NAS (2015b) conclusion that research is warranted to better define the climate, economic, political, ethical,
legal and other dimensions of potential climatic interventions.

    In summary, although research on climate interventions is warranted, the possibility of geoengineering can hardly be seen as alleviating the overall burden being placed on young people by continued high fossil fuel emissions. We concur with the assessment (NAS, 2015b) that such climate interventions are no substitute for the reduction of carbon dioxide emissions
needed to stabilize climate and avoid deleterious consequences of rapid climate change.

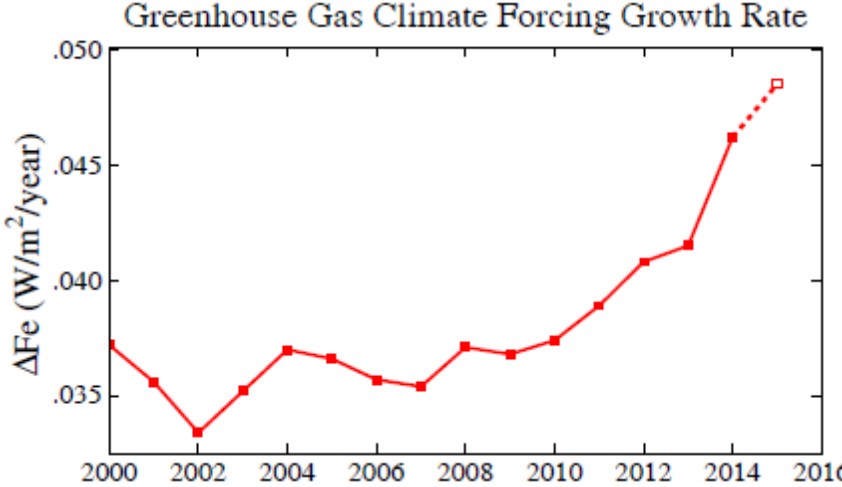

**Figure 14.** Recent growth rate of total GHG effective climate forcing; points are 5-year running means, except 2015 point is a 3-year mean. See Fig. 8 for individual gases.

## 11. Discussion

Global temperature is now far above its range during the preindustrial Holocene, attaining at least the warmth of the Eemian period, when sea level reached +6-9 m relative to today. Also Earth is now out of energy balance, implying that more warming will occur, even if atmospheric GHG amounts are stabilized at today's level. Furthermore, the GHG climate forcing is not only still growing, the growth rate is actually accelerating, as shown in Fig. 14, which is extracted
from data in our Fig. 8.

This summary, based on real world data for temperature, planetary energy balance, and GHG changes, differs from a common optimistic perception of progress toward stabilizing climate. That optimism may be based on the lowered warming target in the Paris Agreement (2015), slowdown in the growth of global fossil fuel emissions in the past few years (Fig. A1),
and falling prices of renewable energies, but the hard reality of the climate physics emerges in Figs. 2, 5, 8 and 14. Although the scenarios employed in climate simulations for the most recent IPCC study (AR5) include cases with rapidly declining GHG growth, the scenarios do nothing to alter reality, which reveals that GHG growth rates not only remain high, they are accelerating.

The need for prompt action implied by these realities may not be a surprise to the relevant
scientific community, because paleoclimate data revealed high climate sensitivity and the dominance of amplifying feedbacks. However, effective communication with the public of the urgency to stem human-caused climate change is hampered by the inertia of the climate system, especially the ocean and the ice sheets, which respond rather slowly to climate forcings, thus allowing future consequences to build up before broad public concern awakens. Some effects of
human-caused global warming are now unavoidable, but is it inevitable that sea level rise of many meters is locked in, and, if so, on what time scale? Precise unequivocal answers to such questions are not possible. However, useful statements can be made.

First, the inertia and slow response of the climate system also allow the possibility of actions to limit the climate response by reducing human-caused climate forcing in coming years and
decades. Second, the response time itself depends on how strongly the system is being forced; specifically, the response might be much delayed with a weaker forcing.

For example, studies suggesting multi-meter sea level rise in a century assume continued high fossil fuel emissions this century (Hansen et al., 2016) or at least a 2°C SST increase (DeConto and Pollard, 2016). Ice sheet response time decreases rapidly in models as the forcing increases, because processes such as hydrofracturing and collapse of marine-terminating ice cliffs spur ice sheet disintegration (Pollard et al., 2015). All amplifying feedbacks, including atmospheric water vapor, sea ice cover, soil carbon release and ice sheet melt could be reduced by rapid emissions phasedown. This would reduce the risk of climate change running out of humanity's control and provide time to assess the climate response, develop relevant technologies, and consider further purposeful actions to limit and/or adapt to climate change.

Concern exists that large sea level rise may be inevitable, because of numerous ice streams on Antarctica and Greenland with inward-sloping beds (beds that deepen upstream) subject to runaway marine ice sheet instability (Mercer, 1978; Schoof, 2007, 2010). Some ice stream instabilities may already have been triggered (Rignot et al., 2014), but the number of ice streams affected and the time scale of their response may differ strongly depending on the magnitude of the forcing (DeConto and Pollard, 2016). Sea level rise this century of say half a meter to a meter, which may be inevitable even if emissions decline, would have dire consequences, yet these are dwarfed by the humanitarian and economic disasters that would accompany sea level rise of several meters (McGranahan et al., 2007). Given the increasing proportion of global population living in coastal areas (Hallegatte et al., 2013), there is potential for forced migrations of hundreds of millions of people, dwarfing prior refugee humanitarian crises, challenging global governance (Biermann and Boas, 2010) and security (Gemenne et al., 2014).

Global temperature is a useful metric, because increasing temperature drives amplifying feedbacks. Global ocean temperature is a major factor affecting ice sheet size, as indicated by both model studies (Pollard et al., 2015) and paleoclimate analyses (Overpeck et al., 2006; Hansen et al., 2016). Eemian ocean warmth, probably not more than about +0.7°C warmer than pre-industrial conditions (McKay et al., 2011; Masson-Delmotte et al., 2013; Section 2.2 above), corresponding to global warmth about +1°C relative to preindustrial, led to sea level 6-9 m higher than today. This implies that, on the long run, the El Niño-elevated 2016 temperature of +1.3°C relative to preindustrial temperature, and even the (+1.05°C) underlying trend to date without the El Niño boost, are probably too high for maintaining our present coastlines.

We conclude that the world has already overshot appropriate targets for GHG amount and global temperature, and we thus infer an urgent need for both (1) rapid phasedown of fossil fuel emissions, (2) actions that draw down atmospheric $CO_2$, and (3) actions that, at minimum, eliminate net growth of non-$CO_2$ climate forcings. These tasks are formidable and, with the exception of the Montreal Protocol agreement on HFCs that will halt the growth of their climate forcing (Appendix A13), they are not being pursued globally. Actions at citizen, city, state and national levels to reduce GHG emissions provide valuable experience and spur technical developments, but without effective global policies the impact of these local efforts is reduced by the negative feedback caused by reduced demand for and price of fossil fuels.

Our conclusion that the world has overshot appropriate targets is sufficiently grim to compel us to point out that pathways to rapid emission reductions are feasible. Peters et al. (2013) note that Belgium, France and Sweden achieved emission reductions of 4-5%/year sustained over 10 or more years in response to the oil crisis of 1973. These rates were primarily a result of nuclear power build programs, which historically has been the fastest route to carbon-free energy (Fig. 2 of Cao et al., 2016). These examples are an imperfect analogue, as they were driven by a desire

for energy independence from oil, but present incentives are even more comprehensive. Peters et al. (2013) also note that a continuous shift from coal to natural gas led to sustained reductions of 1-2%/year in the UK in the 1970s and in the 2000s, 2%/year in Denmark in 1990-2000s, and 1.4%/year in the USA since 2005. Furthermore, these examples were not aided by the economy-wide effect of a rising carbon fee or tax (Hsu, 2011; Ackerman and Stanton, 2012; Hansen, 2014), which encourages energy efficiency and carbon-free energies.

In addition to $CO_2$ emission phase-out, large $CO_2$ extraction from the air is needed and a halt of growth of non-$CO_2$ climate forcings to achieve the temperature stabilization of our scenarios. Success of both $CO_2$ extraction and non-$CO_2$ GHG controls requires a major role for developing countries, given that they have been a large source of recent deforestation (IPCC, 2013) and have a large potential for reduced emissions. Ancillary benefits of the agricultural and forestry practices needed to achieve $CO_2$ drawdown, such as improved soil fertility, advanced agricultural practices, forest products, and species preservation, are of interest to all nations. Developed nations have a recognized obligation to assist nations that have done little to cause climate change yet suffer some of the largest climate impacts. If economic assistance is made partially dependent on verifiable success in carbon drawdown and non-$CO_2$ mitigation, this will provide incentives that maximize success in carbon storage. Some activities, such as soil amendments that enhance weathering, might be designed to support both $CO_2$ and other GHG drawdown.

Considering our conclusion that the world has overshot the appropriate target for global temperature, and the difficulty and perhaps implausibility of negative emissions scenarios, we would be remiss if we did not point out the potential contribution of demand-side mitigation that can be achieved by individual actions as well as by government policies. Numerous studies (e.g. Hedenhus et al., 2014; Popp et al., 2010) have shown that reduced ruminant meat and dairy products is needed to reduce GHG emissions from agriculture, even if technological improvements increase food yields per unit farmland. Such climate-beneficial dietary shifts have also been linked to co-benefits that include improved sustainability and public health (Bajzelj et al., 2014; Tilman and Clark, 2014). Similarly, Working Group 3 of IPCC (2014) finds "robust evidence and high agreement" that demand-side measures in the agriculture and land use sectors, especially dietary shifts, reduced food waste, and changes in wood use have substantial mitigation potential, but they remain under-researched and poorly quantified.

There is no time to delay. $CO_2$ extraction required to achieve 350 ppm $CO_2$ in 2100 was ~100 PgC if 6%/year emission reductions began in 2013 (Hansen et al., 2013a). Required extraction is at least ~150 PgC in our updated scenarios, which incorporate growth of emissions in the past four years and assume that emissions will continue at approximately current levels until a global program of emission reductions begins in four years (in 2021 relative to 2020; see Figs. 9, 10 for reduction rates). The difficulty of stabilizing climate was thus markedly increased by a delay in emission reductions of eight years, from 2013 to 2021. Nevertheless, if rapid emission reductions are initiated soon, it is still possible that at least a large fraction of required $CO_2$ extraction can be achieved via relatively natural agricultural and forestry practices with other benefits. On the other hand, if large fossil fuel emissions are allowed to continue, the scale and cost of industrial $CO_2$ extraction, occurring in conjunction with a deteriorating climate and costly dislocations, may become unmanageable. Simply put, the burden placed on young people and future generations may become too heavy to bear.

# Appendix A: Additional figures, tables and explanatory information

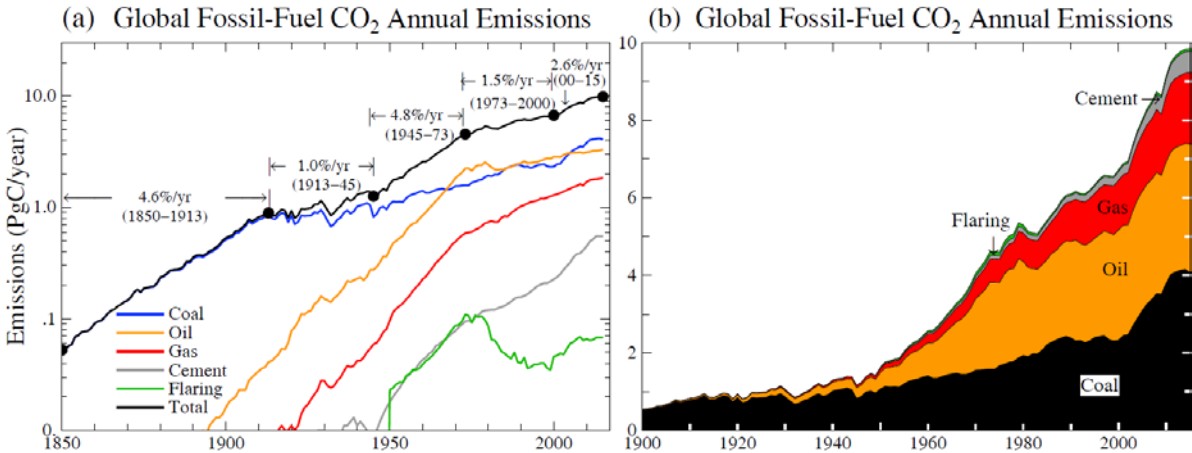

**Figure A1.** $CO_2$ emissions from fossil fuels and cement use based on Boden et al. (2017) through 2014, extended using BP (2016) energy consumption data. (a) is log scale and (b) is linear. Growth rates r in (a) for an n-year interval are from $(1+r)^n$ with end values being 3-year means to minimize noise.

## A1. Fossil Fuel $CO_2$ Emissions

$CO_2$ emissions from fossil fuels in 2015 were only slightly higher than in 2014 (Fig. A1). Such slowdowns are common, usually reflecting the global economy. Given rising global population and the fact that nations such as India are still at early stages of development, the potential exists for continued emissions growth. Fundamental changes in energy technology are needed for the world to rapidly phase down fossil fuel emissions.

Emissions are growing rapidly in emerging economies; while growth slowed in China in the past two years, emissions remain high (Fig. 1). The Kyoto Protocol (1997), a policy instrument of the Framework Convention (UNFCCC, 1992), spurred emission reductions in some nations, and the collapse of the Soviet Union caused a large decrease of emissions by Russia (Fig. 1b). However, growth of international ship and air emissions (Fig. 1b) largely offset these reductions and the growth rate of global emissions actually accelerated from 1.5%/year in 1973-2000 to ~2.5%/year after 2000 (Fig. A1). China is now the largest source of fossil fuel emissions, followed by the U.S. and India, but on a per capita historical basis the U.S. is 10 times more accountable than China and 25 times more accountable than India for the increase of atmospheric $CO_2$ above its preindustrial level (Hansen and Sato, 2016). Tabular data for Figs. 1 and A1 are available on the web page www.columbia.edu/~mhs119/Burden.

## A2. Transient Climate Response to cumulative $CO_2$ Emissions (TCRE)

The transient climate response (TCR), defined as the global warming at year 70 in response to a 1%/year $CO_2$ increase, for our simple Green's function climate model is 1.89°C with energy imbalance of 1.52 $W/m^2$ at that point; this TCR is in the middle of the range reported in the IPCC AR5 report (IPCC, 2013). We calculate the transient climate response to cumulative carbon emissions (TCRE) of our climate plus carbon cycle model as in Section 10.8.4 of IPCC (2013), i.e., TCRE = TCR × CAF/$C_0$, where $C_0$ = preindustrial atmospheric $CO_2$ mass = 590 PgC and CAF = Catm/Csum, Catm = atmospheric $CO_2$ mass minus $C_0$ and Csum = cumulative $CO_2$ emissions (all evaluated at year 2100).

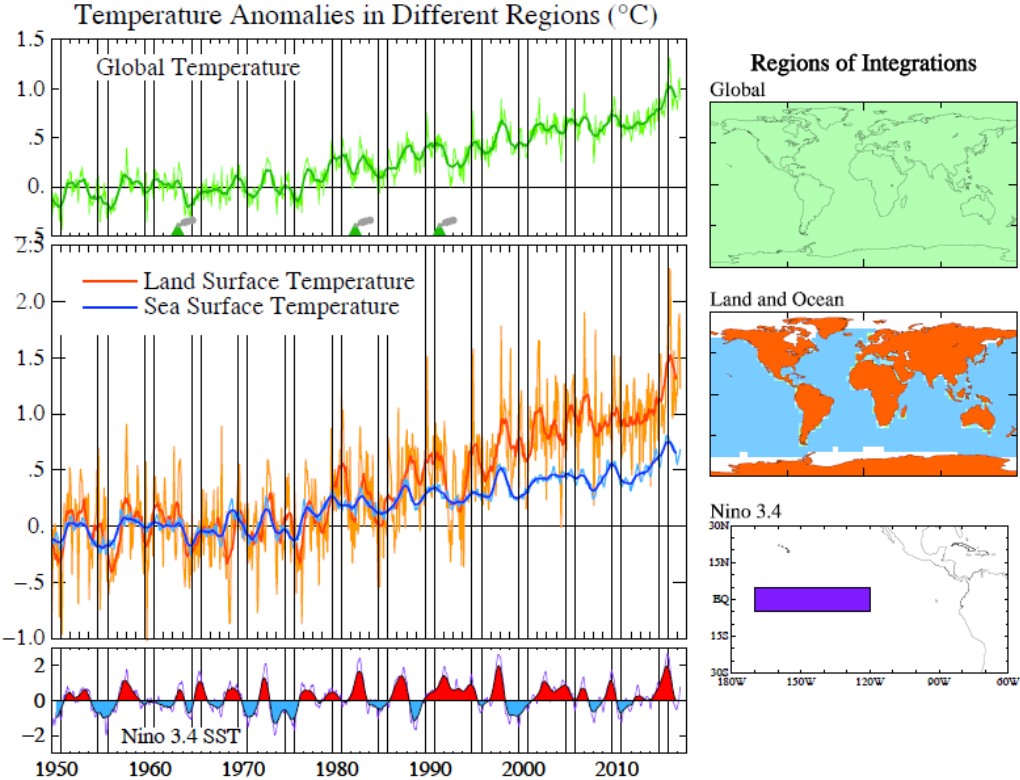

**Figure A3a.** Monthly (thin lines) and 12-month running mean (thick lines or filled colors for Niño 3.4) global, global land, sea surface, and Niño 3.4 temperatures. Temperatures are relative to 1951-1980 base period for the current GISTEMP analysis, which uses NOAA ERSST.v4 for sea surface temperature.

We find TCRE = 1.54°C per 1000 PgC at 2100 with constant emissions (which yields cumulative emissions of 1180 PgC at 2100, which is near the midpoint of the range assessed by IPCC, i.e., 0.8°C to 2.5°C per 1000 PgC (IPCC, 2013). Our two cases with rapidly declining emissions never achieve 1000 PgC emissions, but TCRE can still be computed using the IPCC formulae, yielding TCRE = 1.31 and 1.25°C per 1000 PgC at 2100 for the cases of −3%/year and −6%/year respective emission reductions. As expected, the rapid emission reductions substantially reduce the temperature rise in 2100.

### A3. Observed Temperature Data and Analysis Method

We use the current Goddard Institute for Space Studies global temperature analysis (GISTEMP), described by Hansen et al. (2010). The analysis combines data from: (1) meteorological station data of the Global Historical Climatology Network (GHCN) described by Peterson and Vose (1997) and Menne et al. (2012), (2) Antarctic research station data reported by the Scientific Committee on Antarctic Research (SCAR), (http://www.antarctica.ac.uk/met/READER), and (3) ocean surface temperature measurements from the NOAA Extended Reconstructed Sea Surface temperature (ERSST) (Smith et al., 2008; Huang et al., 2015).

Surface air temperature change over land is about twice SST change (Fig. A3a), and thus global temperature change is 1.3 times larger than the SST change. Note that the Arctic Ocean and parts of the Southern Ocean are excluded in the calculations because of inadequate data, but these regions are also not sampled in most paleo analyses and the excluded areas are small. Land area included covers 29% of the globe and ocean area included covers 65% of the globe.

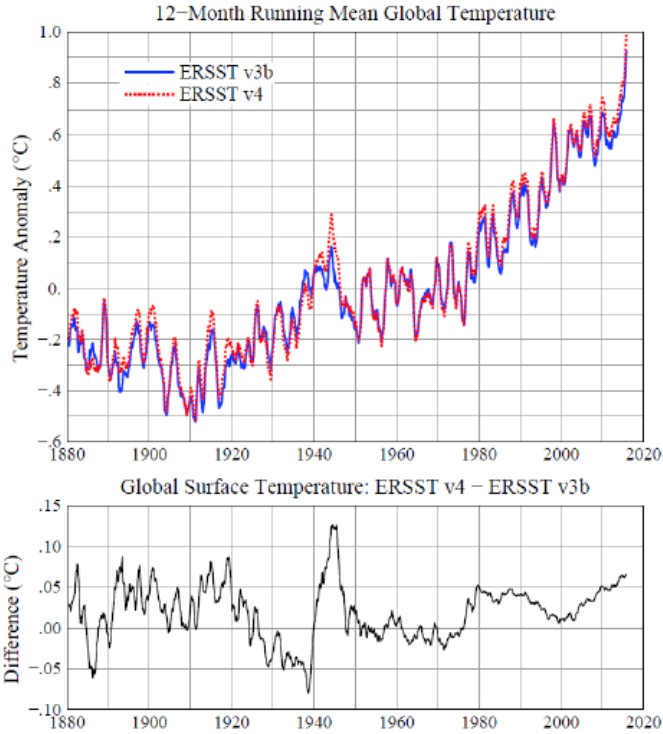

**Figure A3b.** Global surface temperature relative to 1951-1980 in the GISTEMP analysis, comparing the current analysis using NOAA ERSST.v4 for sea surface temperature with results using ERSST.v3b.

The present analysis uses GHCN.v3.3.0 (Menne et al., 2012) for land data and ERSST.v4 for sea surface temperature (Huang et al., 2015). Update from GHCN.v2 used in our 2010 analysis to GHCN.v3 had negligible effect on global temperature change over the past century (see graph on http://www.columbia.edu/~mhs119/Temperature/GHCN_V3vsV2/). However, the adjustments to SST to produce ERSST.v4 have a noticeable effect, especially in the period 1939-1945, as shown by the difference between the two data sets (lower graph in Fig. A3b). This change is of interest mainly because it increases the magnitude of an already unusual global temperature fluctuation in the 1940s, making the 1939-1945 global temperature maximum even more pronounced than it was in ERSST.v3 data. Thompson et al. (2008) show that two natural sources of variability, the El Niño/Southern Oscillation and (possibly related) unusual winter Arctic warmth associated with advection over high Northern Hemisphere latitudes, partly account for global warmth of 1939-1945, and they suggest that the sharp cooling after 1945 is a data flaw, due to a rapid change in the mix of data sources (bucket measurements and engine room intake measurements) and a bias between these that is not fully accounted for.

Huang et al. (2015) justify the changes made to obtain version 4 of ERSST, the changes including more complete input data in ICOADS Release 2.5, buoy SST bias adjustments not present in version 3, updated ship SST bias adjustments using Hadley Nighttime Marine Air Temperature version 2 (HadNMat2), and revised low-frequency data filling in data sparse regions using nearby observations. ERSST.v4 is surely an improvement in the record during the past half century when spatial and temporal data coverages are best. On the other hand, the largest changes between v3 and v4 are in 1939-1945, coinciding with World War II and changes in the mix of data sources. Several hot spots appear in the Southern Hemisphere ocean during WWII in the v4 data, and then disappear after the war (Fig. A3c). These hot spots coincide with the locations of large SST changes between v3 and v4 (Fig. A3c), which leads us to suspect that

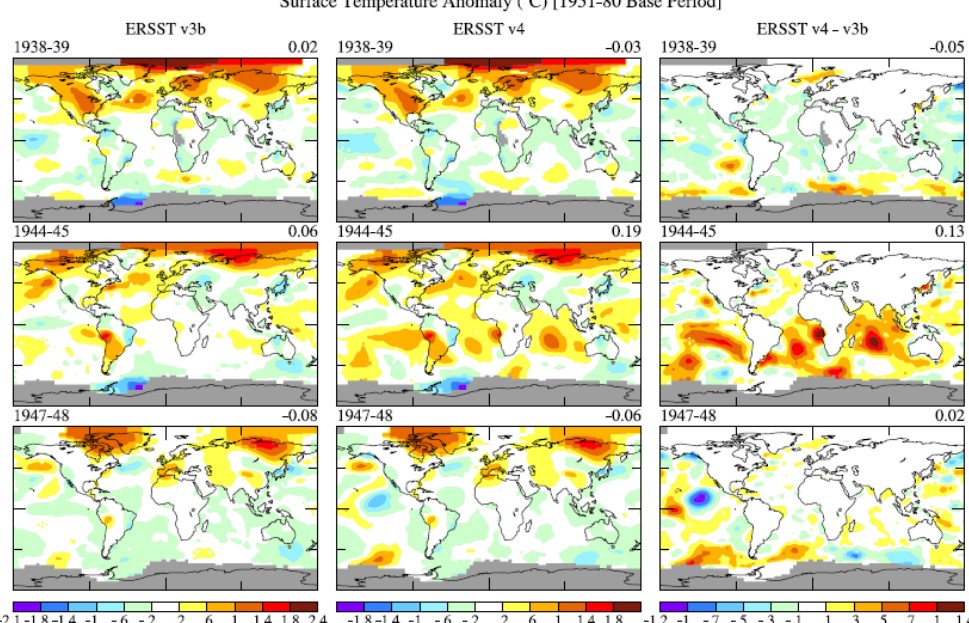

**Figure A3c.** Temperature anomalies in three periods relative to 1951-1980 comparing results obtained using ERSST.v3b (left column), ERSST.v4 (center column), and their difference (right column).

the magnitude of the 1940s global warming maximum (Fig. 2) is exaggerated; i.e., it is partly spurious. We suggest that this warming spike warrants scrutiny in the next version of the SST analysis. However, the important point is that these data adjustments and uncertainties are small in comparison with the long-term warming. Adjustments between ERSST.v3b and ERSST.v4 increase global warming over the period 1950-2015 by about 0.05°C, which is small compared with the ~1°C global warming during that period. The effect of the adjustments on total global warming between the beginning of the 20th century and 2015 is even smaller (Fig. A3b).

## A4.  Recent Global Warming Rate

Recent warming removes the illusion of a hiatus of global warming since the 1997-98 El Niño (Fig. 2). Several studies, including Trenberth and Fasullo (2013), England et al. (2014), Dai et al. (2015), Rajaratnam et al. (2015) and Medhaug et al. (2017), showed that temporary plateaus are consistent with expected long-term warming due to increasing atmospheric GHGs. Other analyses of the 1998-2013 plateau illuminate the roles of unforced climate variability and natural and human-caused climate forcings in climate change, with the Interdecadal Pacific Oscillation (a recurring pattern of ocean-atmosphere climate variability) playing a major role in the warming slowdown (Kosaka and Xie, 2013; Huber and Knutti, 2014; Meehl et al., 2014; Fyfe et al., 2016; Medhaug et al., 2017).

## A5.  Coincidence of 1880-1920 Mean and Preindustrial Global Mean Temperatures

The Framework Convention (UNFCCC, 1992) and Paris Agreement (2015) define goals relative to 'preindustrial' temperature, but do not define that period. We use 1880-1920, the earliest time with near global coverage of instrumental data, as the zero-point for temperature anomalies. Although human-caused increases of GHGs would be expected to have caused a small warming by then, that warming was at least partially balanced by cooling from larger than average

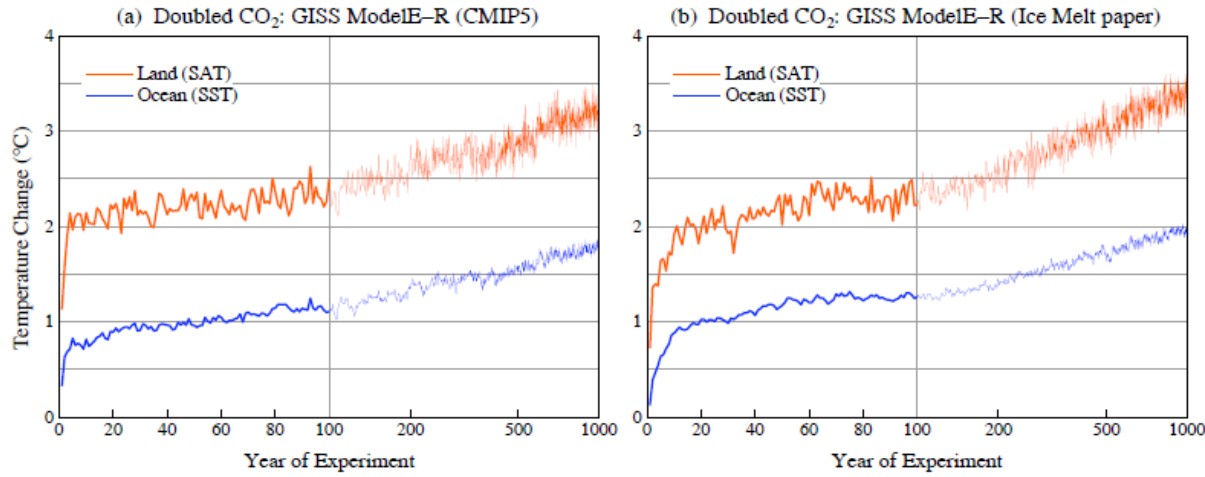

**Figure A6.** 1000-year temperature response in two versions of GISS ModelE-R. (a) version used for CMIP5 simulations (Schmidt et al. 2014), which has higher resolution (40-layer atmosphere at 2°×2.5°, 32-layer ocean at 1°×1.25°), (b) version used by Hansen et al. (2016), which has coarse resolution (20-layer atmosphere at 4°×5°, 12-layer ocean at 4°×5°) and includes two significant improvements to small scale ocean mixing, cf. section 3.2 of Hansen et al. (2016).

volcanic activity in 1880-1920. Extreme Little Ice Age conditions may have been ~0.1C cooler than the 1880-1920 mean (Abram et al., 2016), but the Little Ice Age is inappropriate to define preindustrial because the deep ocean temperature did not have time to reach equilibrium. Thus preindustrial global temperature has uncertainty of at least 0.1°C, and the 1880-1920 period, which has the merit of near-global data, yields our best estimate of preindustrial temperature.

## A6. Land versus Ocean Warming at Equilibrium

Observations (Fig. A3a) show surface air temperature (SAT) over land increasing almost twice as much as sea surface temperature (SST) during the past century. This large difference is likely partly due to the thermal inertia of the ocean, which has not fully responded to the climate forcing due to increasing GHGs. However, land warming is heavily modulated by the ocean temperature, so land temperature too has not achieved its equilibrium response.

We use long climate model simulations to examine how much the ratio of land SAT change over ocean SST change (the observed quantities) is modified as global warming approaches its equilibrium response. This ratio is ~1.8 in years 901-1000 of doubled $CO_2$ simulations (Fig. A6) for two versions of GISS modelE-R (Schmidt et al., 2014; Hansen et al., 2016).

## A7. Earth's Energy Imbalance

Hansen et al. (2011) inferred an Earth energy imbalance with the solar cycle effect removed of $+0.75 \pm 0.25$ W/m², based on an imbalance of 0.58 W/m² during the 2005-2010 solar minimum, based on the analysis of von Schuckmann and Le Traon (2011) for heat gain in the upper 2 km of the ocean and estimates of small heat gains by the deep ocean, continents, atmosphere, and net melting of sea ice and land ice. The von Schuckmann and Le Traon analysis for 2005-2015 (Fig. 5) yields a decade-average 0.7 W/m² heat uptake in the upper 2 km of the ocean; addition of the smaller terms raises the imbalance to at least $+0.8$ W/m² for 2005-2015, consistent with the recent estimate of $+0.9 \pm 0.1$ W/m² by Trenberth et al. (2016) for 2005-2015. Other recent analyses including the most up-to-date corrections for ocean instrumental biases yield $+0.4 \pm 0.1$


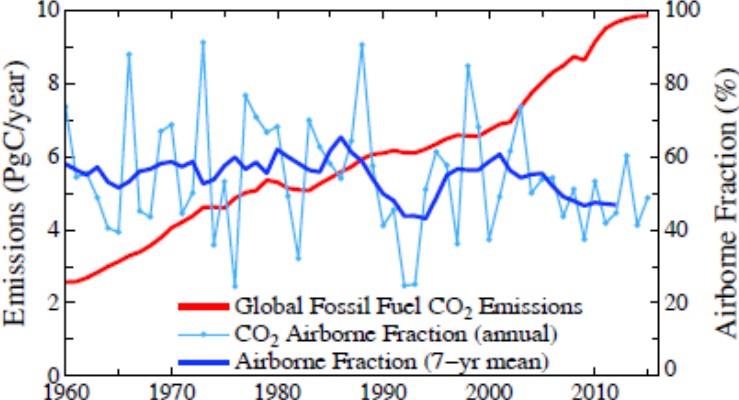

**Figure A8.** Fossil fuel $CO_2$ emissions (left scale) and airborne fraction, i.e., the ratio of observed atmospheric $CO_2$ increase to fossil fuel $CO_2$ emissions.

W/m² by Cheng et al. (2017) for the period 1960-2015 and +0.7 ± 0.1 W/m² by Dieng et al.
(2017) for the period 2005-2013. We conclude that the estimate of +0.75 ± 0.25 W/m² for the current Earth energy imbalance averaged over the solar cycle is still valid.

### A8.  $CO_2$ and $CH_4$ Growth Rates

Growth of airborne $CO_2$ is about half of fossil fuel $CO_2$ emissions (Fig. A8), the remaining
portion of emissions being the net uptake by the ocean and biosphere (Ciais et al., 2013). Here we use the Keeling et al. (1973) definition of airborne fraction, which is the ratio of quantities that are known with good accuracy: the annual increase of $CO_2$ in the atmosphere and the annual amount of $CO_2$ injected into the atmosphere by fossil fuel burning. The data reveal that, even as fossil fuel emissions have increased by a factor of four over the past half century, the ocean and
biosphere have continued to take up about half of the emissions (Fig. A8, right-hand scale). This seemingly simple relation between emissions and atmospheric $CO_2$ growth is not predictive as it depends on the growth rate of emissions being maintained, which is not true in cases with major changes in the emission scenario, so we use a carbon cycle model in Section 7 to compute atmospheric $CO_2$ as a function of emission scenario.
Oscillations of annual $CO_2$ growth are correlated with global temperature and with the El Niño/La Niña cycle[15]. Correlations (Fig. 6) are calculated for the 12-month running means, which effectively remove the seasonal cycle and monthly noise. Maxima of the $CO_2$ growth rate lag global temperature maxima by 7-8 months (Fig. 6b) and lag Niño3.4 [latitudes 5N-5S, longitudes 120-170W] temperature by ~10 months. These lags imply that the current $CO_2$
growth spike (Fig. 6 uses data through January 2017), associated with the 2015-16 El Niño, is well past its maximum, as Niño3.4 peaked in December 2015 and the global temperature anomaly peaked in February 2016.
CH₄ growth rate has varied over the past two decades, probably driven primarily by changes in emissions, as observations of $CH_3CCl_3$ show very little change in the atmospheric sink for

---

[15] One mechanism for greater than normal atmospheric $CO_2$ growth during El Niños is the impoverishment of nutrients in equatorial Pacific surface water and thus reduced biological productivity that result from reduced upwelling of deep water (Chavez et al., 1999). However, the El Niño/La Niña cycle seems to have an even greater impact on atmospheric $CO_2$ via the terrestrial carbon cycle through effects on the water cycle, temperature, and fire, as discussed in a large body of literature (referenced, e.g., by Schwalm et al., 2011).

$CH_4$ (Montzka et al., 2011; Holmes et al., 2013). Recent box-model inversions of the $CH_4$-$CH_3$-$CCl_3$ system have argued for large fluctuations in the atmospheric sink over this period but there is no identified cause for such changes (Rigby et al., 2017; Turner et al., 2017; Prather and Holmes, 2017). Future changes in the sink could lead to increased atmospheric $CH_4$ separate from emission changes, but this effect is difficult to project and not included in the RCP
scenarios (Voulgarakis et al., 2013).

       Carbon isotopes provide a valuable constraint (Saunois et al., 2016) that aids analysis of which $CH_4$ sources[16] contribute to the $CH_4$ growth resurgence in the past decade (Fig. 7). Schaefer et al. (2016) conclude that the growth was primarily biogenic, thus not fossil fuel, and located outside the tropics, most likely ruminants and rice agriculture. Such an increasing
biogenic source is consistent with effects of increasing population and dietary changes (Tilman and Clark, 2014). Nisbet et al. (2016) concur with Schaefer et al. (2016) that the $CH_4$ growth is from biogenic sources, but from the latitudinal distribution of growth they conclude that tropical wetlands[17] have been an important contributor to the $CH_4$ increase. Their conclusion that increasing tropical precipitation and temperature may be major factors driving $CH_4$ growth
suggests the possibility that the slow climate-methane amplifying feedback might already be significant. There is also concern that global warming will lead to a massive increase of $CH_4$ emissions from methane hydrates and permafrost (O'Connor et al., 2010), but as yet there is little evidence for a substantial increase of emissions from hydrates or permafrost either now or over the last 1,000,000 years (Berchet et al., 2016; Warwick et al., 2016; Quiquet et al., 2015).
Schwietzke et al. (2016) use isotopic constraints to show that the fossil fuel contribution to atmospheric $CH_4$ is larger than previously believed, but total fossil fuel $CH_4$ emissions are not increasing. This conclusion is consistent with the above studies, and it does not contradict evidence of increased fossil fuel $CH_4$ emissions at specific locations (Turner et al., 2016). A recent inverse model study, however, contradicts the satellite studies and finds no evidence for
increased US emissions (Bruhwiler et al., 2017). The recent consortium study of global $CH_4$ emissions finds with top-down studies that the recent increase is likely due to biogenic (natural and human sources) sources in the tropics, but it is difficult to attribute the magnitude of the rise to tropical wetlands alone (Saunois et al., 2017)


## A9. CO₂ Emissions in Historical Period

For land use $CO_2$ emissions in the historical period, we use the values labeled Houghton/2 by Hansen et al (2008), which were shown in the latter publication to yield good agreement with observed $CO_2$. We use fossil fuel $CO_2$ emissions data for 1850-2013 from Boden et al. (2016).
BP (2016) fuel consumption data for 2013-2015 are used for the fractional annual changes of each nation to allow extension of the Boden analysis through 2015. Emissions were almost flat from 2014 to 2015, due to economic slowdown and increased use of low-carbon energies, but, even if a peak in global emissions is near, substantial decline of emissions is dependent on acceleration in the transformation of energy production and use (Jackson et al., 2016).

---

[16] Estimated human-caused $CH_4$ sources (Ciais et al., 2013) are: fossil fuels (29%), biomass/biofuels (11%), Waste and landfill (23%), ruminants (27%) and rice (11%)

[17] Wetlands compose a majority of natural $CH_4$ emissions and are estimated to be equivalent to about 36% of the anthropogenic source (Ciais et al., 2013)

## A10. Tables of Effective Climate Forcings, 1850-2100

```
--------------------------------------------------------------------------
```
**Table A10a.**  Effective forcings (W/m2) in 1850-2015 relative to 1850
```
--------------------------------------------------------------------------
```

| Year | CO$_2$ | [a]CH$_4$ | [b]CFCs | N$_2$O | [c]O$_3$ | [d]TA+SA | [e]Volcano | Solar | Net |
|------|-------|-------|-------|-------|-------|-------|-------|-------|-------|
| 1850 | 0.000 | 0.000 | 0.000 | 0.000 | 0.000 | 0.000 | -0.083 | 0.000 | -0.083 |
| 1860 | 0.024 | 0.013 | 0.000 | 0.004 | 0.004 | -0.029 | -0.106 | 0.032 | -0.058 |
| 1870 | 0.048 | 0.027 | 0.000 | 0.008 | 0.009 | -0.058 | -0.014 | 0.048 | 0.068 |
| 1880 | 0.109 | 0.041 | 0.000 | 0.011 | 0.014 | -0.097 | -0.026 | -0.049 | 0.003 |
| 1890 | 0.179 | 0.058 | 0.000 | 0.014 | 0.018 | -0.146 | -0.900 | -0.070 | -0.847 |
| 1900 | 0.204 | 0.077 | 0.001 | 0.017 | 0.023 | -0.195 | -0.040 | -0.063 | 0.024 |
| 1910 | 0.287 | 0.115 | 0.002 | 0.022 | 0.026 | -0.250 | -0.072 | -0.043 | 0.087 |
| 1920 | 0.348 | 0.160 | 0.003 | 0.029 | 0.032 | -0.307 | -0.215 | -0.016 | 0.034 |
| 1930 | 0.425 | 0.206 | 0.004 | 0.037 | 0.036 | -0.364 | -0.143 | 0.014 | 0.215 |
| 1940 | 0.494 | 0.247 | 0.005 | 0.043 | 0.045 | -0.424 | -0.073 | 0.037 | 0.374 |
| 1950 | 0.495 | 0.291 | 0.009 | 0.052 | 0.056 | -0.484 | -0.066 | 0.055 | 0.408 |
| 1960 | 0.599 | 0.365 | 0.027 | 0.061 | 0.078 | -0.621 | -0.106 | 0.102 | 0.505 |
| 1970 | 0.748 | 0.461 | 0.076 | 0.075 | 0.097 | -0.742 | -0.381 | 0.093 | 0.427 |
| 1980 | 0.976 | 0.568 | 0.185 | 0.097 | 0.115 | -0.907 | -0.108 | 0.169 | 1.095 |
| 1990 | 1.227 | 0.659 | 0.303 | 0.125 | 0.117 | -0.997 | -0.141 | 0.154 | 1.447 |
| 2000 | 1.464 | 0.695 | 0.347 | 0.150 | 0.117 | -1.084 | -0.048 | 0.173 | 1.814 |
| 2005 | 1.619 | 0.651 | 0.356 | 0.162 | 0.123 | -1.125 | -0.079 | 0.019 | 1.770 |
| 2010 | 1.766 | 0.710 | 0.364 | 0.177 | 0.129 | -1.163 | -0.082 | 0.028 | 1.929 |
| 2015 | 1.927 | 0.730 | 0.373 | 0.195 | 0.129 | -1.199 | -0.100 | 0.137 | 2.192 |
```
--------------------------------------------------------------------------
```

```
--------------------------------------------------------------------------
```
**Table A10b.**  Effective forcing (W/m2) in 2016-2100 relative to 1850
```
--------------------------------------------------------------------------
```

| Year | CO$_2$ | [a]CH$_4$ | [b]CFCs | N$_2$O | [c]O$_3$ | [d]TA+SA | [e]Volcano | Solar | Net |
|------|-------|-------|-------|-------|-------|-------|-------|-------|-------|
| 2016 | 1.942 | 0.698 | 0.367 | 0.192 | 0.130 | -1.207 | -0.100 | 0.097 | 2.119 |
| 2020 | 2.074 | 0.702 | 0.373 | 0.201 | 0.130 | -1.234 | -0.100 | -0.008 | 2.139 |
| 2030 | 2.347 | 0.708 | 0.343 | 0.226 | 0.130 | -1.296 | -1.057 | -0.008 | 1.393 |
| 2040 | 2.580 | 0.735 | 0.301 | 0.254 | 0.123 | -1.350 | -0.100 | 0.027 | 2.569 |
| 2050 | 2.803 | 0.766 | 0.267 | 0.288 | 0.117 | -1.396 | -0.100 | 0.062 | 2.807 |
| 2060 | 3.017 | 0.791 | 0.243 | 0.322 | 0.111 | -1.433 | -1.208 | 0.097 | 1.940 |
| 2070 | 3.222 | 0.804 | 0.229 | 0.358 | 0.105 | -1.462 | -0.100 | 0.132 | 3.289 |
| 2080 | 3.421 | 0.792 | 0.215 | 0.391 | 0.098 | -1.484 | -0.100 | 0.167 | 3.500 |
| 2090 | 3.614 | 0.722 | 0.199 | 0.427 | 0.091 | -1.495 | -1.240 | 0.167 | 2.484 |
| 2100 | 3.801 | 0.619 | 0.191 | 0.456 | 0.085 | -1.500 | -0.100 | 0.167 | 3.719 |
```
--------------------------------------------------------------------------
```

[a]CH$_4$: CH$_4$-induced changes of tropospheric O$_3$ and stratospheric H$_2$O are included
[b]CFCs: This includes all GHGs except CO$_2$,CH$_4$,N$_2$O and O$_3$.
[c]O$_3$: Half of troposphere O$_3$ forcing + stratosphere O$_3$ forcing from IPCC (2013).
[d]TA+SA: tropospheric aerosols and surface albedo forcings combined.
[e]Volc: volcanic forcing is zero when there are no stratospheric aerosols
Annual data are available at http://www.columbia.edu/~mhs119/Burden/.

CO$_2$, CH$_4$ and N$_2$O forcings are calculated with analytic formulae of Hansen et al. (2000).  CH$_4$ forcing includes the factor 1.4 to convert adjusted forcing to effective forcing, thus incorporating the estimated effect of a CH$_4$ increase on tropospheric ozone and stratospheric water vapor.  Our CH$_4$ adjusted forcing is significantly (~17%) higher that the values in IPCC (2013), but (~9%) smaller than values of Etminan (2017).  Our factor 1.4 to convert direct radiative forcing to effective forcing is in the upper portion of the indirect effects discussed by Myhre et al. (2013), so our net CH$_4$ forcing agrees with Etminan et al. (2017) within uncertainties.

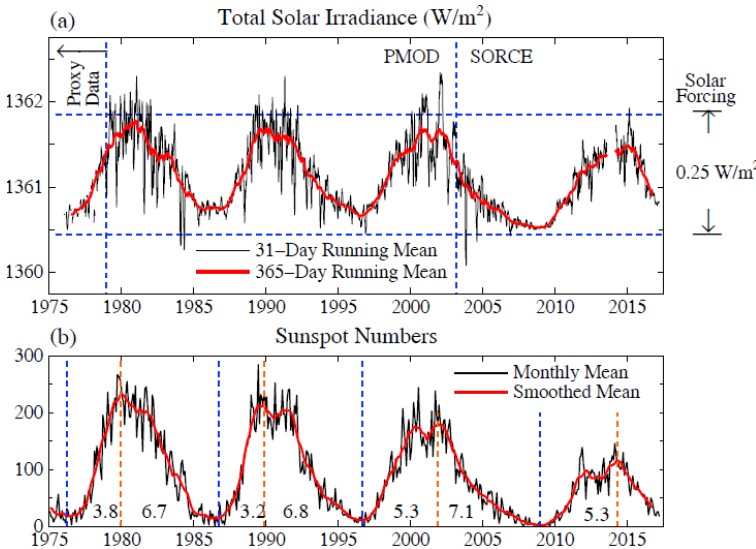

**Figure A11.** Solar irradiance and sunspot number in the era of satellite data. Left scale is the energy passing through an area perpendicular to Sun-Earth line. Averaged over Earth's surface the absorbed solar energy is ~240 W/m$^2$, so the full amplitude of the measured solar variability is ~0.25 W/m$^2$.

## A11. Solar Irradiance

Solar irradiance has been measured from satellites since the late 1970s. Fig. A11 is a composite of several satellite-measured time series. Data through 28 February 2003 are an update of Frohlich and Lean (1998) obtained from Physikalisch Meteorologisches Observatorium Davos, World Radiation Center. Subsequent update is from University of Colorado Solar Radiation & Climate Experiment (SORCE). Historical total solar irradiance reconstruction is available at http://lasp.colorado.edu/home/sorce/data/tsi-data/ Data sets are concatenated by matching the means over the first 12 months of SORCE data. Monthly sunspot numbers support the conclusion that the solar irradiance in the current solar cycle is significantly lower than in the three preceding solar cycles.

The magnitude of the change of solar irradiance from the prior solar cycle to the current solar cycle is of the order of −0.1 W/m$^2$, which is not negligible but is small compared with greenhouse gas climate forcing. On the other hand, the variation of solar irradiance from solar minimum to solar maximum is of the order of 0.25 W/m$^2$, so the high solar irradiance in 2011 - 2015 contributes to the increase of Earth's energy imbalance between 2005-2010 and 2010-2015.

## A12. Alternative Scenario

Simulated global temperature for the climate forcings of the "alternative scenario" discussed in Section 6 are shown in Fig. A12. The climate model, with sensitivity 3°C for doubled $CO_2$, is the same as used for Fig. 12.

## A13. Non-$CO_2$ GHGs

$CO_2$ is the dominant forcing in future climate scenarios. Growth of non-$CO_2$ GHG climate forcing is likely to be even smaller, relative to $CO_2$ forcing, than in recent decades (Fig. 8) if there is a strong effort to limit climate change. Indeed, recent agreement to use the Montreal

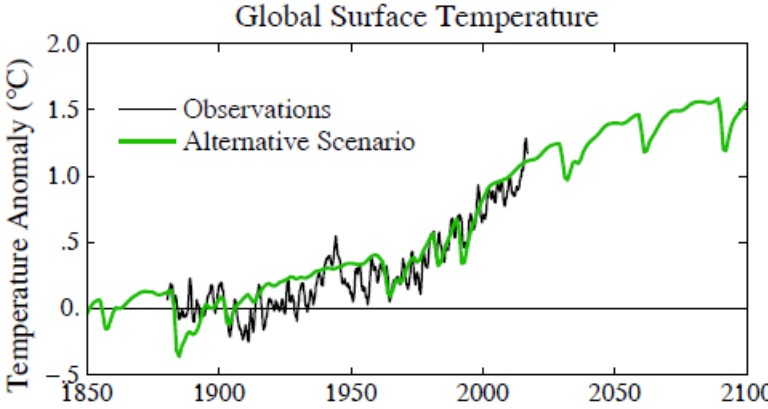

**Figure A12.** Simulated global temperature with historical climate forcings to 2000 followed by the alternative scenario. Historical climate forcings are discussed in the main text.

Protocol (2016) to phase down production of minor trace gases, the hydrofluorocarbons (HFCs), should cause annually added forcing of Montreal Protocol Trace Gases (MPTGs) + Other Trace Gases (OTGs) (red region in Fig. 8) to become near zero or slightly negative, thus at least partially off-setting growth of other non-$CO_2$ GHGs, especially $N_2O$.

Methane ($CH_4$) is the largest climate forcing other than $CO_2$ (Fig. 4). The $CH_4$ atmospheric lifetime is only about 10 years (Prather et al., 2012), so there is potential to reduce this climate forcing rapidly if $CH_4$ sources are reduced. Our climate simulations, based on the RCP6.0 non-$CO_2$ GHG scenarios, follow an optimistic path in which $CH_4$ increases moderately in the next few decades to 1960 ppb in 2070 and then decreases rapidly to 1650 ppb in 2100, yielding a forcing change of −0.1 W/m². However, the IPCC (Kirtman et al., 2013) uses a more modern chemical model projection for the RCP anthropogenic emissions and gives a less beneficial view with a decrease to only 1734 ppb and a forcing change of −0.03 W/m². RCP2.6 makes a more optimistic assumption: that $CH_4$ will decline monotonically to 1250 ppb in 2100, yielding a forcing of −0.3 W/m² (relative to today's 1800 ppb $CH_4$), but the IPCC projections of RCP2.6 reduce this to −0.2 W/m² (Kirtman et al., 2013).

Observed atmospheric $CH_4$ amount (Fig. A13a) is diverging on the high side of these optimistic scenarios. The downward offset (~20 ppb) of $CH_4$ scenarios relative to observations (Fig. A13a) is due to the fact that RCP scenarios did not include a data adjustment that was made in 2005 to match a revised $CH_4$ standard scale (E. Dlugokencky, pers. comm.), but observed $CH_4$ is also increasing more rapidly than in most scenarios. Reversal of $CH_4$ growth is made difficult by increasing global population, the diverse and widely distributed nature of agricultural sources, and global warming "in the pipeline," as these trends create an underlying tendency for increasing $CH_4$. The discrepancy between observed and assumed $CH_4$ growth could also be due in part to increased natural sources or changes in the global OH sink (Dlugokencky et al., 2011; Turner et al., 2017). Evidence for increased natural sources in a warmer climate are suggested by glacial-interglacial $CH_4$ increases of the order of 300 ppb, and contributions to observed fluctuations cannot be ruled out on the basis of recent budgets (Ciais et al., 2013).

Methane emissions from rice agriculture and ruminants potentially could be mitigated by changing rice growing methods (Epule et al., 2011) and inoculating ruminants (Eckard et al., 2010; Beil, 2015)], but that would require widespread adoption of new technologies at the farmer level. California, in implementing a state law to reduce GHG emissions, hopes to dramatically cut agricultural $CH_4$ emissions (see www.arb.ca.gov/cc/scopingplan/scopingplan.htm), but

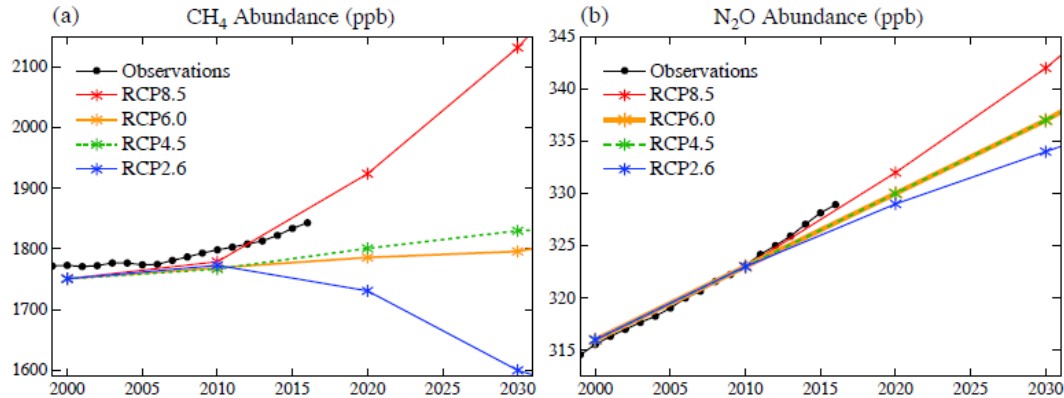

**Figure A13.** Comparison of observed $CH_4$ and $N_2O$ amounts with RCP scenarios. RCP 6.0 and 4.5 scenarios for $N_2O$ overlap. Observations are from NOAA/ESRL Global Monitoring Division. Natural sources and feedbacks not included in RCP scenarios may contribute to observed growth (see discussion).

California has one of the most technological and regulated agricultural sectors in the world. It is not clear that this level of management can occur in the top agricultural $CH_4$ emitters like China, India and Brazil. Methane leaks from fossil fuel mining, transportation and use can be reduced, indeed, percentage leakage from conventional fossil fuel mining and fuel use has declined

substantially in recent decades (Schwietzke et al, 2016), but there is danger of increased leakage with expanded shale gas extraction (Caulton et al., 2014; Petron et al., 2013; Howarth, 2015; Kang et al., 2016).

Observed $N_2O$ growth is exceeding all scenarios (Fig. A13b). Major quantitative gaps remain in our understanding of the nitrogen cycle (Kroeze and Bouwman, 2011), but fertilizers

are clearly a principal cause of $N_2O$ growth (Röckmann and Levin 2005; Park et al., 2012). More efficient use of fertilizers could reduce $N_2O$ emissions (Liu and Zhang, 2011), but considering the scale of global agriculture, and the fact that fixed N is an inherent part of feeding people, there will be pressure for continued emissions at least comparable to present emissions. In contrast, agricultural $CH_4$ emissions are inadvertent and not core to food production. Given

the current imbalance [emissions exceeding atmospheric losses by about 30% (Prather et al., 2012)] and the long $N_2O$ atmospheric lifetime ($116 \pm 9$ years; Prather et al., 2015) it is nearly inevitable that $N_2O$ will continue to increase this century, even if emissions growth is checked. There can be no expectation of an $N_2O$ decline that offsets the need to reduce $CO_2$.

The Montreal Protocol has stifled and even reversed growth of specific trace gases that

destroy stratospheric ozone and cause global warming (Prather et al., 1996; Newman et al., 2009). The anticipated benefit over the 21st century is a drop in climate forcing of $-0.23$ W/m$^2$ (Prather et al., 2013). Protocol amendments that add other gases such as HFCs are important; forcings of these gases are small today, but without the Protocol their potential for growth is possibly as large as $+0.2$ W/m$^2$ (Prather et al., 2013).

We conclude that a 0.25 W/m$^2$ decrease of climate forcing by non-$CO_2$ GHGs is plausible, but requires a dramatic change from the growing abundances of these gases today. Achievement requires (i) successful phase-out of MPTGs ($-0.23$ W/m$^2$), (ii) reduction of $CH_4$ forcing by 0.12 W/m$^2$, and (iii) limiting $N_2O$ increase to 0.1 W/m$^2$. A net negative forcing of $-0.25$ W/m$^2$ for non-$CO_2$ gases would allow $CO_2$ to be 365 ppm, rather than 350 ppm, while yielding the same

total GHG forcing. Thus potential reduction of non-$CO_2$ gases is helpful, but it does not alter the need for rapid fossil fuel emission reduction.

## Acknowledgments

Support of the Climate Science, Awareness and Solutions program has been provided by the Durst family, the Grantham Foundation for Protection of the Environment, Jim and Krisann Miller, Gary Russell, Gerry Lenfest, the Flora Family Foundation, Elisabeth Mannschott, Alexander Totic and Hugh Perrine, which is gratefully acknowledged. DJB acknowledges funding through a Leverhulme Trust Research Centre Award (RC-2015-029). EJR acknowledges support from Australian Laureate Fellowship FL12 0100050

We appreciate the generosity, with data and advice, of Tom Boden, Ed Dlugokencky, Robert Howarth, Steve Montzka, Larissa Nazarenko and Nicola Warwick, the thoughtful reviews of the anonymous reviewers and several SC commenters, and assistance of the editor James Dyke.

The authors declare that they have no conflicts of interest. The first author (JH) notes that he is a plaintiff in the lawsuit Juliana et al. vs United States.

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
