# Peer review of "Young People's Burden: Requirement of Negative CO2 Emissions"

_Earth System Dynamics, 2016_

## Short Comment (SC1) · 4 Oct 2016

The chief editors of Earth System Dynamics (ESD) post this note to clarify issues related to the open review process and discussion phase of manuscripts in this journal.

1. The manuscript has not been accepted for publication in ESD. Its availability on this web page is part of an open and transparent scientific evaluation process.

2. The purpose of the discussion forum is to transparently evaluate the scientific content of the manuscript by other scientists. The forum is open only to scientific comments and reviews.

3. At the end of this process, the handling editor will decide whether or not this manuscript will be published in Earth System Dynamics or not.

4. Comments that are not of scientific nature or of direct relevance to the issues raised in the manuscript or which contain personal insults can be removed by the editor.

5. Earth System Dynamics does not recommend media attention of manuscripts that are in the discussion phase, as these manuscripts are not scientifically evaluated regarding their contents. If suitable for media coverage, this is recommended to take place after a manuscript has been reviewed, accepted, and been published in Earth System Dynamics.

---

## Short Comment (SC2) · 6 Oct 2016

typo in line 190: replace "ezaggerated" by "exaggerated".

---

## Short Comment (SC3) · 11 Oct 2016

Soil Biology is our only way to rapidly and massively draw down CO2 from the air to offset our ongoing and past carbon emissions, It Can safely and naturally restore the hydrological cycles by increasing biogenic aerosols and cloud albedo that can readily cool the planet by the 3 watts/m2 needed to offset the now locked in greenhouse warming effects and avoid the Storms of Our Grandchildren.

The French have lead the way recognizing Soil Carbons' value and committing to build Soil Carbon by 0.40% annually. Putting them on the road to Carbon Negativity before any industrialized country. 25 nations have signed on to 4p1000. 100 of the 196 countries in Paris submitted plans to reduce CO2 via agriculture, forestry and replacing soil carbon into their programmes. http://4p1000.org/understand

A combination of Best Management Practices, (BMPs), for Agriculture, Grazing & Forestry with bioenergy systems which build soil carbon can deliver the giga-tons of carbon necessary into the soil sink bank.

Ag BMPs; 1 GtC, New Forest & BMPs; 1 GtC Pyrolitic Bioenergy, Cooking Stoves; nearly 1/2 GtC Industrial Pyrolitic Bioenergy; 2 GtC Holistic Grazing; 2+ GtC

Over 6 GtC, So soils & biota can do more than half the 10 GtC reduction job, feeding carbon to life instead of death. Carbon Sequestration Cascade; Each Black Carbon gram (biochar & humus) can increase Water Retention by 8 grams, and can support 10 grams of Green Carbon, which each can feed up to 10 more grams of fungal mycelium White Carbon growth

Carbon has been fundamental to life since the birth of our planet. It's the source of all wealth and the conduit of all joy. Carbon cycles among and between billions of interconnected earthlings, whose fates teeter on the element's return trip to the soil. Only the generous reciprocity inherent to life macrocycles can restore abundance and harmony to the planet of the living. May we celebrate a happy Intended Anthropocene, anointed in water & Soil rather than Oil and Blood.

For a complete review of the current science & industry applications of Biochar please see my USBI 2016 Presentation; http://usbi2016.org/schedule/

(Slides & down load:) http://biochar-us.org/presentation/civilization-soil-hall-marks-unintended-intended-anthropocene

How thermal conversion technologies can integrate and optimize the recycling of valuable nutrients while providing energy and building soil carbon, I believe it brings together both sides of climate beliefs. A reconciling of both Gods' and mans' controlling hands. "The Civilization of Soil", Hall Marks of The Unintended & Intended Anthropocene (The full text & citations are on my LinkedIn page; https://www.linkedin.com/pulse/civilization-soils-hall-marksunintended-intended-erich-j-knight?trk=prof-post I have come to see a new definition of the Anthropocene, divided into two parts, The Unintended and the Intended. The measure which will codify this new age, a manifestation of the Intended Anthropocene, will be increased soil carbon content & with that the restoration of ecosystems and climate stability world wide.

The primary measurement and establishment of the Unintended is the Ag revolution 12K years ago. Measured by many proxies of Land use changes, (LUC), The Intended Anthropocene has a single very hard measurement of radionuclides, so we have a hard date of inception; 1964, the test ban treaty. (rocks & sediments don't lie). Around this time Rachael Carson also clued us in to chemical LUC in "Silent Spring". Since that time the offending measures have declined.

So the Holocene, the age of man, can be measurably divided into Intended & Unintended eras. Now we are at a new Intended age which I call; The Diamond Age.

If I May be so bold,... As I speak for Biologic Carbon... I speak for the very center of life itself. We have been burning it for over one million years, exploiting it out of the soil for 12,000 years and combusting fossil carbon for over 150 years. Now, we can grow nano-structured fossil carbons into an unprecedented variety of materials and even human tissues.

The Stone Age did not end for a lack of stones, as well, the Combustion Age will not end for lack of fossil fuels. Nanotechnology and Terra Preta Soil Technology has thrust The Diamond Age upon us, with it, the rectification of the Carbon and Nitrogen Cycles, Restoring Soil Ecology, In turn rectifying the Hydrologic and Climate Cycles, this train is leaving the station, either get on board or be left in the combusted soot and CO2 pollution of history! Since we have filled the air, filling the seas to full, Soil is the only ubiquitous and economic place to put it.

Thank you for your efforts

[Figure]

[Figure]

[Figure]

**The Civilization of Soils**

**Hallmarks of the
Unintended & Intended Anthropocene**

**Erich J. Knight
Shenandoah Gardens
erichjknight@gmail.com**

**Fig. 1.** "The Civilization of Soil"

---

## Author Comment (AC1) · 11 Oct 2016

A second sentence has been added to the authors' Conflict Statement (line 1030), which now reads:

The authors declare that they have no conflicts of interest. The first author (JH) notes that he is a plaintiff in the lawsuit Juliana et al vs United States (2016).

The misspelling of exaggerated in line 190 noted by Bernardus Mettes (SC2) has been corrected.

The manuscript revised to include these changes has been uploaded.

Please also note the supplement to this comment:

[Figure]

http://www.earth-syst-dynam-discuss.net/esd-2016-42/esd-2016-42-AC1-supplement.pdf

**[ESDD](ESDD)**
[Figure]

**Supplement:**

**Young People's Burden: Requirement of Negative CO$_2$ Emissions**

**James Hansen,[1] Makiko Sato,[1] Pushker Kharecha,[1] Karina von Schuckmann,[2] David J Beerling,[3] Junji Cao,[4] Shaun Marcott,[5] Valerie Masson-Delmotte,[6] Michael J Prather,[7] Eelco J Rohling,[8,9] Jeremy Shakun,[10] Pete Smith[11]**

[1]Climate Science, Awareness and Solutions, Columbia University Earth Institute, New York, NY 10115 [2]Mercator Ocean, 10 Rue Hermes, 31520 Ramonville St Agne, France [3]Leverhulme Centre for Climate Change Mitigation, University of Sheffield, Sheffield S10 2TN, UK [4]Key Lab of Aerosol Chemistry and Physics, SKLLQG, Institute of Earth Environment, Xi'an 710061, China [5]Department of Geoscience, 1215 W. Dayton St., Weeks Hall, University of Wisconsin-Madison, Madison, WI 53706 [6]Institut PierreSimon Laplace, Laboratoire des Sciences du Climat et de l'Environnement (CEA-CNRS-UVSQ) Université Paris Saclay, Gif-sur-Yvette, France [7]Earth System Science Department, University of California at Irvine, CA [8]Research School of Earth Sciences, The Australian National University, Canberra, 2601, Australia [9]Ocean and Earth Science, University of Southampton, National Oceanography Centre, Southampton, SO14 3ZH, UK [10]Department of Earth and Environmental Sciences, Boston College, Chestnut Hill, MA 02467 [11]Institute of Biological and Environmental Sciences, University of Aberdeen, 23 St Machar Drive, AB24 3UU, UK

**E-mail**: jeh1@columbia.edu

**Keywords**: climate change, carbon budget, intergenerational justice

**Abstract**

The rapid rise of global temperature that began about 1975 continues at a mean rate of about 0.18°C/decade, with the current annual temperature exceeding +1.25°C relative to 1880-1920. Global temperature has just reached a level similar to the mean level in the prior interglacial (Eemian) period, when sea level was several meters higher than today, and, if it long remains at this level, slow amplifying feedbacks will lead to greater climate change and consequences. The growth rate of climate forcing due to human-caused greenhouse gases (GHGs) increased over 20% in the past decade mainly due to resurging growth of atmospheric CH$_4$, thus making it increasingly difficult to achieve targets such as limiting global warming to 1.5°C or reducing atmospheric CO$_2$ below 350 ppm. Such targets now require "negative emissions", i.e., extraction of CO$_2$ from the atmosphere. If rapid phasedown of fossil fuel emissions begins soon, most of the necessary CO$_2$ extraction can take place via improved agricultural and forestry practices, including reforestation and steps to improve soil fertility and increase its carbon content. In this case, the magnitude and duration of global temperature excursion above the natural range of the current interglacial (Holocene) could be limited and irreversible climate impacts could be minimized. In contrast, continued high fossil fuel emissions by the current generation would place a burden on young people to undertake massive technological CO$_2$ extraction, if they are to limit climate change. Proposed methods of extraction such as bioenergy with carbon capture and storage (BECCS) or air capture of CO$_2$ imply minimal estimated costs of 104-570 trillion dollars this century, with large risks and uncertain feasibility. Continued high fossil fuel emissions unarguably sentences young people to either a massive, possibly implausible cleanup or growing deleterious climate impacts or both, scenarios that should provide both incentive and obligation for governments to alter energy policies without further delay.

[Figure]

**Fig. 1.** Fossil fuel (and cement manufacture) $CO_2$ emissions based on Boden et al (2016) with BP data used to infer 2014-2015 estimates. Europe/Eurasia is Turkey plus Boden et al categories Western Europe and Centrally Planned Europe. Asia Pacific is sum of Centrally Planned Asia, Far East and Oceania. Middle East is Boden et al Middle East less Turkey. Russia is Russian Federation since 1992 and 0.6 of USSR in 1850-1991. Ships/Air is sum of bunker fuels of all nations. Can+Aus is Canada+Australia.

**1. Introduction**

Almost all nations agree on the need to limit fossil fuel emissions to avoid dangerous human-made climate change, as formalized in the 1992 Framework Convention on Climate Change (UNFCCC 1992). The Paris Agreement (2015) seeks to limit global warming to well below 2°C relative to pre-industrial levels, with an aspirational goal of staying below 1.5°C. We advocate a stricter goal, based on restoring Earth's energy balance and limiting the period when global temperature is above the range of the Holocene; temperature stability of the Holocene has allowed sea level to be stable for the past several millennia in which civilization developed. This goal leads to a $CO_2$ target of 350 ppm, which can be adjusted as $CO_2$ declines and empirical climate data accumulates (Hansen et al 2008, 2013, 2016). Either target, 1.5°C or 350 ppm, requires rapid phasedown of fossil fuel emissions.

Despite widespread recognition of the risks posed by climate change, global fossil fuel emissions continue at a high rate that tends to make these targets increasingly improbable. Emissions are growing rapidly in emerging economies; while growth slowed in China in the past two years, emissions remain high (Fig. 1). The Kyoto Protocol (1997), a policy instrument of the Framework Convention (UNFCCC 1992), spurred emission reductions in some nations, and the collapse of the Soviet Union caused a large decrease of emissions by Russia (Fig. 1b). However, growth of international ship and air emissions (Fig. 1b) largely offset these reductions and the growth rate of global emissions actually accelerated from 1.5%/year in 1973-2000 to ~2.5%/year after 2000 (Fig. A1). China is now the largest source of fossil fuel emissions, followed by the U.S. and India, but on a per capita historical basis the U.S. is 10 times more accountable than China and 25 times more accountable than India for the increase of atmospheric $CO_2$ above its preindustrial level (Hansen and Sato 2016). Tabular data for Figs. 1 and A1 are available on the web page www.columbia.edu/~mhs119/Burden.

In response to this situation, a lawsuit [Juliana et al vs United States 2016, hereafter J et al vs US 2016] was filed against the United States asking the U.S. District Court, District of Oregon, to require the U.S. government to produce a plan to rapidly reduce emissions. The suit requests that the plan reduce emissions at the 6%/year rate that Hansen et al (2013) estimated as the requirement for lowering atmospheric $CO_2$ to a level of 350 ppm. At a hearing in Eugene Oregon on 9 March 2016 the United States and three interveners (American Petroleum Institute,

National Association of Manufacturers, and the American Fuels and Petrochemical Association) asked the Court to dismiss the case, in part based on the argument that the requested rate of fossil fuel emissions reduction was implausible. Magistrate Judge Coffin stated that he was "troubled" by the severity of the requested emissions reduction rate, but he also noted that some of the alleged climate change consequences, if accurate, could be considered "beyond the pale", and he rejected the motion to dismiss the case. Judge Coffin's ruling must be certified by a second judge, after which the case can proceed to trial. It is anticipated that the plausibility of achieving the emission reductions needed to stabilize climate will be a central issue at the trial.

Urgency of initiating emissions reductions is well recognized (IPCC 2013, 2014; Huntingford et al 2012; Friedlingstein et al 2014; Rogelj et al 2016a) and was stressed in the paper that the lawsuit J et al vs US (2016) uses to prescribe an emissions reduction scenario (Hansen et al 2013). The climate research community also realizes that the goal to keep global warming less than $1.5°C$ probably requires negative net $CO_2$ emissions later this century if high global emissions continue in the near-term (Fuss et al 2014; Anderson 2015; Rogelj et al 2016b; Sanderson et al 2016). The Intergovernmental Panel on Climate Change (IPCC) reports (IPCC 2013, 2014) do not address environmental and ecological feasibility and impacts of large-scale $CO_2$ removal, but recent studies (Smith et al 2016; Williamson 2016) are taking up this crucial issue and raising the question of whether large-scale negative emissions are even feasible.

Our aim is to contribute to understanding of the threshold-required rate of $CO_2$ emissions reduction via an approach that is transparent to non-scientists. We consider the potential for reductions of non-$CO_2$ GHGs to minimize the human-made climate forcing, the potential for improved agricultural practices to store more soil carbon, and the potential drawdown of atmospheric $CO_2$ from reforestation and afforestation. Quantitative examination reveals the merits of these actions to ameliorate demands on fossil fuel $CO_2$ emission phasedown, but also the limitations, thus clarifying the urgency of government actions to rapidly advance the transition to carbon-free energies to meet the climate stabilization targets they have set.

We first describe the status of global temperature change and then summarize the principal climate forcings that drive long term climate change. We show that observed global warming is consistent with knowledge of changing climate forcings, Earth's measured energy imbalance, and the canonical estimate of climate sensitivity, i.e., about $3°C$[1] global warming for doubled atmospheric $CO_2$. We illustrate updates of GHG observations and calculate a notable acceleration during the past decade of the growth rate of GHG climate forcing. For future fossil fuel emissions we consider both the IPCC Representative Concentration Pathways (RCP) scenarios, and simple emission growth rates that are helpful for determination of the plausibility of required emission changes. We use a precisely defined Green's function calculation of global temperature with canonical climate sensitivity for each emissions scenario, thus allowing us to determine the amount of $CO_2$ that must be extracted from the air – effectively the climate debt – to achieve the targets of returning atmospheric $CO_2$ to less than 350 ppm or limiting global warming to less than $1.5°C$ above preindustrial levels. We discuss alternative extraction technologies and their estimated costs, and finally we consider the potential alleviation of $CO_2$ extraction requirements that might be obtained via special efforts to reduce non-$CO_2$ GHGs.
* * *
[1] IPCC (2013) finds that $2×CO_2$ equilibrium sensitivity is likely in the range $3 ± 1.5°C$, as was estimated by Charney et al. (1979). Median sensitivity in recent model inter-comparisons is $3.2°C$ (Andrews et al 2012; Vial et al 2013).

[Figure]

**Fig. 2.** Global surface temperature relative to 1880-1920 based on GISTEMP analysis (Appendix A). (a) Annual and 5-year means since 1880, (b) 12- and 132-month running means since 1970. Black squares in (b) are calendar year (Jan-Dec) year means used to construct (a). (b) uses data through August 2016.

**2. Global Temperature Change**

The United Nations 1992 Framework Convention on Climate Change (UNFCCC 1992) stated its objective as '…stabilization of GHG concentrations in the atmosphere at a level that would prevent dangerous anthropogenic interference with the climate system'. The 15[th] Conference of the Parties (Copenhagen Accord 2009) concluded that this objective required a goal to '…reduce global emissions so as to hold the increase of global temperature below 2°C…' and that the 2015 Conference of the Parties should consider the possibility of strengthening the temperature limit to below 1.5°C. Indeed, the Paris Agreement (2015) modified the objective 'to holding the increase of global average temperature to well below 2°C above pre-industrial levels and to pursue efforts to limit the temperature increase to 1.5°C above preindustrial levels…'.

Defining a target for limiting human interference with natural climate requires quantitative assessment of ongoing and paleo temperature changes, with the latter especially helpful for characterizing long-term ice sheet and sea level response versus temperature. We examine the modern period with near-global instrumental temperature data in the context of the current and previous (Holocene and Eemian) interglacial periods for which less precise proxy-based temperatures have recently emerged. The Holocene, now over 11,700 years in duration, has had relatively stable climate. The Eemian, lasting from about 130,000 to 115,000 years ago, was moderately warmer than the Holocene.

**2.1. Modern Temperature**

The several analyses of temperature change since 1880 are in close agreement (Hartmann et al 2013). Thus we can use the current GISTEMP analysis (see Supporting Information), which is updated monthly and available (http://www.columbia.edu/~mhs119/Temperature/).

The popular measure of global temperature is the annual-mean global-mean value (Fig. 2a), which is publicized at the end of each year. However, as discussed by Hansen et al. (2010), the 12-month running mean global temperature is more informative and removes monthly "noise" from the record just as well as the calendar year average. For example, the 12-month running

mean for the past 35 years (Fig. 2b) defines clearly the super-El Niños of 1997-98 and 2015-16 and the 3-year cooling after the Mount Pinatubo volcanic eruption in the early 1990s.

Global temperature in each month for the past year has been at or near a record for the month. Perhaps helped by a popular "spiral" temperature visualization (Hope 2016), this has tended to create a popular impression that global temperature may be spiraling out of control. This series of monthly records is likely to terminate soon and the 12-month running mean is expected to decline as it has after prior El Niños. However, the year-to-date temperature is so far above the prior record already that even the steepest post-El Niño decline cannot prevent a 2016 annual record temperature.

One effect of the recent warming is to remove unequivocally the illusion of a global warming hiatus after the 1997-98 El Niño. Several studies, including Trenberth and Fasullo (2013), England et al. (2014), Dai et al. (2015) and Rajaratnam et al (2015), showed that temporary plateaus are consistent with expected long-term warming due to increasing atmospheric GHGs. Other analyses of this specific plateau help illuminate the roles of unforced climate variability and natural and human-caused climate forcings in observed climate change, with the Interdecadal Pacific Oscillation (a recurring pattern of ocean-atmosphere climate variability) playing a major role in the warming slowdown (Kosaka and Xie, 2013; Meehl et al, 2014; Fyfe et al, 2016).

Global temperature defined by the linear fit over recent decades has now reached +1.06°C relative to the 1880-1920 average (Fig. 2), the 12-month running mean temperature through August 2016 is 1.30°C, and the 2016 temperature likely will be near +1.25°C. The present global warming rate, based on a linear fit through the past 45 years (dashed line in Fig. 2b) is +0.18°C per decade. At this rate, the trend line of global temperature, which is a relevant measure of mean temperature, will reach +1.5°C in about 2040 and +2°C in the late 2060s. However, the warming rate can accelerate or decelerate, depending on policies that affect GHG emissions, developing climate feedbacks, and other factors discussed below.

**2.2. Temperature during current and prior interglacial periods**

Holocene temperature has been reconstructed at centennial-scale resolution from 73 globally distributed proxy temperature records by Marcott et al (2013). This record shows a decline of 0.6°C from early Holocene maximum temperature to a "Little Ice Age" minimum in the early 1800s [that minimum being better defined by higher resolution data of Abram et al (2016)].

Concatenation of the modern and Holocene temperature records (Fig. 3) assumes, based on Abram et al (2016), that the 1880-1920 mean temperature is 0.1°C warmer than the Little Ice Age minimum. The early Holocene maximum in the Marcott et al (2013) data is thus at +0.5°C relative to the 1880-1920 mean of the modern data. However, model simulations suggest that the reconstructed early Holocene maximum may be exaggerated due to limitations of the proxy data, especially potential seasonality bias, as discussed by Marcott et al (2013) and Liu et al (2014).

Even though we cannot be certain that the current year is warmer than any single year earlier in the Holocene due to centennial smoothing of the Holocene stack and original resolution of the underlying proxy records (Marcott et al 2013), we conclude that the ongoing global warming trend (1.06°C over 115 years, Fig 2b) is already well above prior centennially smoothed Holocene temperature. Further, we suggest that these smoothed temperatures are relevant to

[Figure]

**Fig. 3.** Estimated average global temperature for the last interglacial (Eemian) period (McKay et al 2011; Clark and Huybers 2009; Turney and Jones 2010), the centennially-smoothed Holocene (Marcott et al 2013) temperature as a function of time, and the 11-year mean of modern data (Fig. 2). Vertical downward arrows indicate likely overestimates (see text).

important climatic features that change on long time scales, such as ocean warming (von Schuckmann et al 2016), ice sheet stability (DeConto and Pollard 2016), shifting climatic zones (Seidel et al 2008), and the frequency of climate extremes (Hansen and Sato 2016). The formal 2σ (95% confidence) uncertainty in the Marcott et al (2013) Holocene temperature curve is only ~0.25°C. Although total uncertainty is larger, because of issues such as discussed by Liu et al (2014), those uncertainties tend to push the early Holocene temperatures lower, increasing the gap between today's temperature and early Holocene temperature (Marcott and Shakun 2015).

We also conclude that the modern trend line of global temperature crossed the early Holocene (smoothed) temperature maximum (+0.5°C) already in about 1985. This conclusion receives support from the accelerating rate of sea level rise, which approached a rate of 3 mm/year at about that date (Fig. 29 of Hansen et al 2016 shows a relevant concatenation of measurements). Such a high rate of sea level rise, which equates to 3 meters per millennium, far exceeds rates of Holocene sea level rise except in the earliest Holocene when melt was still coming from the final decay of mid-latitude ice sheets (Dutton et al 2015).

The Framework Convention (UNFCCC 1992) and Paris Agreement (2015) define goals relevant to 'preindustrial' temperature, but do not define that period. We use 1880-1920, the earliest time with good global coverage of instrumental data, as the zero-point for temperature anomalies. Alternatively, one might argue for defining preindustrial as the Little Ice Age minimum temperature, but the deep ocean did not have time to reach equilibrium with those brief conditions, and global mean Little Ice Age temperature was probably only ~0.1C cooler than the 1880-1920 mean (Abram et al 2016).

The important point is that the relevant mean global temperature has already risen out of the centennial Holocene range. Global warming is already having substantial adverse climate impacts (IPCC 2014), including extreme events (NAS 2016), and there is widespread agreement that 2°C warming would commit the world to multi-meter sea level rise (Levermann et al 2013; Clark et al 2016), and a case has been made that this could unfold within 50-150 years (Hansen et al 2016).

The prior interglacial period, the Eemian, was warmer than the Holocene and sea level reached heights 6-9 m (20-30 feet) higher than today (Dutton et al (2015). McKay et al (2011)

estimated peak Eemian annual global ocean SST as $+0.7°C \pm 0.6°C$, while models, as described by Masson-Delmotte (2013), give more confidence to the lower part of that range.  Global ocean

235 SST response to climate forcings is typically 70-75% as large as the global mean (land + ocean) surface temperature response and that same proportion is found empirically in the warming of the past century (http://www.columbia.edu/~mhs119/Temperature/T_moreFigs/).  Thus the McKay et al data are equivalent to a global Eemian temperature $+1°C$ relative to the Holocene. Clark and Huybers (2009) and Turney and Jones (2010) estimated global temperature in the

240 Eemian as 1.5-2°C warmer than the Holocene (Fig. 3), but Bakker and Renssen (2014) analyzed the likely error in Eemian temperature estimates caused by the assumption that maximum Eemian temperatures at all proxy temperature sites occurred simultaneously and also the effect of proxy biases towards summer conditions, concluding that these biases could exaggerate Eemian temperature by $1.1 \pm 0.4°C$.  Thus, consistent with the discussion of Masson-Delmotte

245 (2013), we conclude that mean Eemian temperature was probably about 1°C warmer than the Holocene.  Given growing indications, discussed above, that the early Holocene was little warmer than the pre-industrial (1880-1920) period, we conclude that Eemian global temperature was not much more than $+1°C$ relative to 1880-1920 global temperature.

These considerations add to the question of whether 2°C, or even 1.5°C, is an appropriate

250 target to protect the well-being of young people and future generations, as modeling projections compared to these targets usually include only fast-feedback processes.  Indeed, Hansen et al (2008) concluded "If humanity wishes to preserve a planet similar to that on which civilization developed and to which life on Earth is adapted, paleoclimate evidence and ongoing climate change suggest that $CO_2$ will need to be reduced from its (then) current 385 ppm to at most 350

255 ppm, but likely less than that."  And further "If the present overshoot of the target $CO_2$ is not brief, there is a possibility of seeding irreversible catastrophic effects."

A danger of the 1.5°C and 2°C temperature targets is that they are far above the Holocene temperature range.  If such temperature levels are allowed to long exist they will spur "slow" amplifying feedbacks (Hansen et al 2013; Rohling et al 2013; Masson-Delmotte et al 2013),

260 which may have potential to run out of humanity's control.  The most threatening slow feedback likely is ice sheet melt and consequent sea level rise, but there are other risks in pushing the climate system far out of its Holocene range.  Methane release from melting permafrost and methane hydrates is also a potentially important feedback, for example, although there are large gaps in our understanding of this feedback including its time-scale (O'Connor et al 2011).

265 Thus in this paper we examine the fossil fuel emission reductions required to restore atmospheric $CO_2$ to 350 ppm or less, so as to keep global temperature close to the Holocene range, in addition to the canonical 1.5°C and 2°C targets.  Quantitative investigation requires consideration of Earth's energy imbalance, changing climate forcings, and climate sensitivity.

[Figure]

**Fig. 4.** Estimated effective climate forcings (update of Hansen et al 2005 through 2015). Forcings are based on actual changes of each gas, except $CH_4$-induced changes of $O_3$ and stratospheric $H_2O$ are included in the $CH_4$ forcing. Oscillatory and intermittent natural forcings (solar irradiance and volcanoes) are excluded. CFCs include not only chlorofluorocarbons, but all Montreal Protocol Trace Gases (MPTGs) and Other Trace Gases (OTGs).

**3. Global Climate Forcings and Earth's Energy Imbalance**

The dominant human-caused drivers (forcings) of climate change are changes of atmospheric GHGs and aerosols. GHGs absorb Earth's infrared (heat) radiation, thus serving as a "blanket" that warms Earth's surface. Aerosols, fine particles in the air that cause visible air pollution, both reflect and absorb solar radiation, but reflection of solar energy to space is their dominant effect, so they cause a cooling that partly offsets GHG warming. Estimated forcings (Fig. 4), an update of Fig. 28b of Hansen et al (2005), are similar to those of Myhre et al (2013) in the most recent IPCC report (IPCC 2013).

Climate forcings in Fig. 4 are the planetary energy imbalance caused by preindustrial-to-present change of each atmospheric constituent. The $CH_4$ forcing includes its indirect effects, as increasing atmospheric $CH_4$ causes tropospheric ozone ($O_3$) and stratospheric water vapor to increase (Myhre et al 2013). Uncertainties in the forcings, discussed by Myhre et al (2013), are typically 10-15% for GHGs. Uncertainty in the aerosol forcing, described by a probability distribution function (Boucher et al 2013), is of order 50%. Our estimate of aerosol + surface albedo forcing ($-1.2$ $W/m^2$) differs from the $-1.5$ $W/m^2$ of Hansen et al (2005), as discussed below, but both are within the range of the distribution function of Boucher et al (2013).

The positive net forcing (Fig. 4) causes Earth to be out of energy balance, with more energy coming in than going out, which drives slow global warming. Eventually Earth will become hot enough to radiate to space an amount of energy matching absorbed sunlight. However, because of the ocean's great thermal inertia (heat capacity), full atmosphere-ocean response to the forcing requires a long time: atmosphere-ocean models suggest that even after 100 years only 60-75% of the surface warming for a given forcing has occurred, the remaining 25-40% still being "in the pipeline" (Hansen et al 2011; Collins et al 2013). Moreover, we will outline in the next section that global warming can activate "slow" feedbacks, such as changes of ice sheets or melting of methane hydrates, so the time for the system to reach a fully equilibrated state is even longer.

GHGs have been increasing for more than a century and Earth has partially warmed in response. Earth's energy imbalance is the portion of the forcing that has not yet been responded to. This imbalance thus defines additional global warming that will occur without further change

[Figure]

**Fig. 5.** Ocean heat uptake in upper 2 km of ocean during 11 years 2005-2015 using analysis method of von Schuckmann and LeTraon (2011). Heat uptake in $W/m^2$ (0.5 and 0.7) refer to global (ocean + land) area, i.e., it is the contribution of the upper ocean to the heat uptake averaged over the entire planet.

of forcings. Earth's energy imbalance can be measured by monitoring ocean subsurface temperatures, because almost all excess energy coming into the planet goes into the ocean (von Schuckmann et al 2016). Most of the ocean's heat content change occurs in the upper 2000 m (Levitus et al 2012), which has been well measured since 2005 when the distribution of diving Argo floats achieved good global coverage (von Schuckmann and Le Traon 2011).

Earth's energy imbalance was about +0.6 $W/m^2$ during 2005-2010 (Hansen et al 2011) as inferred from heat gain in the upper 2 km of ocean using the von Schuckmann and Le Traon (2011) analysis and adding the smaller heat gain by the deep ocean (Purkey and Johnson 2013), continents, atmosphere, and net melting of sea ice and land ice. Accounting for the declining solar irradiance during 2005-2010 (Fig. A3), Hansen et al (2011) inferred that the energy imbalance with the solar cycle effect removed was +0.75 ± 0.25 $W/m^2$. Here we update the von Schuckmann and Le Traon analysis with data for 2005-2015 (Fig. 5) finding 0.7 $W/m^2$ heat uptake in the upper 2000 m of the ocean. The 11-year period now available (2005-2015) should practically eliminate a solar cycle influence. The small heat gains noted above add of order 0.1 $W/m^2$ to the global heat gain (Rhein et al 2013), so our current analysis is consistent with a planetary energy imbalance of +0.75 ± 0.25 $W/m^2$. The value 0.75 $W/m^2$ is near the middle of a range of estimates by several investigators (von Schuckmann et al 2016; Trenberth et al 2016).

**4. Climate Sensitivity, a Consistency Check and Slow Feedbacks**

Climate sensitivity has been a fundamental issue since at least the 19[th] century when Tyndall (1861) and Arrhenius (1896) stimulated interest in the effect of a $CO_2$ change on climate. Doubled atmospheric $CO_2$, a forcing of about 4 $W/m^2$, is now a standard forcing in studies of climate sensitivity. The Charney et al (1979) study concluded that equilibrium sensitivity, i.e., global warming after a time sufficient for the planet to restore energy balance with space, was 3°C ± 1.5°C for 2×$CO_2$ or 0.75°C per $W/m^2$ forcing. The central value found in a wide range of modern climate models (Flato et al 2013) and in empirical paleoclimate studies remains 3°C for 2×$CO_2$, but with an uncertainty that is still of order 1°C (Rohling et al 2012a).

An important consistency check is obtained by comparing the estimated net climate forcing (2.5 $W/m^2$, Fig. 4), Earth's energy imbalance (~0.75 $W/m^2$), observed global warming, and climate sensitivity. Observed warming since 1880-1920 is 1.06°C with the effect of El Niño/La

340 Niña oscillations removed (Fig. 2b).  Global warming between 1700-1800 and 1880-1920 was ~0.1°C (Abram et al 2016; Marcott et al 2013), so 1750-2015 warming was ~1.16°C.  Taking climate sensitivity as 0.75°C per $W/m^2$ forcing, global warming of 1.16°C implies that 1.55 $W/m^2$ of the total 2.5 $W/m^2$ forcing has been "used up" to cause observed warming.  Thus 0.95 $W/m^2$ forcing should remain to be responded to, i.e., the expected planetary energy imbalance is

345 0.95 $W/m^2$, reasonably consistent with the observed $0.75 \pm 0.25$ $W/m^2$.  If we instead use the aerosol + surface albedo forcing $-1.5$ $W/m^2$ estimated by Hansen et al (2005), the net climate forcing is 2.2 $W/m^2$ and the forcing not responded to is 0.65 $W/m^2$, which is also within the observational error of Earth's energy imbalance.

 An important matter to bear in mind is that the sensitivity 3°C for $2\times CO_2$ (0.75°C per $W/m^2$)

350 is the "fast-feedback" climate sensitivity, i.e., it does not include "slow" climate feedbacks that will occur if global temperature long remains above the Holocene level (Hansen et al 2008; Rohling et al 2012a).  Slow feedbacks include large-scale shrinking of ice sheets as Earth warms and the enhanced release of GHGs as the ocean, soil, and continental shelves warm.  These slow feedbacks are strongly amplifying, indeed, they are the reason that natural long-term climate

355 oscillations are so large in response to even small long-term global-average forcings (Rohling et al 2012b; Masson-Delmotte et al 2013).

 The fast-feedback climate sensitivity is the appropriate sensitivity to use in interpretation of recent climate change, because we use observed change of GHGs and because ice sheet change so far is small.  However, the need to avoid the emergence of slow feedbacks motivates the

360 criterion that energy balance should be restored at a global temperature close to and eventually within the Holocene range (Hansen et al. 2008, 2013).

 Earth's present energy imbalance is causing heat to accumulate in the ocean, where it contributes to melting of ice shelves (Rignot et al 2013).  Rising temperatures also increase the risk of $CO_2$ and $CH_4$ release from drying soils, thawing permafrost (Schadel et al 2016; Schuur et

365 al 2015) and warming continental shelves (Kvenvolden 1993, Judd et al 2002).  Time scales for the slow feedbacks are not well established, but recent modeling and empirical evidence suggest that substantial ice sheet and sea level changes could occur within periods as short as several decades (Rohling et al 2013; Pollard et al 2015; Hansen et al 2016).  If large planetary energy imbalance continues, there is a danger that the warming driving slow feedbacks will be so far

370 advanced that consequences such as large sea level rise proceed out of humanity's control.

 Quantification of requirements for stabilizing climate depends on knowledge of ongoing changes of the two largest GHG forcings, $CO_2$ and $CH_4$. It is also necessary to understand how we are changing GHG emissions directly through industrial and agricultural activities (designated 'anthropogenic emissions' and included in the SRES and RCP scenarios) and

375 indirectly through climate change (the slow feedbacks noted above, designated somewhat paradoxically 'natural emissions' changes, and not included in the SRES and RCP scenarios).

[Figure]

**Fig. 6.** (a) Global $CO_2$ annual growth based on NOAA data (http://www.esrl.noaa.gov/gmd/ccgg/trends/).
Dashed curve is for a single station (Mauna Loa). Red curve is monthly global mean relative to the same
month of prior year; black curve is 12-month running mean of red curve. (b) $CO_2$ growth rate is highly
correlated with global temperature, the $CO_2$ change lagging global temperature change by 8 months.

**5. Observed CO₂ and CH₄ Growth Rates**

Annual increase of atmospheric $CO_2$, averaged over a few years, grew from less than 1 ppm/year
50 years ago to more than 2 ppm/year today (Fig. 6), with the global mean and Mauna Loa $CO_2$
amounts now exceeding 400 ppm (Betts et al 2016). The large oscillations of the annual growth
are correlated with global temperature and with the El Niño/La Niña cycle[2]. Correlations are
calculated for the 12-month running means, which effectively remove the seasonal cycle and
monthly noise. Maxima of the $CO_2$ growth rate lag global temperature maxima by ~8 months
(Fig. 6b) and lag Niño3.4 [latitudes 5N-5S, longitudes 120-170W] temperature by ~10 months.
These lags imply that the current $CO_2$ growth spike (Fig. 6 uses data through July 2016),
associated with the 2015-16 El Niño, may not have reached its maximum yet, as Niño3.4 peaked
in December 2015 and global temperature peaked in February 2016.

[Figure]

**Fig. 7.** Fossil fuel $CO_2$ emissions (left scale) and airborne fraction, i.e., the ratio of observed atmospheric
$CO_2$ increase to fossil fuel $CO_2$ emissions.
* * *
[2] One mechanism for greater than normal atmospheric $CO_2$ growth during El Niños is the impoverishment of
nutrients in equatorial Pacific surface water and thus reduced biological productivity that result from reduced
upwelling of deep water (Chavez et al., 1999). However, the El Niño/La Niña cycle seems to have an even greater
impact on atmospheric $CO_2$ via the terrestrial carbon cycle through effects on the water cycle, temperature, and fire,
as discussed in a large body of literature (referenced, e.g., by Schwalm et al., 2011).

[Figure]

**Fig. 8.** Global CH$_4$ from Dlugokencky (2016), NOAA/ESRL (www.esrl.noaa.gov/gmd/ccgg/trends_ch4/). End months for the three indicated slopes are January 1984, May 1992, August 2006, and January 2016

Growth of airborne CO$_2$ (defined as the increase in atmospheric CO$_2$ above preindustrial levels) appears to be about half of fossil fuel emissions (Fig. 7), the remaining portion being net uptake by the ocean and biosphere (Ciais et al 2013). Here we use the Keeling et al. (1973) definition of airborne fraction, which is the ratio of quantities that are known with good accuracy: the annual increase of CO$_2$ in the atmosphere and the annual amount of CO$_2$ injected into the atmosphere by fossil fuel burning. The data reveal that, even as fossil fuel emissions have increased by a factor of four over the past half century, the ocean and biosphere have continued to take up about half of the emissions (Fig. 7, right-hand scale). This seemingly simple relation between emissions and atmospheric CO$_2$ growth is not predictive as it depends on the growth rate of emissions being maintained, and, indeed, it is not expected to continue in cases with major changes in the emission scenario, so we use a carbon cycle model in Section 7 to compute atmospheric CO$_2$ as a function of emission scenario.

Atmospheric CH$_4$ stopped growing between 1998 and 2006, indicating that its sources and sinks were nearly in balance, but growth resumed in the past decade (Fig. 8). Growth of CH$_4$ exceeds 10 ppb/year in 2014 and 2015, almost as fast as in the 1980s. Turner et al. (2016) suggest that increased fossil fuel emissions in the U.S. may be a major cause of renewed global CH$_4$ growth. However, CH$_4$ isotope data imply that resumed growth was mainly from wetlands, especially in the tropics but with a contribution from high latitudes of the Northern Hemisphere (Bousquet et al 2011; Dlugokencky et al 2011). The CH$_4$ changes over the past two decades are driven primarily by changes in emissions as observations of CH$_3$CCl$_3$ show very little change in the atmospheric sink for CH$_4$ (Montzka et al. 2011; Holmes et al. 2013). Future changes in the sink, however, are expected to lead to increased atmospheric CH$_4$ separate from emission changes, but these are difficult to project in the RCP scenarios (Voulgarakis et al. 2013).

The continued growth of atmospheric CO$_2$ and the reaccelerating growth of CH$_4$ raise important questions related to prospects of stabilizing climate. How consistent are scenarios for phasing down climate forcing with reality revealed by observational data? What changes to emissions are required to stabilize climate? We address these issues below.

[Figure]

**Fig. 9.** GHG climate forcing growth rate with historical data being 5-year running means, except data for 2014 and 2015 are 3- and 1-year means. (a) includes scenarios used in IPCC AR3 and AR4 reports, and (b) has AR5 scenarios. $N_2O$, MPTGs and OTGs (Montreal Protocol Trace Gases and Other Trace Gases) data are from NOAA/ESRL Global Monitoring Division.

**6. GHG Climate Forcing Growth Rates and Emission Scenarios**

Insight is obtained by comparing the growth rate of GHG climate forcing based on observed GHG amounts with past and present GHG scenarios. We examine forcings of IPCC SRES (2000) scenarios used in the AR3 and AR4 reports (Fig. 9a) and RCP scenarios (IPCC 2013) used in the AR5 report (Fig. 9b). We include the "alternative scenario" of Hansen et al (2000) in which $CO_2$ and $CH_4$ emissions decline such that global temperature stabilizes near the end of the century.[3] We use the same radiation equations for observed GHG amounts and scenarios, so errors in the radiation calculations do not alter the comparison. Equations for GHG forcings are from Table 1 of Hansen and Sato (2004) with the $CH_4$ forcing using an efficacy factor 1.4 to include effects of $CH_4$ on tropospheric $O_3$ and stratospheric $H_2O$ (Hansen et al 2005).

The growth of GHG climate forcing peaked at ~0.05 $W/m^2$/year (5 $W/m^2$/century) in 1978-1988, then falling to a level 10-25% below IPCC SRES (2000) scenarios during the first decade of the 21[st] century (Fig. 9a). The decline was due to (1) decline of the airborne fraction of $CO_2$ emissions (Fig. 7), (2) slowdown of $CH_4$ growth (Fig. 8), and (3) the Montreal Protocol, which initiated phase-out of gases that destroy stratospheric ozone.

The situation in 2000 seemed ripe for a pathway to climate stabilization more rapid than any of the IPCC scenarios. The slowing growth of forcings was partly good fortune, but also due to
* * *
[3]This scenario is discussed by Hansen and Sato (2004). $CH_4$ emissions decline moderately, producing a small negative forcing. $CO_2$ emissions (not captured and sequestered) are assumed to decline until in 2100 fossil fuel emissions just balance uptake of $CO_2$ by the ocean and biosphere. $CO_2$ emissions continue to decline after 2100.

[Figure]

**Fig. 10.** Fossil fuel emission scenarios. Scenarios in (a) have constant emissions in 2015-2020 and then simple specified rates of emission increase or decrease. IPCC (2013) RCP scenarios are shown in (b).

the prescience of the Montreal Protocol, whose design and implementation to reduce the ozone-depleting gases also allowed it to be used to slow or reverse growth of some other GHGs. The alternative scenario aimed to extend the downward trend in the growth rate of climate forcing by: (1) slowing the growth of $CO_2$ emissions, as may occur with a substantial rising price on carbon emissions to accelerate development of carbon-free energies, (2) a global effort to reduce $CH_4$ emissions, (3) continued use and tightening of the Montreal Protocol to constrain trace GHGs. The slowly decreasing forcing of this alternative scenario would have kept global warming well below 1.5°C, for climate sensitivity 0.75°C/W/m$^2$ (Fig. A4).

However, in reality, in the absence of a universally rising carbon price and substantial support for energy research and development, global fossil fuel $CO_2$ emissions accelerated, from 1.5%/year in 1973-2000 to ~2.5%/year after 2000 (Figs. 1 and S1). The growth rate of GHG forcing now exceeds the alternative scenario by ~70% (Fig. 9a). New scenarios must begin from current reality, and, as a consequence or recent growth, ambitious targets for limiting global warming now require much steeper emissions reductions.

The new IPCC (2013) RCP scenarios (Fig. 9b) initiate in 2011 and fan out into an array of potential futures driven by assumptions about energy demand, fossil fuel prices, and climate policy, chosen to be representative of an extensive literature on possible emissions trajectories (Moss et al 2010; van Vuuren et al 2011; Meinshausen et al 2011). Numbers on the RCP scenarios (8.5, 6.0, 4.5 and 2.6) refer to the GHG climate forcing (W/m$^2$) in 2100.

As a complement to RCP scenarios, we define scenarios simply by percent annual emission decrease or increase. We consider rates −6%/year, −3%/year, constant emissions, and +2%/year; emissions stop increasing in the +2%/year case when they reach 25 Gt/year (Fig. 10a). Scenarios with decreasing emissions are preceded by constant emissions for 2015-2020, in recognition that some time is required to achieve policy change and implementation. Note similarity of RCP 2.6 with −3%/year, RCP 4.5 with constant emissions, and RCP 8.5 with +2%/year (Fig. 10).

Scenario RCP2.6 has the world moving into negative growth of GHG forcing 25 years from now (Fig. 9b), through rapid reduction of GHG emissions and $CO_2$ capture and storage. Already in 2015 there is a huge gap between reality and RCP2.6. Closing the gap (0.01 W/m$^2$) between actual growth of GHG climate forcing in 2015 and RCP2.6 (Fig. 9b), with $CO_2$ alone, would require extraction from the air of more than 0.7 ppm of $CO_2$ or 1.5 GtC in the single year (2015). We discuss the plausibility and estimated costs of scenarios with $CO_2$ extraction in Section 9.

[Figure]

**Fig. 11.** (a) Atmospheric $CO_2$ for emission scenarios of Fig. 10a. (b) Atmospheric $CO_2$ including effect of $CO_2$ extraction that increases linearly after 2020 (after 2015 in +2%/year case). 1 ppm is ~2.12 GtC.

**7. Future $CO_2$ for Assumed Emission Scenarios**

We must model Earth's carbon cycle, including ocean uptake of carbon, deforestation, forest regrowth and carbon storage in the soil, for the purpose of simulating future atmospheric $CO_2$ as a function of fossil fuel emission scenario. Fortunately, the convenient dynamic-sink pulse-response function version of the well-tested Bern carbon cycle model (Joos et al 1996) does a good job of approximating more detailed models, and it produces a good match to observed industrial-era atmospheric $CO_2$. Thus we use this relatively simple model, described elsewhere (Joos et al 1996; Kharecha and Hansen 2008 and references therein), to examine the effect of alternative fossil fuel use scenarios on the growth or decline of atmospheric $CO_2$. For land use $CO_2$ emissions in the historical period, we use the values labeled Houghton/2 by Hansen et al (2008), which were shown in the latter publication to yield good agreement with observed $CO_2$. We use fossil fuel $CO_2$ emissions data for 1850-2013 from Boden et al (2016). BP fuel consumption data for 2013-2015 is used with the fractional annual changes for each nation to allow extension of the Boden analysis through 2015. Emissions were almost flat from 2014 to 2015, due to economic slowdown and increased use of low-carbon energies, but, even if a peak in global emissions is near, substantial decline of emissions is dependent on acceleration in the transformation of energy production and use (Jackson et al 2016)..

The scenarios shown in Figs. 10a and 11a are the baseline cases without any anthropogenic $CO_2$ removal. We illustrate five cases with $CO_2$ removal in Fig. 11b that achieve atmospheric $CO_2$ targets of either 350 ppm or 450 ppm in 2100, with cumulative removal amounts listed in parentheses. The rate of $CO_2$ extraction in all cases increases linearly from zero in 2010 to the value in 2100 that achieves the atmospheric $CO_2$ target (350 ppm or 450 ppm). The amount of $CO_2$ that must be extracted from the system exceeds the difference between the atmospheric amount without extraction and the target amount, e.g., constant $CO_2$ emissions and no extraction yields 546 ppm for atmospheric $CO_2$ in 2100, but to achieve a target of 350 ppm the required extraction is 328 ppm, not 546 – 350 = 196 ppm. The well-known reason (Cao and Caldeira 2010) is that ocean out gassing increases, and vegetation productivity and ocean $CO_2$ uptake decrease with decreasing atmospheric $CO_2$, as explored in a wide range of Earth System models (Jones et al 2016).

**8. Simulations of Global Temperature Change**

Analysis of future climate change and policy options to alter that change must address various
uncertainties. One useful way to treat uncertainty is to use results of many models and construct
probability distributions (Collins et al 2013). Such distributions have been used to estimate the
remaining budget for fossil fuel emissions for a specified likelihood of staying under a given
global warming limit and to compare alternative policies for limiting climate forcing and global
warming (Rogelj et al 2016a,b).

Our aim here is a fundamental, transparent calculation that clarifies how future warming
depends on the rate of fossil fuel emissions. We use best estimates for fundamental uncertain
quantities such as climate sensitivity. If these estimates are accurate, actual temperature should
have about equal chances of falling higher or lower than the calculated value. Among the
important uncertainties in projections of future climate forcings and climate change are climate
sensitivity, the effects of ocean mixing and dynamics on the climate response function discussed
below, and aerosol climate forcing. We provide all defining data so that others can easily repeat
calculations with alternative choices.

We calculate global temperature change T at time t in response to any climate forcing
scenario using the Green's function (Hansen 2008)

$$T(t) \ = \ \int R(t) \, [dF/dt] \, dt \tag{1}$$

where R(t) is the product of equilibrium global climate sensitivity and the dimensionless climate
response function (percent of equilibrium response), dF/dt is the annual increment of net forcing,
and the integration begins before human-made climate forcing is substantial. Our response
function reaches 75% response in 100 years, a rate that Hansen et al (2011) conclude is
representative of the real world, based on observations of Earth's energy imbalance; this
imbalance is an immediate consequence of the time required for the ocean surface temperature to
respond to changing climate forcing. Our results can be exactly reproduced, or altered with
alternative choices for climate forcings, climate sensitivity and response function, as we tabulate
the forcings in Table S1 and the response function is exactly defined.[4]

We use equilibrium fast-feedback climate sensitivity ¾ °C per W/m$^2$ (3°C for 2×CO$_2$). This
is consistent with current climate models (Collins et al 2013: Flato et al 2013) and paleoclimate
evidence (Rohling et al 2012a; Masson-Delmotte et al 2013; Bindoff and Stott 2013).

CO$_2$ is the dominant forcing in scenarios for future climate. The growth of non-CO$_2$ GHG
climate forcing is likely to be even smaller, relative to CO$_2$ forcing, than it has been in recent
decades (Fig. 9), especially if there is a strong effort to limit climate change. Indeed, recent
agreement to use the Montreal Protocol (2016) to phase down emissions of minor trace gases
should cause added forcing of Montreal Protocol Trace Gases (MPTGs) + Other Trace Gases
(OTGs) (red region in Fig. 9) to become near zero or slightly negative, thus at least partially off-
setting growth of the N$_2$O climate forcing. Some N$_2$O increase may be inevitable, because its
emissions are largely associated with food production, and population is not expected to stabilize
before mid-century at the earliest (Ciais et al 2013; Kroeze and Bouwman 2011).
* * *
[4]We use the "intermediate" response function in Fig. 5 of Hansen et al. (2011), which gives best agreement with
Earth's energy imbalance. Fractional response is 0.15, 0.55, 0.75 and 1 at years 1, 10, 100 and 2000 with these
values connected linearly in log (year), cf. Fig. 5 of Hansen et al (2011).

[Figure]

**Fig. 12.** Climate forcings used in our climate simulations; Fe is effective forcing, as discussed in connection with Fig. 4. (a) Future GHG forcing uses four alternative fossil fuel emission growth rates. (b) GHG forcings are altered based on $CO_2$ extractions of Fig. 11.

The net effect of nitrogen emissions is complex because of both diminishing and amplifying feedbacks (Kroeze and Bouwman 2011), e.g., fertilizers can increase uptake of carbon by the biosphere and affect tropospheric $O_3$, but it is expected that more efficient use of fertilizers can reduce emissions and $N_2O$ growth (Liu and Zhang 2011). $CH_4$ is responsible for the largest non-$CO_2$ GHG forcing, with potential to significantly exacerbate or alleviate the magnitude of global warming, so we address the range of $CH_4$ possibilities in Section 11. Here we use RCP6.0 for the non-$CO_2$ GHGs, a scenario in which warming by these gases, compared to $CO_2$, is small.

We take tropospheric aerosol plus surface albedo forcing as $-1.2$ W/m$^2$ in 2015, presuming the aerosol and albedo contributions to be $-1$ W/m$^2$ and $-0.2$ W/m$^2$, respectively. We assume a small increase this century as global population rises and increasing aerosol emission controls in emerging economies tend to be offset by increasing development elsewhere, so aerosol + surface forcing is $-1.5$ W/m$^2$ in 2100. The temporal shape of the historic aerosol forcing curve (Table S1) is from Hansen et al (2011), which in turn was based on the Novakov et al (2003) analysis of how aerosol emissions have changed with technology change.

Historic stratospheric aerosol data (Table S1, annual version), an update of Sato et al (1993), include moderate 21$^{st}$ century aerosol amounts (Bourassa et al 2012). Future aerosols, for realistic variability, include three volcanic eruptions in the rest of this century with properties of the historic Agung, El Chichon and Pinatubo eruptions, and a background stratospheric aerosol forcing $-0.1$ W/m$^2$. This leads to mean stratospheric aerosol climate forcing $-0.25$ W/m$^2$ for the 21$^{st}$ century, similar to the prior century. Reconstruction of historical solar forcing (Coddington et al 2015; Kopp et al 2016), based on data in Fig. A3, is extended with an 11-year cycle.

[Figure]

590

**Fig. 13.** Simulated global temperature for forcings of Fig. 12. Observations as in Fig. 2. Gray area is 2σ (95% confidence) range for centennially-smoothed Holocene maximum, but there is further uncertainty about the magnitude of the Holocene maximum, as noted in the text and discussed by Liu et al (2014).

595       Individual and net climate forcings for the several fossil fuel emission reduction rates are shown in Fig. 12a,c. Scenarios with linearly growing $CO_2$ extraction at rates required to yield 350 or 450 ppm airborne $CO_2$ in 2100 are in Fig. 12b,d. These forcings and the assumed climate response function define expected global temperature for the entire industrial era (Fig. 13).
      A stark summary of alternative futures emerges from Fig. 13. If emissions grow 2%/ year,

600 modestly slower than the 2.6%/year growth of 2000-2015, warming reaches ~3°C by 2100. Warming is close to 2°C if emissions are constant until 2100. Furthermore, both scenarios launch Earth onto a course of more dramatic change well beyond the initial 2-3°C global warming, because: (1) warming continues beyond 2100 as the planet is still far from equilibrium with the climate forcing, and (2) warming of 2-3°C would unleash strong slow feedbacks,

605 including melting of ice sheets and increases of GHGs, thus continuing growing climate change. Reducing global emissions at a rate of 3%/year (or more steeply) maintains global warming at less than 1.5°C above preindustrial, but the temperature at the end of the century continues to be 0.5°C or more above the prior Holocene maximum with consequences that are difficult to foresee, especially due to the likelihood of initiating substantial amplifying slow feedbacks.

610       Desire to avoid slow feedbacks, including ice sheet shrinkage and sea level rise, spurs the need to get global temperature back into the Holocene range. This goal needs to be achieved on the time scale of a century or less, as paleoclimate evidence indicates that the response time of sea level to climate change is 1-4 centuries (Grant et al 2012, 2014) for natural climate change, and it is unlikely that the response would be slower to a stronger, more rapid human-made

615 climate forcing. The scenarios that reduce $CO_2$ to 350 ppm succeed in getting temperature back close to the Holocene maximum by 2100 (Fig. 13b), but they require extractions of atmospheric $CO_2$ that range from 72 ppm in the scenario with 6%/year emission reductions to 768 ppm in the scenario with +2%/year emission growth.
      Scenarios ranging from constant emissions to +2%/year emissions growth can be made to

620 yield 450 ppm in 2100 via extraction of 160-600 ppm of $CO_2$ from the atmosphere (Fig. 12b). However, these scenarios still yield warming more than 1.5°C above the preindustrial level (more than 1°C above the early Holocene maximum). Consequences of such warming and the plausibility of extracting such huge amounts of atmospheric $CO_2$ are considered below.

**9. CO₂ Extraction: Plausibility and Cost**

625 The above calculations show the need for extraction of $CO_2$ from the air, also called negative emissions, in addition to reducing emissions of GHGs. A goal of 100 GtC (47 ppm $CO_2$) extraction in the 21st century was chosen by Hansen et al (2013), because it is comparable to net emissions from historic deforestation and land use (Ciais et al 2013), and thus it is likely to be about as much as can be achieved via relatively natural reforestation and afforestation (Canadell

630 and Raupach 2008) and improved agricultural practices that increase soil carbon (Smith 2016).

We differentiate between the limited carbon that can be extracted from the air by improved agricultural and forestry practices and additional "technological extraction" by intensive negative emission technologies that might be used to remediate overshoot of the $CO_2$ level needed to assure an acceptable long-term climate state. We assume that improved practices will aim at

635 optimizing agricultural and forest carbon uptake via relatively natural approaches, compatible with delivering a range of ecosystem services from the land (Smith 2016; Smith et al 2016) In contrast, proposed technological extraction and storage of $CO_2$ does not have co-benefits and remains unproven at relevant scales (NRC 2015). Improved practices have local benefits in agricultural yields and forest products and services (Smith et al 2016), which may help minimize

640 net costs. Developed countries recognize a financial obligation to less developed countries that have done little to cause climate change (Paris Agreement 2015)[5]. We suggest that at least part of developed country support should be channeled through an agricultural and forestry program, with continual evaluation and adjustment to reward and encourage progress (Bustamante et al 2014). Non-$CO_2$ GHGs could be included in the improved practices program. We do not

645 estimate the program cost, but we assume that such a program will be carried out, if there is to be hope of stabilizing climate. Thus the costs we estimate for additional technological extraction of $CO_2$ are a minimum cost.

Here we first reexamine the question of whether a concerted global effort on carbon storage in forests and soil might have potential to provide a carbon sink substantially larger than 100 GtC

650 this century. Smith et al. (2016) estimate that reforestation and afforestation together have carbon storage potential of about 1.1 GtC/year. However, as forests mature, their uptake of atmospheric carbon decreases (termed "sink saturation"), thereby limiting $CO_2$ drawdown. Taking 50 years as the average time for tropical, temperate and boreal trees to experience sink saturation yields 55 GtC as the potential storage in forests this century.

655 Smith (2016) shows that soil carbon sequestration and soil amendment with biochar compare favorably with other negative emission technologies with less impact on land use, water use, nutrients, surface albedo, and energy requirements, but understanding of and literature on biochar are limited (NRC 2015). Smith concludes that soil carbon sequestration has potential to store 0.7 GtC/year. However, as with carbon storage in forest, there is a saturation effect. A

660 commonly used 20-year saturation time (IPCC 2006) would yield storage of 14 GtC soil carbon storage, while an optimistic 50-year saturation time would yield 35 GtC. Use of biochar to improve soil fertility provides additional carbon storage with potential rate as high as 0.7-1.8 GtC/year (Woolf et al 2010; Smith 2016). Larger industrial-scale biochar carbon storage is
* * *
[5] Another conceivable source of financial support for $CO_2$ drawdown might be legal settlements with fossil fuel companies, analogous to penalties that courts have imposed on tobacco companies, but with the funds directed to the international "improved practices" program.

conceivable, but belongs in the category of intensive negative emission technologies, discussed
below, whose environmental impacts and costs require scrutiny. We conclude that 100 GtC is an
appropriate estimate for potential carbon extraction via an ambitious concerted global-scale
effort to improve agricultural and forestry practices with carbon drawdown as a prime objective.

Copious $CO_2$ extraction is conceivable via other intensive negative emission technologies,
including (1) burning of biofuels in power plants with capture and sequestration of resulting $CO_2$
(Creutzig et al 2015), and (2) direct air capture of $CO_2$ and sequestration (Keith 2009; NRC
2015), and (3) grinding and spreading of minerals such as olivine to enhance the geological
weathering process (Taylor et al 2016). However, energy, land and water requirements of these
technologies impose economic and biophysical limits on $CO_2$ extraction (Smith et al 2016).

The popular concept of bioenergy with carbon capture and storage (BECCS) requires large
areas, high fertilizer and water use, and may compete with other vital land use such as agriculture
(Smith, 2016). Costs estimates are ~$150-350/tC for crop-based BECCS (Smith et al 2016).

Direct air capture has less area and water needs than BECCS and no fertilizer requirement,
but it has high energy use, has not been demonstrated at scale, and cost estimates exceed those of
BECCS (Socolow et al 2011; Smith et al 2016). Keith et al (2006) have argued that, with strong
research and development support and industrial-scale pilot projects sustained over decades, it
may be possible to achieve costs ~$200/tC, thus comparable to BECCS costs; however other
assessments are higher, reaching $1400-3700/tC (NRC 2015). Carbon capture and storage
(CCS) from a stream of nearly 100 percent $CO_2$ at fossil fuel burning sites is more efficient and
thus less expensive than direct air capture, but CCS at power plants is properly included in our
scenarios as one of the mechanisms competing to achieve phase-down of fossil fuel emissions,
along with energy efficiency, renewable energies, and nuclear power.

Enhanced weathering via soil amendment with crushed silicate rock is a candidate negative
emission technology that also limits coastal ocean acidification as chemical products liberated by
weathering increase land-ocean alkalinity flux (Kohler et al 2010; Taylor et al 2016). If two-
thirds of global croplands were amended with basalt dust, as much as 2-5 GtC/year might be
extracted, depending on application rate (Taylor et al 2016), but energy costs from mining,
grinding and spreading likely reduce this by 10-25% (Moosdorf et al 2014). Although such
large-scale enhanced weathering is speculative, there are potential co-benefits for temperate and
tropical agroecosystems that could affect its practicality, and may put some enhanced weathering
into the category of improved agricultural and forestry practices. Benefits include fertilizing of
crops that increases yield and reduces use and cost of other fertilizers, increasing crop protection
from insect herbivores and pathogens thus decreasing pesticide use and cost, neutralizing soil
acidification to improve yield, and suppression of GHG ($N_2O$ and $CO_2$) emissions from soils
(Edwards et al 2016; Kantola et al 2016). Cost of enhanced weathering might be reduced by
deployment with reforestation and afforestation and with crops used for BECCS.

For cost estimates, we first consider restoration of airborne $CO_2$ to 350 ppm in 2100 (Fig.
11b), which would keep global warming below 1.5°C and bring global temperature back close to
the Holocene maximum by end-of-century (Fig. 13b). This scenario keeps the temperature
excursion above the Holocene level small enough and brief enough that it has the best chance of
avoiding ice sheet instabilities and multi-meter sea level rise (Hansen et al 2016). If fossil fuel
emission phasedown of 6%/year had begun in 2013, as proposed by Hansen et al (2013), this

scenario would have been achieved via the 100 GtC carbon extraction from improved agricultural and forestry practices.

Now, with assumption that global emissions will be comparable to today's level through 2020, Figs. 11b and 13b show that 6%/year emissions reduction starting in 2021 leaves a requirement to extract 72 ppm $CO_2$ (153 GtC) from the air during this century. Emission reductions of 3%/year leave a requirement of extracting 112 ppm $CO_2$ (Fig. 13b) by 2100. Constant emissions and +2%/year emissions growth would require extractions of 328 and 768 ppm $CO_2$ to reach 350 ppm in 2100.

The lowest cost is for the case of 6%/year emissions reduction. We assume that 100 GtC will be stored in the biosphere via improved agricultural and forestry practices. We do not mean to diminish the magnitude or cost of this task, but we must assume that it will occur if climate change impacts are to be minimized, and further we expect that developed countries will recognize their obligations to provide assistance required to achieve success.

The remaining 53 GtC, at the rate $150-350/tC estimated for BECCS and other intensive negative emission technologies (Fig. 3f of Smith et al 2016), would cost $8-18.5 trillion, thus $100-230 billion per year if spread uniformly over 80 years. In contrast, continued high emissions, say between constant emissions and +2%/year, require extraction of 695-1628 GtC, which corresponds to $104-570 trillion dollars or $1.3-7 trillion dollars per year over 80 years.[6] Such extraordinary cost, along with the land area, fertilizer and water requirements (Smith et al 2016) suggest that, rather than the world being able to buy its way out of climate change, continued high emissions may force humanity to largely live with the climatic consequences.

**10. Climate Forcing Contribution of Non-CO₂ GHGs**

GHG climate forcing is surging, not declining, the annual rate having increased more than 20% in just the past five years (Fig. A5). This recent surge in the growth rate of the GHG climate forcing is led by increasing growth of $CH_4$, but $CO_2$ is by far the largest cause of continued growth of the GHG climate forcing (Fig. 9). Given the difficulty and cost of reducing $CO_2$, we must ask about the alternative of reducing non-$CO_2$ GHGs. Could realistic reductions of these other gases substantially alter the $CO_2$ abundance required to meet a target climate forcing?

Methane ($CH_4$) is the largest climate forcing other than $CO_2$ (Fig. 3). The $CH_4$ atmospheric /lifetime is only about 10 years (Prather et al 2012), so there is potential to reduce this climate forcing rapidly if $CH_4$ sources are reduced. Our climate simulations, employing RCP6.0 for non-$CO_2$ gases, make an optimistic assumption that future $CH_4$ , after a moderate increase in the next few decades, will decrease from its present ~1800 ppb to 1650 ppb in 2100, yielding a forcing $-0.1$ W/m$^2$. RCP2.6 makes a more optimistic assumption: that $CH_4$ will decline monotonically to 1250 ppb in 2100, yielding a forcing $-0.3$ W/m$^2$ (relative to today's 1800 ppb $CH_4$), based on radiation equations identified in section 6.
* * *
[6] For reference, the United Nations global peacekeeping budget is about $10B/year. National military budgets are larger: the 2015 USA military budget was $596B and the global military budget was $1.77 trillion (SIPRI 2016).

[Figure]

**Fig. 14.** Comparison of observed CH4 and N2O amounts and RCP scenarios. RCP 6.0 and 4.5 scenarios for N2O overlap. Observations are from NOAA/ESRL Global Monitoring Division.

Actual atmospheric CH4 abundance (Fig. 14) is diverging on the high side from these optimistic scenarios. The downward offset (~20 ppb) of CH4 scenarios relative to observations (Fig. 14) is due to the fact that RCP scenarios did not include a data adjustment that was made in 2005 to match a revised CH4 standard scale (E. Dlugokencky, priv comm). In addition, observed CH4 is increasing more rapidly than in most scenarios.

Carbon isotopes provide a valuable constraint on which CH4 sources[7] contribute to the CH4 growth resurgence in the past decade (Fig. 7). Specifically, Schaefer et al (2016) conclude that the growth was primarily biogenic, thus not fossil fuel, and located outside the tropics, most likely ruminants and rice agriculture. Such an increasing biogenic source is consistent with effects of increasing population and dietary changes (Tilman and Clark 2014). These sources potentially could be mitigated [by changing rice growing methods (Epule et al 2011) and inoculating ruminants (Eckard et al, 2010; Beil 2015)], but that would require widespread adoption of new technologies at the farmer level. Concerning fossil fuels, it is feasible to reduce CH4 leaks, yet enhanced shale gas extraction of CH4 may yield even greater leakage (Caulton et al., 2014; Petron et al., 2014; Howarth, 2015).

Slow climate feedbacks could increase CH4 because natural emissions from methane hydrates, permafrost and natural wetlands[8] are expected to increase in response to global warming (O'Connor et al 2010). On the other hand, as yet there is little evidence for substantial emissions from hydrates or permafrost (Warwick et al., 2016). Predicting such emissions has large uncertainty, because drought conditions eliminate wetland CH4 emissions (while greatly increasing CO2 release from soil carbon), and, in addition, CH4 created in anoxic zones is mostly oxidized in the water column before reaching the atmosphere (Reeburgh, 2007).

All reasonable effort to reduce methane is appropriate, recognizing that its mitigation effort will be different than that for fossil fuel CO2 in that it should include a focus on agriculture in developing countries with adoption of new practices and technology at the farm level. However,
* * *
[7] Estimated human-caused CH4 sources (Ciais et al., 2013) are: fossil fuels (29%), biomass/biofuels (11%), Waste and landfill (23%), ruminants (27%) and rice (11%)

[8] Wetlands compose a majority of natural CH4 emissions and are estimated to be equivalent to about 36% of the anthropogenic source (Ciais et al., 2013)

given increasing global population and global warming "in the pipeline," there is an underlying tendency for greater emissions. The current $CH_4$ increases (Fig. 14a) show that the mitigation paths envisaged with RCP scenarios projecting methane decreases are not close to present reality. Nevertheless, it is plausible that human-caused emissions could achieve a pathway to a moderate reduction of $CH_4$ forcing in 2100, but it seems unlikely that the reduction could be larger than of the order of 0.1 W m$^{-2}$.

There is less leverage with $N_2O$, whose growth is exceeding all scenarios (Fig. 14b). Major quantitative gaps remain in our understanding of the nitrogen cycle (Kroeze and Bouwman 2011), but fertilizers are clearly a principal cause of $N_2O$ growth (Röckmann and Levin 2005; Park et al 2012). More efficient use of fertilizers could reduce $N_2O$ emissions, but considering the scale of global agriculture, and the fact that fixed N is an inherent part of feeding people, there will be pressure for continued emissions at least comparable to present emissions. In contrast, agricultural $CH_4$ emissions are inadvertent and not core to food production. Given the current imbalance [emissions exceeding atmospheric losses by about 30% (Prather et al., 2012)] and the long $N_2O$ atmospheric lifetime (116 ± 9 years; Prather et al 2015) it is nearly inevitable that $N_2O$ will continue to increase this century, even if emissions growth is checked. There can be no expectation of an $N_2O$ decline that offsets the need to reduce $CO_2$.

The Montreal Protocol has been a success in stifling and even reversing the growth of trace gases that can destroy ozone and cause global warming (Prather et al 1996; Newman et al 2009). Amendments to this protocol to achieve phasedown of additional gases are important, but mainly for the objective of limiting the growth of these trace gas climate forcings rather than with an expectation of obtaining a large net reduction of climate forcing by MPTGs + OTGs (Fig. 3).

**11. Discussion**

We conclude that the world has already overshot targets for atmospheric temperature and greenhouse gas amount required to maintain a safe long-term environment for humanity and assure the well-being of young people and future generations. Earth's paleoclimate history tells us that, if we wish to avoid locking in multi-meter sea level rise with loss of functionality of most coastal cities (Clark et al 2016), our target should be to keep global temperature close to the Holocene range, which requires an absolute reduction in current GHG climate forcing and global temperature.

Thus we infer an urgent need for both (1) rapid phasedown of fossil fuel emissions, and (2) actions that draw down atmospheric $CO_2$ and, at minimum, eliminate net growth of non-$CO_2$ climate forcings. These tasks are formidable and are not now being pursued effectively.

Although economic and political analysis is outside the scope of this paper, our conclusion that the world has already overshot appropriate targets is sufficiently grim to compel us to point out that pathways minimizing climate impacts are feasible and have other benefits. The underlying policy required to spur rapid reduction of fossil fuel emissions is a transparent steadily rising carbon fee that makes fossil fuels include their costs to society (Ackerman and Stanton 2012; Hsu 2011; Hansen 2014), which encourages energy conservation (reduced consumption), energy efficiency, and technology development of carbon-free energy. A rising global carbon fee, which could be achieved by agreement of a few major powers (Hsu 2011), is the crucial underlying policy needed to spur private investment, innovations and consumer

choices, but it does not obviate the need for government energy planning, energy efficiency and pollution regulations, and support for energy research and development.

Governments have shown the ability to achieve high rates of emissions reduction, e.g., Peters et al (2013) note that Belgium, France and Sweden achieved emission reductions of 4-5%/year sustained over 10 or more years in response to the oil crisis of 1973. These rates were primarily a result of nuclear power build programs, which historically has been the fastest route to carbon-free energy (Fig. 2 of Cao et al 2016). Peters et al also note that a continuous shift to natural gas led to sustained reductions of 1-2%/year in the UK in the 1970s and in the 2000s, 2%/year in Denmark in 1990-2000s, and 1.4%/year in the USA since 2005. None of these examples were aided by the broad economy-wide effect of a rising carbon fee, although high oil prices in the 1970s partially simulated that effect. What is needed to achieve rates presently demanded by the climate crisis is a combination of a rising carbon fee along with government support of technological advances, which has historically received the smallest share of total research budgets in OECD countries.

Our scenarios show that, in addition to $CO_2$ emission phase-out, there must be large $CO_2$ extraction from the air and a net halt of growth of non-$CO_2$ GHG climate forcings. Success with both $CO_2$ extraction and non-$CO_2$ GHG controls requires a major role for developing countries. Ancillary benefits of the agricultural and forestry practices needed to achieve $CO_2$ drawdown, such as improved soil fertility, advanced agricultural practices, forest products, and species preservation, are of interest to all nations. Developed nations have a recognized obligation to assist nations that have done little to cause climate change yet suffer some of the largest climate impacts. If economic assistance is made partially dependent on verifiable success in carbon drawdown and non-$CO_2$ mitigation, this will provide incentives that maximize success in carbon storage. Similar considerations apply to incentives for reducing trace gas emissions, and, as we have discussed, some activities such as soil amendments that enhance weathering might be designed to support both $CO_2$ and other GHG drawdown.

Considering our conclusion that the world has overshot the appropriate target for global temperature, and the difficulty and perhaps implausibility of negative emissions scenarios, we would be remiss if we did not point out the potential contribution of demand-side mitigation that can be achieved by individual actions as well as by government policies. Numerous studies (e.g. Hedenhus et al 2014; Popp et al 2010) have shown that reduced ruminant meat and dairy products is needed to reduce GHG emissions from agriculture, even if technological improvements increase food yields per unit farmland. Such climate-beneficial dietary shifts have also been linked to co-benefits that include improved sustainability and public health (Bajzelj et al 2014; Tilman and Clark 2014). Similarly, Working Group 3 of IPCC (2014) finds "robust evidence and high agreement" that demand-side measures in the agriculture and land use sectors, especially diet shifts, reduced food waste and changes in wood consumption have substantial mitigation potential, but they remain under-researched and poorly quantified.

If rapid emission reductions are initiated soon, it is still possible that at least a large fraction of required $CO_2$ extraction can be achieved via relatively natural agricultural and forestry practices with other benefits. On the other hand, if large fossil fuel emissions are allowed to continue, the scale and cost of industrial $CO_2$ extraction, occurring in conjunction with a deteriorating climate with growing economic effects, may become unmanageable. Simply put, the burden placed on young people and future generations may become too heavy to bear.

**Appendix A: Additional figures and tables**

[Figure]

**Fig. A1.** $CO_2$ emissions from fossil fuel use and cement manufacture, based on data of Boden et al (2016) through 2013, with results extended using BP(2016) energy consumption data. (a) is log scale and (b) is linear. Growth rates r in (a) for an n year interval from $(1+r)^n$ with end-year amount the mean for three years to minimize noise.

**A1. Fossil Fuel CO₂ Emissions**

$CO_2$ emissions from fossil fuels in 2015, based on preliminary data from BP (2016), were only slightly higher than in 2014 (Fig. A1). Such slowdowns are common, and usually reflect the global economic situation. Given rising global population and the fact that many nations, including the soon-to-be-most-populous India, are still at early stages of development, the potential exists for continued growth of emissions. Fundamental changes in energy technology will be needed if the world is to rapidly change energy course and phase down fossil fuel emissions.

**A2. Temperature Data and Analysis Method**

We use the current Goddard Institute for Space Studies global temperature analysis (GISTEMP), which is the analysis method described by Hansen et al. [2010] but with updated input data. The analysis combines data from three sources: (1) monthly mean meteorological station data of the Global Historical Climatology Network (GHCN) described by Peterson and Vose [1997] and Menne et al. [2012], (2) monthly mean data from Antarctic research stations of the Scientific Committee on Antarctic Research (SCAR), as reported by the SCAR Reference Antarctic Data for Environmental Research project (http://www.antarctica.ac.uk/met/READER), and (3) ocean surface temperature measurements from the NOAA Extended Reconstructed Sea Surface Temperature (ERSST) [Smith et al., 2008; Huang et al., 2015].

[Figure]

**Fig. A2a.** Global surface temperature (12-month running mean) relative to 1951-1980 in the GISTEMP analysis, comparing the current analysis using NOAA ERSST.v4 for sea surface temperature with results using the prior ERSST.v3b.

[revised manuscript text omitted]

**A5. Growth Rate of Total GHG Climate Forcing**

965 In the past several years the growth rate of climate forcing by GHGs has accelerated sharply, in contrast to most scenarios, which presumed that the GHG climate forcing would be declining. Ozone is not well-mixed, so its changes are not well-measured and are not fully accounted for in Fig.5. However, the effective $CH_4$ forcing, which is included, includes about half of the tropospheric $O_3$ change.

Table A1. Effective Forcing (W/m2) Relative to 1850 except Volcanic Aerosols

| Year | $CO_2$ | $CH_4$* | CFCs | $N_2O$ | $O_3$$ | TA+SA | Volcano | Solar | Net |
|------|--------|---------|------|--------|--------|-------|---------|-------|-----|
| 1850 | 0.000 | 0.000 | 0.000 | 0.000 | 0.000 | 0.000 | −0.083 | 0.000 | −0.083 |
| 1860 | 0.024 | 0.012 | 0.000 | 0.004 | 0.004 | −0.029 | −0.106 | 0.032 | −0.059 |
| 1870 | 0.048 | 0.025 | 0.000 | 0.007 | 0.009 | −0.058 | −0.014 | 0.048 | 0.065 |
| 1880 | 0.109 | 0.039 | 0.000 | 0.010 | 0.014 | −0.097 | −0.026 | −0.049 | −0.001 |
| 1890 | 0.179 | 0.054 | 0.000 | 0.013 | 0.018 | −0.146 | −0.900 | −0.070 | −0.850 |
| 1900 | 0.204 | 0.073 | 0.001 | 0.016 | 0.023 | −0.195 | −0.040 | −0.063 | 0.018 |
| 1910 | 0.287 | 0.109 | 0.002 | 0.020 | 0.026 | −0.250 | −0.072 | −0.043 | 0.079 |
| 1920 | 0.348 | 0.150 | 0.003 | 0.027 | 0.032 | −0.307 | −0.215 | −0.016 | 0.022 |
| 1930 | 0.425 | 0.194 | 0.004 | 0.035 | 0.036 | −0.364 | −0.143 | 0.014 | 0.200 |
| 1940 | 0.494 | 0.232 | 0.005 | 0.041 | 0.045 | −0.424 | −0.073 | 0.037 | 0.356 |
| 1950 | 0.495 | 0.274 | 0.009 | 0.049 | 0.056 | −0.484 | −0.066 | 0.055 | 0.387 |
| 1960 | 0.599 | 0.342 | 0.027 | 0.057 | 0.078 | −0.621 | −0.106 | 0.102 | 0.478 |
| 1970 | 0.748 | 0.433 | 0.076 | 0.071 | 0.097 | −0.742 | −0.381 | 0.093 | 0.395 |
| 1980 | 0.976 | 0.532 | 0.185 | 0.091 | 0.115 | −0.907 | −0.108 | 0.169 | 1.054 |
| 1990 | 1.227 | 0.618 | 0.303 | 0.118 | 0.117 | −0.997 | −0.141 | 0.154 | 1.399 |
| 2000 | 1.464 | 0.651 | 0.347 | 0.141 | 0.117 | −1.084 | −0.048 | 0.173 | 1.761 |
| 2005 | 1.619 | 0.651 | 0.356 | 0.153 | 0.123 | −1.125 | −0.079 | 0.019 | 1.716 |
| 2010 | 1.766 | 0.665 | 0.364 | 0.167 | 0.129 | −1.163 | −0.082 | 0.028 | 1.874 |
| 2015 | 1.927 | 0.684 | 0.373 | 0.183 | 0.129 | −1.199 | −0.100 | 0.137 | 2.134 |

**$CH_4$:$CH_4$-induced changes of tropospheric $O_3$ and stratospheric $H_2O$ are included.**
$$O_3$ half of tropospheric $O_3$ forcing + stratospheric $O_3$ forcing from IPCC (2013)
Annual data are available in a longer version of the table available at
http://www.columbia.edu/~mhs119/Burden/ .

Table A2. Effective Forcing (W/m2) Relative to 1850 except Volcanic Aerosols

| Year | $CO_2$ | #$CH_4$ | CFCs | $N_2O$ | $$O_3$ | TA+SA | Volcano | Solar | Net |
|------|--------|---------|------|--------|--------|-------|---------|-------|-----|
| 2016 | 1.942 | 0.654 | 0.367 | 0.180 | 0.130 | −1.207 | −0.100 | 0.097 | 2.062 |
| 2020 | 2.074 | 0.658 | 0.373 | 0.189 | 0.130 | −1.234 | −0.100 | −0.008 | 2.082 |
| 2030 | 2.347 | 0.663 | 0.343 | 0.212 | 0.130 | −1.296 | −1.057 | −0.008 | 1.335 |
| 2040 | 2.580 | 0.688 | 0.301 | 0.238 | 0.123 | −1.350 | −0.100 | 0.027 | 2.507 |
| 2050 | 2.803 | 0.717 | 0.267 | 0.271 | 0.117 | −1.396 | −0.100 | 0.062 | 2.741 |
| 2060 | 3.017 | 0.740 | 0.243 | 0.302 | 0.111 | −1.433 | −1.208 | 0.097 | 1.870 |
| 2070 | 3.222 | 0.753 | 0.229 | 0.337 | 0.105 | −1.462 | −0.100 | 0.132 | 3.215 |
| 2080 | 3.421 | 0.741 | 0.215 | 0.367 | 0.098 | −1.484 | −0.100 | 0.167 | 3.425 |
| 2090 | 3.614 | 0.676 | 0.199 | 0.401 | 0.091 | −1.495 | −1.240 | 0.167 | 2.413 |
| 2100 | 3.801 | 0.580 | 0.191 | 0.428 | 0.085 | −1.500 | −0.100 | 0.167 | 3.652 |

**$CH_4$: $CH_4$-induced changes of tropospheric $O_3$ and stratospheric $H_2O$ are included**
$$O_3$: Half of tropospheric $O_3$ forcing + stratospheric $O_3$ forcing from IPCC 2013
Annual data are available in a longer version of the table available at
http://www.columbia.edu/~mhs119/Burden/ .

**Acknowledgments**

Support of the Climate Science, Awareness and Solutions program has been provided by the Durst family, the Grantham Foundation for Protection of the Environment, Jim and Krisann Miller, Gary Russell, Gerry Lenfest, the Flora Family Foundation, Elisabeth Mannschott, Alexander Totic and Hugh Perrine, which is gratefully acknowledged.  DJB acknowledges funding through a Leverhulme Trust Research Centre Award (RC-2015-029).  EJR acknowledges support from Australian Laureate Fellowship FL12 0100050

We appreciate the generosity, with data and advice, of Tom Boden, Ed Dlugokencky and Steve Montzka.

The authors declare that they have no conflicts of interest.  The first author (JH) notes that he is a plaintiff in the lawsuit Juliana et al vs United States (2016).

---

## Referee Comment (RC1) · Anonymous Referee #1 · 14 Oct 2016

I like this paper, and I support the authors' decision to act as advocates for strong action on slowing climate change. Mention of the lawsuit, Juliana et al. vs United States, is not critical to the scientific issues discussed, but it frames the scientific discussion. Hansen et al. first assess how much climate change is acceptable before dangerous human influence activates slow climate feed backs that will significantly raise sea level. Their choice is subjective. Based on paleoclimate evidence, they determine a target of 350 ppm for CO2 by end of the 21st century (where, depending on changes in emissions of other other non-CO2 GHGs, this target will be adjusted up or down), which restores Earth's energy balance and keeps climate in the range of the past 10,000 years. This target is stricter than other suggestions (e.g., the <1.5 C agreed on at the Paris Conference of Parties meeting), but their argument for it seems reasonable to me. They conclude that, to achieve this goal and avoid an enormous future financial burden,

emissions of CO2 from fossil fuel combustion must be decreased by 6%/yr starting in 2021. Even at this relatively large rate of CO2 emission reduction, additional CO2 must be removed through changes in agriculture and forestry practices to enhance uptake by the terrestrial biosphere. Other scenarios for reducing CO2 emissions are considered, and the costs of not acting by 2020 would be quite large. Much of the paper builds on previous work by Hansen and co-authors. Its novelty is in showing the dramatic reductions to CO2 emissions necessary to keep climate change by 2100 within the bounds of Holocene climate. The methods used are scientifically justifiable and transparent enough that others who disagree with the paper's threshold for climate change or emission-reduction scenarios can explore other approaches. Most of the paper is well-written and clear. I recommend the paper for publication in Earth System Dynamics with minor revisions.

General comments: 1. There are many errors with the references including missing punctuation, citations from text missing in the reference list, and errors in citations. 2. Large uncertainties associated with some climate-related parameters used (e.g., climate sensitivity and current and future aerosol forcing) are not accounted for in the analysis. 3. Do the authors see a role for geoengineering in stabilizing climate within the range of the Holocene?

Specific comments: L32-33: I worry about the susceptibility of carbon stored in the terrestrial biosphere to human interference, e.g., from biomass burning. L59: I suggest making explicit here the role of non-CO2 GHGs in the target atmospheric CO2 abundance of 350 ppm. L173-174: It seems the 12-month running mean must end 6 months before August, 2016 (which I assume is the last month of actual T anomaly used), so the last part of the curve must be the actual T anomaly. L262-264: CO2 released from melting permafrost ecosystems is also potentially important and would offset other practices to enhance uptake by the terrestrial biosphere. L270: How does someone reconcile the RFs in Fig. 4 with those in Table A1? Are the differences between the figure and the table for CO2 and CH4 attributed to the increase in RF from 1750 to 1850?

[Figure]

Why not make both 1750 to present? L283: RF for CO2 here is about 10% greater than in AR5; although still potentially within IPCC's uncertainties, calling them "similar" is too vague. L415-425: Some of the discussion on atmospheric CH4 in section 10 seems more appropriate here on observed CH4 growth rate and reasons behind the changes. In addition to Schaefer et al. (mentioned in section 10), two new papers are appropriate. Nisbet et al. ((2016), Rising atmospheric methane: 2007–2014 growth and isotopic shift, Global Biogeochem. Cycles, 30, doi:10.1002/2016GB005406), which reaches a similar conclusion to Schaefer et al. as to the role of microbial sources in driving the increase in atmospheric CH4 since 2007, but with more emphasis on tropical wetlands than anthropogenic agricultural sources, and Schwietzke et al. ((2016), Upward revision of global fossil fuel methane emissions based on isotopic database, Nature, vol. 538, doi:10.1038/nature19797), which suggests that emissions of CH4 from fossil fuel sources are significantly larger than inventories indicate, but that there is no significant trend in emissions. All 3 studies (Schaefer, Nisbet, and Schwietzke) make use of isotopic constraints. L417: Turner et al. is contradicted by other studies with stronger observed constraints, so why include it? L420: The contributions of high northern latitudes to the increase since 2007 was only in 2007; various inverse model studies indicate climatological emissions (or less) in the years following 2007. L624: Section 9 is least clear and seems incomplete. While I recognize that a full economic analysis is beyond the domain of this study, selective discussion of CO2 extraction costs without comparison with other important costs (e.g., cost of BAU, converting to renewable sources of energy, etc.) makes the section seem incomplete. Perhaps a summary of pertinent costs in a table would help. L668-676: Recent life-cycle analysis (e.g., DeCicco et al., Climatic Change (2016) 138: 667. doi:10.1007/s10584-016-1764-4) suggest that liquid biofuels result in greater GHG emissions than using petroleum. How do these results figure into this discussion? L715-719: How does this cost compare in relative terms with others? Why assume it will happen? L753-762: Comments regarding L415-425 could apply here too. L822-823: A shift from what to natural gas? L994: As mentioned earlier, values for CO2 and CH4 are not in agreement with what

is plotted in Fig. 4.

Edits: L190: exaggerated L470: as a consequence of recent growth L504: remove extra "." L636: "." needed after "Smith et al., 2016)" L709: suggest "Now, assuming global...." L737: delete "/" L900: was in ERSST...

---

## Short Comment (SC4) · 18 Oct 2016

**Expanded Comment: Young People's Burden: Requirement of Negative CO₂ Emissions**

**Michael Beenstock**

**Hebrew University of Jerusalem**

**Introduction**

The central point made by the authors is that in the absence of a rapid phasedown in carbon emissions our children and grandchildren will face a crippling burden of carbon extraction. This argument rests on two claims. The first is that sustainable global temperature is its Holocene average, which according to the authors was reached in 1985. If global temperature continues to exceed this benchmark, irreversible feedbacks may be seeded with catastrophic consequences. The second is that the authors have used the correct model to calculate the young people's burden.

Their benchmark for sustainable temperature needs further justification. Also, they do not carry out validation tests of their model, which meet contemporary scientific standards. Furthermore, their analysis of intergenerational justice, which is one of their keywords, is incomplete and ignores the extensive literature on this subject, including the Stern Review (Stern 2007), which refers specifically to the issue discussed by the authors.

**The Hiatus in Global Warming?**

The authors write (p 5), "One effect of recent warming is to remove unequivocally the illusion of a global warming hiatus after 1997-8 El Ninð." This strong claim is based on the fact that in 2015 global temperature was fractionally higher than in 2011, and that by August 2016 it was 0.1° C higher (Fig 2). The data for 2016 are incomplete, and Figure 2 even suggests a large increase in global temperature in 2017. The data for 2015 do not show "unequivocally" that the hiatus has ended, or even there was no hiatus to begin with. Even if the data for 2016 remain at their current level, a single year's data does not justify the claim that the hiatus has unequivocally ended, or never even existed. Matters would be different if by 2020 global temperature was about 0.7° C higher than today according to the projections of IPCC.

The fact that some claim (p 5) that, "…temporary plateaus are consistent with expected long-term warming due to increasing atmospheric GHGs" when others do not, is not a sufficient argument that all is well with the anthropogenic theory of global warming upon which the authors completely rely. The authors' pronouncements regarding the end of the hiatus is premature to say the least.

Drawing major conclusions from short-term changes in global temperature has characterized climate science during the last 50 years. In the 1970s climate scientists pronounced the onset of New Ice Age following the cooling that took place in the 1960s (figure 2). President Nixon was persuaded by climate scientists to set-up a special committee to study the problem, but by the end of the 1970s the increase in global temperature eliminated the scare of a New Ice Age. Between the mid 1970s and mid 1990s global temperature increased once more by $0.5°$ C. The scare of a New Ice Age was rapidly replaced by the scare of anthropogenic global warming. However, as Figure 2 shows, between the mid 1990s and mid 2010s global temperature stabilized despite the acceleration in GHG forcing noted by the authors (Fig 6 – 8). By 2015 global temperature should have increased by about another $0.7°$ C according to IPCC projections, but all the major climate change models over-predicted global temperature (Beenstock, Reingewertz and Paldor 2016). Policy makers understandably question whether in ten years' time the latest climatic scare won't be replaced by another.

**The Holocene Benchmark**

The authors assume that the benchmark for sustainable global temperature should be the Holocene average. There are two issues here. Whereas global temperature has been measured directly since 1880, global temperature during the Holocene is measured indirectly. The second is why the Holocene average rather than some other benchmark?

Presumably the bars in Figure 3(b) represent confidence intervals, which as the authors recognize are small ($0.25°$ C). Since the Holocene benchmark is a crucial parameter in their analysis, they need to explain in depth how direct measures of temperature for 1985 can be compared with reconstructed temperatures during the last 18,000 years. Common sense suggests that reconstructions from thousands of years ago must be highly speculative. The controversy over the Hockey Stick Theory shows that even what happened only a 1000 years ago is subject

to widespread disagreement. So how reliable can the claim be that current global temperature is the hottest it has ever been in the last 10,000 years or so?

Suppose, for argument's sake, that there is no measurement error in global temperature during the Holocene. Why should global temperature during the Holocene be relevant for establishing sustainable temperatures in the 21$^{st}$ and 22$^{nd}$ centuries? The authors need to answer this question first. Why should the average serve as the right benchmark? Why not the maximum, which occurred about 7000 years ago? Since the issue is sustainability, the authors need to show why global temperature was not sustainable 7000 years ago. The authors do not provide adequate justification for their benchmark. They simply remark (p7), "A danger of the 1.5$^{o}$ C and 2$^{o}$ C temperature targets is that they are far above the Holocene temperature range. If such temperature levels are allowed to long exist they will spur "slow" amplifying feedbacks, (references) which may have potential to run out of humanity's control." In this context, they mention ice sheet melt, sea level rise and methane release, all of which are speculative. The truth is that we do not know. It is perfectly legitimate to speculate about "unknown unknowns" as do the authors, but it is quite a different matter to base strong policy proposals on such speculations.

The authors are not alone in making the implicit assumption that the unknown unknowns must always be adverse. History is replete with resolutions to unknown unknowns, which have benefited mankind, and with the discovery of solutions to what appeared to be social and economic time-bombs. For example, in 1866 the renowned British economist W.S. Jevons predicted that by 1900 the Industrial Revolution would come to an end as the world ran out of coal. In the 1970s Club of Rome scientists (Meadows et al 1972) predicted that the world was running out of natural resources, which among other considerations prompted the UN initiative on the establishment of a New International Economic Order. In 1975 the UN Conference on International Cooperation was launched to deal with these issues.

**Model Validation**

Beenstock, Reingewertz and Paldor (2012. 2016) observe that climatologists do not, on the whole, use contemporary statistical methods to analyse their data.and to validate their models empirically. They rely instead on validation methods, which were vitiated by statistical theory in the 1970s and 1980.  Specifically, global temperature, GHG forcings, solar irradiance, and other variables that are hypothesized to be related, are nonstationary. Variables are nonstationary when

their sample moments, such as means and variances, depend on time. For example, because global temperature has been increasing, its sample mean must increase. The same applies to GHGs, which have been increasing over time. It has been known since Yule (1897) that nonstationary time series may be spuriously correlated. Spurious correlation arises when independent time series happen to be correlated simply because they depend on time. It is well known that spurious correlations may even exceed 0.95 despite the fact that the variables involved are completely unrelated.

In Beenstock et al (2012) we explained that the methodological solution to the spurious regression problem was discovered by Granger and Engle (1987) for which they were awarded the Nobel Prize in Economics in 2004. We noted there that whereas global temperature and solar irradiance are difference stationary (their changes are stationary), anthopogenic forcings such as GHGs and aerosols are not. In fact, they are stationary in second differences (changes in changes). This phenomenon greatly complicates testing the anthropogenic theory of global warming. Our main result was that the partial correlation between global temperature and GHGs is a spurious regression phenomenon. On the other hand, we found that global temperature depends on solar irradiance and the change in GHGs rather than their levels. The latter result is, however, inconsistent with the anthropogenic theory of global warming because it implies that an increase in GHG concentrations has only a temporary effect on global temperature; it does not affect global temperature in the long-term. Alternatively, it means that to reduce global temperature, the growth rate in GHGs must be negative, i.e.it is insufficient to lower the growth in GHGs and carbon extraction must be permanent and on-going.

In Figure 6(b) the authors show that since 1960 the growth in $CO_2$ is correlated (after 8 months) 0.51 with the level of global temperature. This result is consistent with the one reported in the previous paragraph. However, the authors fail to appreciate that it is inconsistent with the anthropogenic theory of global warming, which hypothesizes a relationship between the level of global temperature and the level of $CO_2$ instead of its rate of growth. Nor are they worried by the fact that their result does not apply at longer or shorter lag orders than 8 months. Cherry-picking the largest correlation, as they do, does not establish what they wish to claim, especially when some of the correlations are negative. Moreover, these simple correlations ignore third variables, such as other GHGs, aerosols and solar irradiance with which $CO_s$ forcings are

correlated. The effect of these third variables might be intermediated by $CO_2$ forcings. Finally, a correlation of 0.51 means that only 25 percent of the variance in global temperature is associated with $CO_2$.

The authors need to use multivariate statistical methods, in which global temperature is related to $CO_2$ as well as other GHG forcings and solar irradiance. The analysis in Figure 6(b) does not meet contemporary statistical standards as represented e.g. by Estrada, Perron and Martinez-Lopez (2013) and the numerous references in Beenstock et al (2012). A related methodological criticism applies to Figure A4 in which the authors report the historic tracking of their model with respect to global temperature during 1880 – 2000. The model appears to track well, but on closer inspection there are some problems. The model systematically over-predicts during 1880 – 1905 after which it under-predicts until the late 1940s. Since 1960 the model appears to track better, but is difficult to see from Figure A4. Note that the model solutions are hindcasts rather than forecasts; the authors calibrated their model to track the past, so the fit should be good. In Beenstock, Reingewertz and Paldor (2016) we propose a methodology for testing historic tracking of outcomes, such as global temperature, that are nonstationary. This methodology tests whether hindcasts are merely spuriously correlated with the data. We applied this methodology to 22 climate change models used by IPCC. All 22 models turned out to be spuriously correlated with the data, despite the fact that the correlations varied between 0.96 and 0.98. To persuade readers that their model is not merely spuriously correlated with the past, the authors need to establish that their hindcasts of global temperature are genuinely correlated with actual global temperature. This is crucial for policy makers because climate models, which fail to track the past can hardly be relied upon to predict the future.

The authors have overlooked important methodological developments in the statistical analysis of nonstationary time series data, such as climate data, despite the fact that they were introduced into climate science almost 20 years ago (Stern and Kaufmann 1997). The authors are not alone in this.

**Pascal's Wager and Intergenerational Justice**

Pascal reasoned that man should act as if God exists because if He does exist, man will spend an eternity in heaven rather than in hell. Pascal's wager has been use by climatologists too. Even if you have doubts about the anthropogenic theory of global warming, act as if you believe in it

because if you don't, future generations might end up in hell on earth contending with carbon extraction costs, according to the authors, of up to $570 trillion.

Modern history shows that younger generations tend to be better-off than their parents and grandparents thanks to scientific progress in medicine, technology and economic growth. The young people, to which the authors refer, will be much better-off than us even according to the more pessimistic projections of the economic effects of climate change reported in the Stern Review (Stern 2007). Just as we are more resilient than our parents and grandparents, so future generations will be more resilient than us. Even a carbon clean-up cost of $570 trillion will be but a fraction of world GDP, especially when this burden is annuitized.

This is not to belittle the issue, but simply to place it in its correct economic proportions, and in the context of intergenerational equity with which the authors are concerned. The authors of the Stern Review (Stern 2007) grappled with this problem because they understood that it might be easier for future generations to cope with clean-up costs bequeathed to them, than it is for the current generation to prevent them. The central issue in this context is the determination of the intergenerational discount rate, which compares monetary values today in terms of monetary values in the distant future e.g. 2116. The intra-generational discount rate takes into account two factors, which translates future monetary values into current monetary values. Because of rising living standards over the life-cycle an individual prefers a dollar today to a dollar when he or she is older. But, even if living standards do not rise, the same individual might prefer today's dollar to one in the future because human beings are impatient; they have a positive rate of time preference. Also, the bird in the hand is worth more than two in the bush; a certain dollar today is worth more than an uncertain dollar in the future.

If, e.g. the discount rate is conservatively set at 3 percent per year, a dollar in 30 years' time is worth 41 cents today. A dollar in 100 years' time is worth only 5.2 cents today. In the Stern Review it was suggested that the intergenerational discount rate should be smaller than its intra-generational counterpart because interpersonal comparisons of time preference between generations are invidious. This controversial suggestion reduced the intergenerational discount rate in the longer term to about 2 percent at which a dollar in 100 years' time would be worth 13.8 cents today. This would mean that the cost of carbon abatement to the current generation must be less than 14 percent of the mitigated burden to our progeny in 100 years' time, if

intergenerational justice requires the current generation to undertake the carbon abatement polices proposed by the authors.

The authors' analysis of intergenerational justice is seriously lacking. Just because a future burden happens to be large does not necessarily mean that the current generation must undertake sacrifices in the name of intergenerational justice to prevent it. If, in addition, the future burden is uncertain e.g. because the authors' model does not represent the truth, but the cost of carbon abatement is more certain, this reasoning applies a fortiori.

**Conclusion**

The main suggestions to the authors are summarized:

1. Qualify the claim regarding the end of the hiatus in global temperature.
2. Provide further justification for the claim that the average temperature during the Holocene serves as a benchmark for sustainability.
3. Provide empirical evidence that the historic simulations of their model regarding global temperature are not spuriously correlated with actual global temperature.
4. Improve the discussion of intergenerational justice by integrating the intergenerational discount rate into the analysis.

**References**

Beenstock M, Reingewertz Y, Paldor N (2012) Polynomial cointegration tests of anthropogenic impacts on global warming. *Earth System Dynamics*, 3, 173-188.

Beenstock M, Reingewertz Y, Paldor N (2016) Testing the historic tracking of climate models. *International Journal of Forecasting*, 32, 1234-1246.

Engle RF, Granger CWR (1987) Cointegration and error correction: representation, estimation and testing. *Econometrica*, 64, 813-836.

Estrada F, Perron P, Martinez-Lopez B (2013) Statistically derived contributions of diverse human influences to twentieth century temperature changes. *Nature Geoscience*, 6, 1050-1055.

Kaufmann A. Stern DI (1997) Evidence for human influence on climate from hemispheric temperature relations. *Nature*, 388, 39-44.

Meadows DH, Meadows DL, Randers J, Behrens WW (1972) *The Limits to Growth*, Universal, New York.

Stern N (2007) *The Economics of Climate Change*. Cambridge University Press, Cambridge and New York.

Yule U. (1897) On the theory of correlation. *Journal of the Royal Statistical Society (A)*, 89, 1-69.

---

## Referee Comment (RC2) · Anonymous Referee #2 · 19 Oct 2016

This paper is an update to Hansen's evolving narrative about anthropogenic climate change and implications, which in recent years has been put forth mostly in grey literature. It is certainly an unusual paper, with its enormous scope, from detailing observational records to evaluating remedial actions. In effect, it is attempting a synthesis with same breadth as all three working groups of the IPCC but from a personal perspective. Is this useful? Is this acceptable practice? I think the answer to the first question is clearly yes. This paper will make a useful entry point for appreciating the full scope of the problem, including providing mechanistic insight. Certainly the IPCC reports or their summaries don't succeed well in this task. The answer to the second question is "maybe", but only if the paper is reviewed across its full scope. As a scientist, I can review the science aspects. But I'm somewhat concerned that my review – from this perspective alone – might lend credence to conclusions about policy which may not be

sound. To help address the need for broad review, I've taken the liberty of also sending the draft to a colleague with somewhat different expertise than my own, although still from a scientific perspective. I suggest that further review from a policy perspective may be warranted. I've appended this colleague's review to the end of mine, below.

General points: (1) An endpoint of year 2100 is taken for future projections. I guess this is still standard practice, but limits the ability to make key points, such as the extent to which some scenarios commit to further warming "in the pipeline". I would suggest that an endpoint of 2150 or 2200 would better frame the discussions. Too late to change? I'd urge that this be given some thought.

(2) The topic of slow feedbacks crops up at various times in the paper, each time as a bit of an aside. I sense that the issue of the further risk associated with flow feedbacks needs some space in the introductory sections of this paper.

Specific points:

36: "the current generation" Strange term for a science paper. Might be conflated with electricity generation.

160-162: This claim needs support. Might be better to cut because the digression to support the claim would be distracting and the point is anyway of secondary importance.

175-179: The 1998 El Nino set a record that was only barely broken for the next 18 years until this most recent El Nino event in 2015/2016. It seems perfectly reasonable to assume current El Nino will set a record that will last similarly long. If so, then the linear projection assumed here will almost certainly overestimate warming over the next few decades. For balance, a bracketing lower bound fit is needed. The simplest alternative might be to include data only through 2014 when making the linear fit.

193-198. This sentence is too long and clumsy. Also, it sets up an irrelevant straw man involving the claim that the current year might be warmest of the entire Holocene.

**ESDD**
198-206: The point of this sentence does not come through for me. In particular "these smoothed temperatures are relevant to important climate features" Which temperatures? Relevant mechanistically or statistically?

207: "because of issues such as discussed" It would be better to spell out briefly what these issues are.

193-212: Underlying this discussion is the need to compare recent decadal temperatures with century-scale averages for the Holocene. To do this rigorously requires a measure of recent warming that has 100-year significance. The need for this has been glossed over in this discussion. Creating such a measure should be doable, e.g. by allowing uncertainty for expected natural decadal variability. At least the need should be laid out to motivate future work with appropriate caveats offered.

233: 0.7C is damn cold! Must be an anomaly, but relative to what?

248. There's too much rounding in this estimate of "not much more than 1 C". Should be computed by combining mean and sigma estimates to 0.1C precision for Eemian relative to Holocene and Holocene relative to 1880-1920.

251: This point appears to be adequately backed by the preceding discussion without reference to the fast vs slow feedback question. Better to make it this way first, and then only add the slow/fast issue as an additional layer, e.g. to be discussed further below.

339-340: Unless I missed it, this is the first mention of a temperature record with El Nino/La Nina removed. Nothing about this is mentioned in the caption to Figure 2, and by eye the ENSO events appear not to have been filtered in the data shown. Needs clarifying.

352-353. Change in vegetation cover should also be included in the laundry list of slow feedbacks.

354-356: This states a plausible theory as fact. Needs rewording, e.g. "they are a
favored explanation for why.."

359-261: Not sure this point about slow feedbacks comes across clearly. I sense this is a framing concept that might better have been developed earlier in the text. See my general comment above.

470: typo: "or"

Figure 9: The y axis introduces the symbol Delta Fe, which has not yet been defined.

510-521 and Figure 11. This is hard to follow and has some inconsistencies. It would help immensely to show the extracted amounts as time series in Figure 10 in Pg/yr or equivalent. Note the community now uses the cgs unit Pg instead of GtC. The text says the extraction starts in 2010 but the caption states 2020. The extracted amount would be better expressed in Pg (or GtC) instead of ppm to avoid seeding unnecessary confusion as to why the extracted amount exceeds the change in atmospheric CO2. It's more useful anyway to express captured amounts in the same units as emissions. Fig. 11 caption needs to clarify what year is used for reporting the cumulative amount captured.

541, Eq. (1). There is a notational problem with Eq. (1). The t on the right-hand side cannot be same as the t on the left. The integration needs to be over t', and the limits of integration need to be spelled out. I'd expect kernel to be a function of t' or of t-t', depending on how t' is defined. It should not be a function of t. Figure 1 Why doesn't the long-term average stratospheric aerosol forcing center on zero? The treatment appears to take the natural stratospheric aerosol background as a forcing, which is hard to reconcile with the definition of a forcing as a perturbation from a natural strate.

617-618 and Section 9. Again, it would be preferable to express the extracted amounts consistently in Pg C rather than ppm.

818. Better to break the sentence before the "e.g." and remove the "e.g.".
Separate review by colleague:

1) The reviewed article by Hansen and colleagues provides a broad and coherent synthesis of observed and predicted warming of the climate system that is highly relevant to current policy targets and discussions. The paper illustrates (and frames) very well the reality and severity of required CO2 emissions reductions and atmospheric CO2 extractions required to maintain a "safe" climate. Though lengthy, the paper is clearly written and is thus accessible to a broad audience.

2) A major criticism is that the paper relies heavily on global surface mean temperature as a benchmark against which past, current and future climate states are compared or deemed safe or "dangerous". One weakness with this approach is that the paper and its methodology does not address (enough) the uncertainties surrounding temperature changes, whether in the paleo proxies interpretation of Eemian and Holocene climates (error bars in Figure 3b are unsatisfactory), or in future predictions from the simple green function model used therein. A possible way to remedy this, in the context of paleo proxies for instance, is to add various timeseries from different studies overlaying the mean for the Holocene era.

3) On a related note, the paper argument to reduce emissions below certain thresholds (e.g. 350-450ppm or 0.5-1°C) rests on using the holocene range as a benchmark for safe climate. The comparison to the Holocene's temperature variations is well described. The authors, for instance, recognize differences in smoothing (line 195) between the centennial window and interannual/decadal window of modern warming and their implications for the climate system. However, it is not convincing that an additional 0.5°C to 1°C increase from the Holocene range (e.g Paris target of 1.5-2°C by 2100) will lead to catastrophic consequences without detailed analysis of climate models's predictions, which this paper does not address.

4) The authors frequently reference paleo temperature changes to infer climate sensitivity and to warn against catastrophic climate change. The authors, for instance, use
the Eemian as an analog for a warmer world, its feedbacks, and associated sea level changes. While interglacial periods and deglaciations can be used to understand the climate system and its slower feedbacks, I am not convinced of their use as analogs for modern and future warming or to infer climate sensitivity. First, the forcing is different: one involve short-wave radiative forcing changes (precession, obliquity, eccentricity), whereas the other involves changes in atmospheric GHG composition and consequent changes in long-wave radiative forcing. The type of forcing and governing timescales may thus involve different sets and contributions of feedbacks (fast or slow), with potentially unique time and forcing dependent climate feedback parameters for natural vs anthropogenic perturbations (e.g. for short term natural variability, Xie et al 2016). I thus find the use of the Eemian as an analog for future warming incomplete without a discussion of the limitations of using such comparisons.

5) The choice of 1970 as the starting year to determine the secular warming trend (0.18°C/decade) in Figure 2b is not well justified in the paper, besides from referring to it as "the present global warming rate". The authors elsewhere refer to the longterm warming of 1°C since 1880-1920 (e.g. Lines 172, 220), but specifically chose 1970 to interpolate warming values for 2040 and 2060 (Line 177). In addition to contributions to the hiatus of the 2000's, Meehl et al (2016) show potential contributions of decadal variability during the warming phase as well. While the authors recognize the effects of decadal variability in driving the hiatus in surface warming (Line 170), they do not do the same for the enhanced warming period. Since the warming response is likely non-linear, using a linear trend since 1900 is probably not appropriate, and perhaps the authors may want to use the IPCC model mean instead for assessing future values.

6) The discussion of the hiatus is very brief and lacks in substance, especially given the paper's focus on global mean surface temperature as a proxy for a changing climate. Furthermore, more precise wording should be used when discussing the hiatus, e.g. in lines 163-167, where the term hiatus in global warming is used vs global surface warming. The recent El Niño warming, while shows a nice visual of an increase in

**ESDD**
surface warming, is, in my opinion, not an accurate way to reject the "hiatus" argument. Rather, the continued increase in ocean heat uptake during the hiatus period (Roemmich et al 2015) is much more powerful, and is a scientifically accurate argument to show continued energy imbalance in the climate system. Discussing the ocean heat content trend in the hiatus context can prevent future communication blunders, e.g. if a future decrease in surface mean temperature ensues due to decadal variability or other poorly known aspects of climate.

7) The conclusion that the world has already overshot a "safe" climate target can not be taken without great reservations given the paper's limited analysis of temperature uncertainties and the relation between global mean temperature and the climate system. Though likely true, this statement seems quite subjective and also likely model dependent. From a communication standpoint, it is a bleak conclusion that may lead to no action rather than the authors's likely intention of motivating urgent changes.

Minor Clarifications, recommendations, and typos:

1) Figure 4: Radiative forcing plot should have uncertainty error bars, as described in paper (line 289-290).

2) Typo in line 190: "ezaggerated"

3) Typo in line 232: there is an extra "("

4) Line 315: Since the paper is serving as an overview, other estimates of energy imbalance/OHC changes should be listed, including those from cited and possibly also other non-cited references.

5) Line 345: The math is somewhat confusing to me without an equation. Math doesn't seem to add up when using the aerosol forcing, or I am not understanding well, hence the need to clarify the equation used, which I assume is the energy balance equation  $C^{T}/dt=F+$  lambdaT

6) Line 370: The ice sheets "slow" feedback is brought up several times and is a major

**ESDD**
motivation for limiting warming beyond 2100. There are for instance several references to ice sheet feedbacks that can drive sea level rise out of "humanity's control", with little description of these feedbacks. However, little is detailed about this feedback (mechanisms, magnitude, etc.). It would serve the paper well to have a more detailed paragraph on the ice sheets feedback.

7) Line 374: SRES is not defined.

8) Line 412: The authors suggest the carbon airborne fraction is not expected to continue with larger growth rate emission scenarios. Is there a reference or detailed basis behind this statement.

1. Figure 9: The contribution from CFCs seems quite high, and the drop is (too?) large in  $\Delta$ F, matching across all gases (CFCs, CO2 CH4 and N2O) which is perhaps too coincidental?

2. Line 483: The authors suggest RCP2.5 requires negative Forcing growth rate 25 years from now, which is confusing. RCP2.5 requires negative emissions not until  $\sim$ 2075. A negative growth rate is required almost immediately. I am not sure how the 25 years came about in the text and in Figure 9b.

3. Line 642: The authors should look into the UNFCCC's REDD+ program, and how their proposed carbon uptake rates compare to the REDD+ efforts and proposed/future plans.

4. Carbon removal section: Since atmospheric carbon extractions is a major feature of this paper, the presentation could benefit from a summary table or a "wedge" figure (similar to Pacala and Socolow 2005) to summarize CO2 extractions technologies and methods, including costs and feasibility. Current reading through dense text is quite burdensome.

5. Line 695: Long list of proposed benefits but no mention of possible negative impacts of basal dust use.

**ESDD**
6. Line 775: The conclusion that future reductions in methane are unlikely is not justified given the speculative nature of the observed recent increase in methane.

7. Line 785 (& Line 570): N2O source from the ocean is not well known either and its future change due to deoxygenation or changes in ventilation rates is poorly known and may have additional or canceling effects on land emissions and feedbacks (Martinez-Rey et al 2015).

8. Figure 12a: There is little difference in temperature outcomes between the 6% and 3% reduction scenarios, which is interesting but barely discussed in the paper. There seems to be an effective rate of emission reduction for large temperature reductions that is not well exploited in the discussion.

References:

Martinez-Rey, J., Bopp, L., Gehlen, M., Tagliabue, A., and Gruber, N.: Projections of oceanic N2O emissions in the 21st century using the IPSL Earth system model, Biogeosciences, 12, 4133-4148, doi:10.5194/bg-12-4133-2015, 2015.

Meehl, G.A., A. Hu, B. D. Santer, and S.-P. Xie, 2016, Interdecadal Pacific Oscillation contributions to twentieth-century global surface temperature trends, Nature Climate Change, published online Aug. 29, 2016, doi:10.1038.

Roemmich, D. et al. Unabated planetary warming and its ocean structure since 2006. Nature Clim. Change 5, 240–245 (2015).

Xie, S.-P., Y. Kosaka, and Y. M. Okumura (2015), Distinct energy budgets for anthropogenic and natural changes during global warming hiatus, Nat. Geosci., doi:10.1038/ngeo2581.

ESDD

---

## Short Comment (SC6) · 26 Oct 2016

A. Ac

alexandeerac@gmail.com

Methane discussion part: I suggest to include work of Shakova et al. (2015): "We suggest that progression of subsea permafrost thawing and decrease in ice extent could result in a significant increase in CH4 emissions from the ESAS." (Shakhova N et al. 2015 The East Siberian Arctic Shelf: towards further assessment of permafrost-related methane fluxes and role of sea ice.Phil. Trans. R. Soc. A 373: 20140451.)

L766 Reference Warwick et al. 2016 is missing in the reference text.

Regarding the discussion on nuclear power and emission reducion rates, I suggest to include following citation (Lawrence et al. 2016, http://dx.doi.org/10.1080/14693062.2016.1179616):

"...intensities of national commitment to nuclear power tend to be inversely related to degrees of success in achieving EU climate policy goals."

---

## Short Comment (SC7) · 3 Nov 2016

Line 125: the word "year" appears twice

Line 190: The word "exaggerated" is misspelled

Line 223: A degree symbol ° is missing

Line 206: I believe the reference to "Hansen and Sato 2016" should be "Hansen et al 2016"

Line 228: Levermann et al 2013 is referenced, but the full citation does not appear in the list of references that begins on pg. 31

Pg. 11, last line in footnote 2: "Schwalm et al., 2011" contains a period, whereas throughout most of the rest of the paper "et al" has no period.

Line 479: Gt/year should probably be GtC/year

Line 676: "Costs estimates" should be "Cost estimates"

Line 683: Comment: Most fossil fuel burning sites put out a stack stream that is about 10% CO2, not "nearly 100% CO2," don't they? An IGCC plant might be different, but there are not many of those operating.

Line 762: The full citation for "Petron et al., 2014" does not appear in the list of references that begins on pg. 31. Plus, here "et al.," contains a period whereas in most of the rest of the paper "et al" has no period.

Pg. 22, footnote 7; in the first line of the footnote the word "Waste" should not be capitalized.

Line 972: The abbreviations "TA+SA" in the column heading are not explained in the text.

Line 1270 the word paleoclimate should be capitalized.

Line 1283 the word hydroxyl is misspelled

Line 1316: The word "since" is misspelled

---

## Author Comment (AC2) · 13 Dec 2016

Thanks for pointing out this typo. It will be corrected in the revised manuscript, which will be submitted within the next few days.

---

## Author Comment (AC3) · 13 Dec 2016

Thank you for drawing attention to the potential importance of soil carbon for drawing down atmospheric CO2 amount. Given the large amount of carbon that is already in the soil, relative to the atmospheric amount, it is clear that this topic deserves careful attention.

There are many potential co-benefits of improved agricultural and forestry practices that enhance the amount of carbon in the soil and biosphere, and it is well recognized that the developed world has some obligation to assist the developing world to achieve good practices.

In our revision to the paper we discuss further the potential magnitudes of carbon that might be sequestered via such activities. It is an important contribution, but cannot

prevent large climate change if fossil fuel emissions remain high.

---

## Author Comment (AC4) · 13 Dec 2016

Thank you for such careful editing, which is much appreciated!

The reference to Hansen and Sato (2016) was indeed intended to be that paper, which deals with regional climate change and national responsibilities.

We have fixed the issue about CO2 concentration in the power stack at fossil fuel burning facilities by using the phrase "the stream of concentrated CO2"

The missing Petron et al. reference is:

Petron, G., Frost, G.J., Trainer, M.K., Miller, B.R., Dlugokencky, E.J., and Tans, P.: Reply to comment on "Hydrocarbon emissions characterization in the Colorado Front Range – A pilot study" by Michael A. Levi, J. Geophys. Res. Atmos., 118, 236-242,

doi:10.1029/2012JD018487, 2013.

Thank you again for the careful editing!

---

## Author Comment (AC5) · 13 Dec 2016

Thank you for these suggestions.

There are several important recent papers on methane measurements and analysis including the implications of isotope data for source identification. We discuss these in our revised manuscript, but nevertheless the cause(s) of the recent reacceleration of atmospheric methane growth remain unclear. See especially: http://www.globalcarbonproject.org/methanebudget/index.htm

Thanks for pointing out the failure to include the Warwick paper in our reference list. Apologies for also failing to include the Levermann et al. paper. Here they are:

Warwick, N.J., Cain, M.L., Fisher, R., France, J.L., Lowry, D., Michel, S.E., Nisbet, E.G.,

Vaughn, B.H., White, J.W.C. and Pyle, J.A.: Using $\delta$13C-CH4 and $\delta$D-CH4 to constrain Arctic methane emissions, Atmos. Chem. Phys. Discuss., doi:10.5194/acp-2016-408, 2016.

Levermann, A., Clark, P.U., Marzeion, B., Milne, G.A., Pollard, D., Radic, V., and Robinson, A.: The multimillennial sea-level commitment of global warming, Proc. Natl. Acad. Sci. USA, published online, doi:10.1073/pnas.1219414110, 2013.

The Lawrence et al paper that you recommend has been retracted.

---

## Author Comment (AC6) · 14 Dec 2016

Thanks for your comments and suggestions on a number of different topics relevant to our paper and its conclusions. Several of these topics are included among the many suggestions and questions raised in the extensive reviews of the three referees of our paper, and the matters should all be substantially clarified in the revised paper, which will be submitted to the journal within the next few days. It will be much more efficient, and the responses to the topics that you raise will be much clearer, if the responses can refer to parts of our revised paper, so I request several more days to complete our response.

Sincere apologies for the time it is taking to complete our revision of the paper and responses to all comments. The comments and the detailed reviews by three referees

have been very helpful, and I trust that you will find the revised version to be much clearer.

---

## Editor Comment (EC1) · J. Dyke (Editor) · 9 Jan 2017

**Changes made to "Young People's Burden: Requirement of Negative $CO_2$ Emissions" in response to the reviews.**

There were very many substantive suggestions and issues raised by the referees – we thank them for doing such a good job on a rather long paper of broad subject matter. We trust you will find that we have been responsive, and we apologize for the time that it took to make the revisions.

We order our discussion as follows. There were several issues raised by more than one of the referees (R1, R2 and R3, where R3 is the additional review sought by Referee R2 and included as the second half of R2's report). We first discuss three topics that were in common to at least two of the referees. Then we respond sequentially to other comments of R1, R2 and R3.

**Related Comments of Referees R1, R2 and R3**

The clarifications in response to these common issues should be useful to general readers who are not familiar with the full range of climate change literature. The changes occur especially via introductory paragraphs to Section 2 and an improved Section 4 on Climate Sensitivity and Feedbacks, where Slow Feedbacks are now broken out as a subsection.

**Comment #1.** Referee R1 notes that the choice of 350 ppm as the end-of-century target for $CO_2$ is subjective. We have a good basis for the <350 ppm portion of this target, but the end-of-century date for achieving it is more subjective. (Evidence supporting the <350 ppm target includes Earth's climate response to paleo $CO_2$ changes and modern measurements of Earth's energy imbalance. The latter data allow easy calculation, without use of a global climate model, of the $CO_2$ change needed to restore Earth's energy balance.) Although the end-of-century date chosen for achieving this reduced $CO_2$ level is somewhat arbitrary, we cite paleo evidence that sea level change lags temperature change by only 1-4 centuries. We also cite evidence that the speed of the climate system response varies inversely with the magnitude of the forcing, thus the response to the human-made forcing will be faster than the response to slow weak paleo forcings. Therefore to leave excessive human-caused forcing in place for centuries, when paleoclimate data tells us that the ultimate response will be large and undesirable, warns us that we had better try to limit the period of the planet's excursion into dangerous territory. Observations of accelerating ice sheet mass loss and improving ice sheet models add to the warning. It would be nice if we could instantly restore the planet's energy balance, but we are forced to choose a conceivable period and end-of-century seems a good choice. We then investigate how difficult that task would be. So it is true that the time scale chosen to aim for getting the planet back in balance is necessarily somewhat subjective, but there is a basis for this approximate time scale.

**Comment #2.** This comment mainly concerns criticism 2) of referee R3: "A major criticism is that the paper relies heavily on global surface temperature as a benchmark against which past, current and future climate states are compared or deemed safe or "dangerous".

Actually, in other papers, we have focused on alternative benchmarks, especially atmospheric $CO_2$ abundance and Earth's energy imbalance. Several authors of our present paper, along with several other carbon cycle and paleoclimate researchers (Berner, Pagani, Raymo, Royer and Zachos), made an analysis ("Target Atmospheric $CO_2$: Where Should Humanity Aim?, Open

Atmos. Sci. J., 2, 217-231, 2008) focused on the $CO_2$ level, concluding that the target on the century time scale should be no higher than 350 ppm, and possibly lower.

In addition, several of the present co-authors have collaborated in analyses of Earth's energy imbalance. Restoring planetary energy balance provides the principal benchmark of our resulting paper titled Assessing "Dangerous Climate Change" (PLOS One, vol. 8, e81648, 2013).

R3 is correct that our present paper seems to rely heavily on temperature as the benchmark. That emphasis is perhaps a reflection of the fact that the international community overwhelmingly uses global temperature in discussing targets (e.g., 1.5°C or 2°C). However, our thinking is driven by more than the temperature benchmark, and we should have made that clearer. That is one of the things we now do in the introductory paragraphs in section 2. It is useful to make clear the relation and consistency of temperature, $CO_2$, and energy balance benchmarks.

**Comment #3.** Referee R2, as one of his two overall comments, and in later specific points, notes that "slow feedbacks" come up multiple times in our paper, but without an overall framing of their role in affecting dangerous climate change.

Indeed, slow feedbacks are an important matter, and it makes sense to discuss them explicitly near the beginning of the paper. Slow feedbacks affect both the time scale issues (Comment #1) and the appropriateness of temperature for defining dangerous climate change (Comment #2). We try to clarify all these matters by revising Section 4 (Climate Sensitivity and Feedbacks) into what we believe is a much clearer discussion of climate sensitivity and fast and slow feedbacks.

**Referee R1**

**Referee #1 General comments:**

**1.** We have changed the punctuation and reference style to those used by Earth System Dynamics. The several papers that were cited in our original manuscript, but were missing from the References, have been added.

**2.** Regarding the uncertainties in climate sensitivity and aerosol climate forcing: (1) we now discuss the uncertainty in climate sensitivity in the revised Section 4, (2) we have changed Section 10, which had been only on non-$CO_2$ GHGs, to include a subsection on aerosols, including discussion of likely future aerosols.

In brief, we suggest that uncertainty about equilibrium climate sensitivity is not a major issue for this paper. Our paper is focused on the amount of $CO_2$ that must be removed from the air to keep global temperature close to the Holocene level. Global temperature is already out of the Holocene range with the present 400 ppm of $CO_2$ and more warming is in the pipeline. The amount of $CO_2$ that must be removed to restore planetary energy imbalance depends mainly on the magnitude of future $CO_2$ emissions, not upon climate sensitivity or upon calculated future global temperature for emission scenarios with high future emissions.

However, our discussion about climate sensitivity is improved, we believe, and our simulations are described well enough that they could be repeated with alternative climate sensitivities.

Aerosols are a significant source of uncertainty, as Referee #1 notes, and a key factor in geoengineering discussion, which is the next topic raised by Referee #1. We have added discussion of aerosols as noted next.

**3.** Referee #1 asks if we see a role for geoengineering to help keep climate within the Holocene range. This topic can lead to protracted discussion and be a distraction – and the paper is already long. However, humans are already geoengineering the climate by altering GHGs and aerosols, so we need to say how the other forcings alter the $CO_2$ requirements. Thus Section 10 is now divided into two parts (10.1 Non-$CO_2$ GHGs and 10.2 Aerosols and Purposeful Climate Intervention), but we minimize the latter discussion.

**Referee #1 Specific comments:**

**L32-33:** To account for concern about the susceptibility of soil carbon to human interference, we altered the sentence in the abstract to make it less definitive. Also the relevant text (Section 9) expresses cautions more clearly.

**L59:** We now note that the $CO_2$ target is also subject to some adjustment depending on the level of other trace gases and aerosols. We include quantification of this in Section 10 (a permanent reduction of some other forcing by 0.25 W/m$^2$, e.g., allows the $CO_2$ target to change from 350 ppm to 365 ppm).

**L173-174:** Regarding the 12-month running means: with each new month of data we compute a new 12-month running mean, which is simply the average of the most recent 12 monthly temperature anomalies. Thus with the present data, which run through November 2016, the final point is the mean anomaly for December 2015 through November 2016.

However, we agree that it may be confusing that there appears to be a 6 month shift on the x-axis in the way that points are plotted in Fig. 2(a) and 2(b). Therefore we have changed Fig. 2 so that the vertical grid line represents January 1 of the year in question. Thus in Fig. 2(a) the square data point for 1880 is now centered 6 months to the right of the grid line. This location thus now coincides with the location of the 12-month running mean (January 1880 through December 1880). The merit of Fig. 2(b) is that it gives 12 meaningful points each year, among other things providing an early indication of the annual mean and a smoother curve that better defines the temperature anomalies associated with events such as a volcanic eruption or El Nino.

**L262-264:** Regarding the potential $CO_2$ released by melting permafrost: this is just one of several examples of why we argue that the target global temperature needs to be in or near the Holocene range. In that case the hypothesized $CO_2$ injection from permafrost does not occur or is small. It is also a reason why the period of overshoot of global temperature, i.e., the period in which temperature remains well above the Holocene range needs to be minimized. This matter is included in the topic "slow feedbacks", which R3 suggests we should discuss more explicitly. We have done so in the new revised section on climate sensitivity and fast and slow feedbacks, as described in our response to R3.

**L270:** Regarding the consistency of the radiative forcings (RFs) in Fig. 4 and Table A1: yes, these are consistent, and the GHG forcings are very close to the values used by IPCC. Our GHG

RFs (as stated in the paper) are calculated from formulae in the Hansen et al. (2000) paper using GHG time series that have been used in the GISS (Goddard Institute for Space Studies) global climate models for the past two decades and continually updated. Some of the GHG amounts in the period prior to air measurements of gas amounts differ slightly from IPCC time series, probably because of the use of different ice core data sources, but differences with IPCC are negligible. For example, the GHG forcing in 1850 relative to 1750 with the GISS GHG time series is 0.253 $W/m^2$. IPCC tables have 0.255 $W/m^2$. The nearly exact agreement is coincidental, as forcings by individual gases differ by as much as ~0.02 $W/m^2$, but differences of that magnitude are also negligible compared to today's GHG forcing, which exceeds 3 $W/m^2$.

Regarding the question of whether we should redo our calculations, starting them at 1750 instead of 1850. That would be a lot of work and would not seem to add anything of significant value. Certainly it does not affect the issues that our paper is aimed at, the burden of $CO_2$ that must be removed from the air this century if we wish to stabilize climate. A problem with 1750-1850 is the absence of data of useful accuracy, for both the climate forcing and climate response. IPCC attempts to define the climate forcing for 1750-1850, but it is really a guesstimate at a small number. They suggest a net forcing of about 0.1 $W/m^2$, as it is presumed that the GHG forcing of about 0.25 $W/m^2$ was offset by increasing atmospheric aerosols during the beginnings of the industrial revolution, but there are no measurements of atmospheric aerosols that would allow the aerosol forcing to be calculated. One might even wonder whether their estimated net small positive climate forcing is not more a reflection of their realization that there seemed to be a slight warming, of order ~0.1°C, over that century. But the temperature change over that period is also very uncertain.

**L283:** The referee notes that it is too vague to say that the RF used for our global temperature calculation is "similar" to the forcing of IPCC, and that our $CO_2$ forcing is about 10% greater than in AR5. The IPCC radiative forcings are for 1750-2011, while ours are for 1750-2015. For the same period (1750-2011) and $CO_2$ change (278 to 390.5 ppm) our formulae yield a forcing 6.7% larger than the IPCC value (1.816 $W/m^2$). For the same period our formulae yield a forcing 3.03 $W/m^2$ for the sum of all well-mixed (long-lived, i.e., excluding stratospheric water vapor and ozone), which is 7% larger than the IPCC central value but well within their range (2.54-3.12 $W/m^2$). The forcing depends not only on the radiative properties of the gases, but also on the global distribution of clouds and other atmospheric complexities. Our formulae are based on calculations with a global climate model with parameterized radiation in which the gaseous absorption has been fit to line-by-line radiation calculations, so we believe they are accurate, but in any case the result is within the range estimated by IPCC. We summarized the comparison to IPCC in a short footnote in Section 3, copied here:

[1] Our GHG forcings, calculated with formulae of Hansen et al. (2000), yield a $CO_2$ forcing 6.7% larger than the central IPCC estimate [Table 8.2 of Myhre et al. (2013)] for the $CO_2$ change from 1750-2011. For all well-mixed (long-lived) GHGs we obtain 3.03 $W/m^2$, which is within the IPCC range 2.83 ±0.29 $W/m^2$.

**L415-425:** As suggested by referee #1, we have reorganized the discussion of methane, moving some of the discussion from section 10 to here (section 5), so section 5 now discusses methane observations and interpretations of changing methane amount and section 10 discusses the

potential for reducing climate forcing via slowdown of methane emissions.  The additional 2016 references noted by the referee were helpful in clarifying the status of knowledge.

**L417:** Regarding the Turner et al. paper, which suggests that fossil fuel mining is introducing new $CH_4$ emission sources in the U.S.: we do need to mention this topic, because there is a great deal of concern that "fracking" for natural gas is a new emission source, so it would be odd if we totally ignored the topic.  However, as the referee notes, other recent papers show that the total fossil fuel source of $CH_4$ is not growing, which we have now clarified.

**L624:** Referee R1 suggests that we provide more information on the economics, e.g., cost comparisons.  SC4 (M. Beenstock) also suggests more economics discussion, e.g., reference to the Stern Review.  Indeed, the net cost of action or inaction on fossil fuel emissions is an issue of overriding practical importance.  We cannot convert our paper into an economics paper, but we can provide a better discussion along the lines suggested by the reviewer and commenter.

The Stern Review is a useful reference because it frames the discussion in broad terms.  In reaching its overall conclusion, that the future cost of inaction on emissions now is likely to exceed the cost of action, the Stern report recommends use of a much smaller "discount" rate in calculating the cost of future climate impacts, which is relevant to our concern that assumptions about future negative emissions can place an enormous and perhaps unrealistic burden on today's young people.  A less discussed but important conclusion of the Stern Review is that the cost of emissions reduction, if measured in terms of its effect on GDP, is uncertain even as to its sign, i.e., it is possible that the actions to reduce emissions could increase overall wealth even though they may depress certain economic sectors.

We have revised and clarified Section 9 along the lines that R1 appears to suggest, including an example calculation comparing the cost of energy infrastructure changes with estimated costs of extraction of $CO_2$ from the air.

**L668-676:** R1 points out a useful paper by DeCicco et al. that exposes the likely high carbon costs of the U.S. biofuel program for vehicles.  That biofuel program is tangential to (and even in opposition to) the potential "improved agricultural practices" that we discuss, and we would be remiss if we did not comment on this, so we added a short paragraph just before Section 9.1.

**L715-719:** Indeed, for the purpose of showing that the cost of $CO_2$ extraction will be high if high emissions continue, there is no need to assume that "improved agricultural and forestry practices" will occur to the extent of 100 PgC $CO_2$ drawdown.  To whatever degree they fail to reach that level, it increases the amount of $CO_2$ that will need to be extracted via "industrial extraction" and thus would increase the cost estimates.  We have reworded the text accordingly.

**L753-762:** We agree with the comments re the methane discussion, and largely moved this discussion to the earlier section, including several new references (see comments re L415-425).

**L822-823:** Ah, we meant from coal to gas, of course – we have fixed that.

**L994:** The (relatively small) differences are due to forcing changes between 1750 and 1880, as discussed with regard to the comments on L270 and L283.

**Edits:** Thanks -- we fixed all of these edits.

**Referee R2**

**General point (1).** Referee R2 suggests that the results would be more useful if we had extended the temperature projections beyond the standard endpoint of year 2100. It is not too difficult to extend the global temperature calculations, using simple assumptions for how the radiative forcing scenarios extend beyond 2100, so we have done that, extending the calculations to 2200, as discussed in the text describing Figure 13.

**General point (2).** Referee R2 observes that the topic of "slow" feedbacks crops up at various points in the paper, often seemingly as an aside.

Indeed, this is an important point. Slow feedbacks are crucial to our analysis, the ice sheet feedback and resulting sea level rise largely leads to our low target for global warming. And the GHG feedback makes the task of reversing GHG buildup all the more difficult. As R2 suggests, the slow feedback topic needs to be more explicit and written more clearly up front.

We now include mention of slow feedbacks in our outline of the paper in [1. Introduction]. Full discussion of slow feedbacks, while still "up front," needs to occur somewhat later, because we need to start with [2. Global Temperature], which is the principal metric of climate change, and proceed via [3. Global Climate Forcings and Earth's Energy Imbalance], which are the fundamental mechanisms of global climate change. These sections are a necessary introduction to [4. Climate Sensitivity and Feedbacks]. Revised sections 1-3 are as in the original paper, except that we have addressed several specific issues raised by the three referees. Section 4 is split into subsections [4.1 Fast-Feedback Climate Sensitivity] and [4.2 Slow Feedbacks].

We believe that this explicit organization of section 4 with discussion of slow feedbacks in its own subsection makes the paper clearer, especially for readers who do not have extensive background in climate studies, and we thank the referee for his/her suggestion. Slow feedbacks are a very large part of the story, the principal reason that it would be dangerous to allow global temperature to long remain far outside the Holocene temperature range.

**L36:** Agreed, "current generation" is ambiguous. We have restated to eliminate that terminology.

**L160-162:** Now we have 11 months of data for 2016, rather than 8 months, so it is now clear that the 2016 global temperature is easily the highest in the period of instrumental data.

**L175-179:** Yes, we agree that future temperatures can be expected to fall both below and above the linear trend line. We can also say with confidence (this relates more to a global temperature issue raised by R3) that the approximate warming trend will continue, because Earth's energy imbalance is the immediate drive for continued warming (rather than annual growth of climate forcings, although the annual growth has some effect on the imbalance). We agree that we

should not let extreme endpoint values cause a misleading trend; when we exclude the final year the calculated trend (0.176°C/decade) still rounds to 0.18°C/decade).

**L193-198, L198-206, L207, L193-212:** Agreed on both points re L193-198, i.e., the sentence is too long and the point being made in the first half of the sentence can just as well be omitted. R2 raises several other valid points about this paragraph. So we have rewritten it in a way that gets to the point and avoids tangents. Also we explicitly mention the issue (seasonal bias in the proxy data) raised by Liu et al. (2014). The question about the significance of the current (smoothed, via linear trend) temperature is better put not as one of statistics, but as a matter of physics. Earth's present (substantial) energy imbalance makes it implausible for the mean temperature over the next several decades to be less than the present (smoothed) temperature.

**L233:** Of course we meant a temperature anomaly of +0.7°C (relative to preindustrial temperature), not an absolute temperature – we have corrected that error.

**L248:** Agreed that Eemian temperature is important for assessing the dangerous level of warming, and although we are relying on analyses made by others, we have now made this discussion clearer and have added one more reference.

**L251:** Yes, it seems unnecessary to bring up slow feedbacks at this point – so we have eliminated that statement.

**L339-340:** What we meant was that the present temperature based on the linear fit is more appropriate than the actual current temperature, which is elevated by the 2015-16 El Nino. We have changed the wording so that there is no impression that we may have done something more formal such as subtracting out a standard El Nino signal.

**L352-353:** We have added vegetation feedback in the new subsection (4.2) on slow feedbacks.

**L354-356:** Well, we might argue that the changes of GHGs and surface albedo are more than a favored explanation. Given a fast-feedback climate sensitivity ~3C for 2xCO2 these boundary forcings account for practically the entire glacial-interglacial temperature change. The results fit well the temperature variations over the entire 800,000 year period with ice core data for GHG amounts (and with ice sheet size inferred from sea level information).

However, in responding to R2's earlier suggestion, we revised this section into "fast feedback" and "slow feedback" subsections, and the referenced sentence was eliminated. Slow feedbacks and their significance are now discussed in a more organized way, which we hope is clearer and a major improvement, because of the importance of slow feedbacks.

**L359-361:** This is fixed via the new subsection on slow feedbacks.

**L470:** Typo fixed.

**Figure 9:** ΔFe, the annual increase of the GHG climate forcing, is now indicated in the caption.

**L510-521 and Figure 11:** O.K., we have switched from GtC to PgC, and everywhere we give extracted amounts in PgC, not ppm. Fig. 11 fixed to show that cumulative amount is in 2100.

**L541:** This comment raises two points: notation in equation (1) and the way we have treated the volcano/stratospheric aerosol forcing in the Green's function calculation. The notation problem is easily solved: yes, we should use t' for the functions under the integration sign. Also we have added the integration limits (from 1850 to time t); it is hard to make this look right using our "typewriter", so we will need to check what the typesetter does.

How to handle the effect of intermittent volcanic aerosols is a nuisance for modelers. The ocean temperature at the beginning of the run (in the case of a global atmosphere-ocean model) is the difficulty, because the ocean temperature is dependent upon the prior history of stratospheric aerosols. With atmosphere-ocean GCMs the proper (but expensive) procedure is to have a long spin-up run with volcanoes sprinkled in time at a climatologic amount (or a constant mean amount), but this is often not done. If the spin-up does not include volcanoes but they are included in the transient climate experiment, they have an inappropriate long-term cooling effect. The mean volcanic aerosol forcing over 1850-2000 in the Sato et al. data set is about -0.3 W/m$^2$.

So what referee R2 is implying is correct: by in effect ignoring volcanic aerosols in the time preceding 1850, the volcanoes after 1850 have a long-term cooling effect. As t goes to infinity the average cooling for -0.3 W/m$^2$ is about 0.2°C, which is not negligible.

It is easy to include a term in the Green's function calculation to avoid an inappropriate long-term aerosol cooling effect, if we assume that volcanic aerosols prior to 1850 had average forcing over time of -0.3 W/m$^2$. That's an approximation, but there is no way to be exact without knowing the specific history of volcanoes before 1850. The effect is not very obvious in the graphs, but we have added that term and we are glad that the referee forced us to think about this.

**L617-618:** Yes, we have changed GtC to PgC everywhere (we use PgC, rather than Pg because we want it to be clear that we refer to the mass of C not the mass of $CO_2$).

**L818:** We have made the suggested change.

**Referee R3**

**Item 2) [Item 1) did not suggest changes]:** Item 2) concerns a major criticism that the paper relies heavily on global mean surface temperature as a benchmark against which past, current and future climate states are compared or deemed safe or dangerous.

This criticism was initially surprising, as the first author has written several papers that explicitly or implicitly make that very criticism. It is true that we superficially reverted here to use of global temperature as the benchmark, but that was because of reliance on that benchmark in all of the international discussions and the great publicity surrounding 2°C or 1.5°C targets, so the broader public and scientific audiences seem to be familiar with it, if not totally reliant on it.

However, this is a very useful criticism, because it reveals that we did not do a good job of defining the rationale of the paper, so some revisions are required. Before defining those let me note two other benchmarks that we have relied on and at times have asserted that they are superior in certain respects.

One of these is atmospheric $CO_2$ amount, or $CO_2$ equivalence to include all greenhouse gases (GHGs). Several of the present authors were co-authors on a 2008 paper: "Target atmospheric $CO_2$: Where should humanity aim?", which provided the scientific basis for the organization 350.org. One big advantage of $CO_2$, over temperature, is that it only requires measurements at one point to know the relevant global amount. Thus, especially for the last 800,000 years for which precise ice core measurements are available, it provides a very precise benchmark.

Another benchmark on which several of the co-authors have spent a lot effort is Earth's energy imbalance. A great merit of this benchmark is that it integrates over all climate forcings, known and unknown, and tells us something very valuable about where climate is heading in the relatively near-term. If we measured Earth's energy balance at only one point in time, it would be useful, but now it is much more powerful as we have rather accurate data for more than a decade, and useful data for several decades. For example, the data show that since the growth rate of GHG climate forcing jumped up about half a century ago and has remained high (as forcing additions from CFC and $CH_4$ declined, $CO_2$ increased) Earth's energy imbalance has remained high, driving continual global warming since ~1970. Thus we can be confident that on decadal time scales global temperature will continue to rise, because there is much more energy coming in than going out.

So why did we choose to focus on global temperature in this paper? One reason is for the sake of communicating with a broader audience, both scientists and non-scientists. A problem with using temperature is that it requires introducing climate sensitivity into the story. However, that can be done in a reasonably simple and transparent way by use of the simple Green's function approach (for converting climate forcing into temperature) with a canonical climate sensitivity of 3°C for doubled $CO_2$. A second reason for the temperature metric is that our paper emphasizes the effect that "slow feedbacks" have on determination of the dangerous level of human interference with the climate system. For the sake of understanding when slow feedbacks kick in or become substantial, global temperature is a very helpful metric, as we now discuss better in the revised paper.

Nevertheless, we are not really relying on temperature as a sole benchmark – we are actually aware of and using these several benchmarks, and relating them to temperature. That is why, e.g., at one point we insert a "consistency check" to verify whether these different benchmarks are consistent in yielding the same conclusion, i.e., we check whether the changes over the industrial era of climate forcings and temperature are consistent with the energy imbalance and the canonical climate sensitivity.

What we have tried to do in revision is explain our approach better. We much appreciate the criticism, because it is an important matter that needed to be made clearer. If we relied on temperature alone, the uncertainties in temperature measurement would strongly restrict our conclusion. However, we argue, and we believe that we have presented substantive support for the argument, that even a (fast-feedback) 1.5°C warming left in place for centuries would be dangerous, in part because it would drive slow feedbacks.

We have borne the criticism about the temperature metric in mind in revising the paper. Two places where we have tried to explicitly make our approach clearer are (1) an added paragraph in the introduction, as the second paragraph, that discusses the multiple metrics, (2) an added paragraph in temperature section, as the second paragraph of that section.

**Item 3):** This comment concerns the question of whether we show that 450 ppm (as opposed to 350 ppm) of $CO_2$ or additional warming of 0.5-1°C constitutes dangerous climate change. This topic relates very much to item 2) above. Perhaps if one considers only the $CO_2$ metric or only the temperature metric, the case for 450 ppm, which would be consistent with 1°C additional warming, being dangerous, does not seem as clear. We believe that our analysis using multiple metrics strongly makes the case that it would be dangerous.

There is another important point that reinforces the danger of such additional climate forcing: namely the fact that the temperature analysis that we make with a Green's function calculation, as well as the temperature analyses made by the IPCC studies with sophisticated general circulation models, are basically fast-feedback climate models. It is good that Referee R2 suggested that we keep this as a separate point, because it really is additional. If global climate is allowed to reach 1.5-2°C, which we make clear is far above the Holocene range, it means that we (humanity) are unleashing the slow feedbacks that inevitably will occur in response to that temperature elevation above the Holocene range. Those slow feedbacks, which include GHG increases associated with global temperature increase (presumably coming from melting permafrost, warming soil, warming ocean, etc.) as well as melting ice and rising sea level. These slow feedbacks not only make it harder to control atmospheric composition and limit global warming, they (ice sheet melt especially) also carry impacts that could be detrimental.

In our revision we have tried to make clearer the limitation and significance of the fast-feedback temperature analysis, as well as the implications of the slow feedbacks.

**Item 4):** This comment concerns the applicability of paleoclimate to inferences about climate sensitivity. For sure the (orbital) mechanisms that initiate glacial-interglacial climate change are very different than the human-caused climate forcings. However, as all climate models and paleoclimate analyses show, the orbital (insolation) forcings are not the mechanisms that

maintain the quasi-equilibrium paleoclimate states. The two dominant mechanisms are changes of atmospheric GHGs (and, to a lesser extent, atmospheric aerosols) and changes of the planetary reflectivity (due especially to changes of ice sheet area, but also vegetation cover). These mechanisms account for most of the paleo global temperature change, and sufficiently good knowledge of those changes exists to allow useful empirical derivation of fast feedback climate sensitivity. However, we must admit that we did a poor job of describing this topic, with regard to both fast feedbacks and slow feedbacks. In revising the section on feedbacks, we believe that we have much improved the clarity of the fast-feedback portion, as well as adding an explicit subsection on slow feedbacks, as suggested by R2.

**Item 5):** This comment is also very useful. R3 notes that we have not justified why we calculate the linear trend of global temperature from 1970 to the present, rather than beginning at some other date. On the one hand, the period chosen is the time of steep temperature increase, and it is worth knowing the warming rate in that period. However, there are better reasons, and we should have given those. First, 1970 is when the growth rate of GHG climate forcing reached a rate such that it overwhelmed other variable climate forcings, as we show in a later figure in the paper (Fig. 9). Second, as a result of the rapidly and continuously increasing GHG forcing, it is also the time when Earth became substantially out of energy balance, as revealed by analyses of ocean heat content. It is this energy imbalance that is the immediate cause of continued global warming. As long as the imbalance continues to be large (in the range 0.5-1 $W/m^2$) we can expect global temperature to continue to rise at a similar rate, even though there is short-term dynamical variability. We have clarified the discussion of the temperature trend accordingly.

**Item 6):** We agree that Earth's energy imbalance, which is practically determined by the continual growth of ocean heat content, is the more relevant measure of global warming and have clarified the discussion in that regard. Also we changed "global warming" to "global surface warming" to be more technically correct.

**Item 7):** We believe that the revised paper now makes clear that the conclusions regarding possible overshooting safe targets are based on much more than the temperature. Uncertainties in temperature data are indeed one of the reasons that more metrics are needed.

R3 raises an important point in the last sentence of item 7): the possibility that "bleak" conclusions lead to no actions rather than motivating urgent changes. Our opinion is that the actions needed to rapidly slow fossil fuel emissions actually make sense on a number of grounds, and have multiple co-benefits, so we have tried to make this point clearer. For example, other reviewers asked for some cost estimates of mitigation actions for comparison with the estimated costs of $CO_2$ extraction, and that comparison is favorable in the sense that costs of slowing down emissions are likely much less than trying to clean the atmosphere later.

**Minor clarifications, recommendations, and typos:**

**1) Figure 4:** the relevant uncertainties for this paper are for the net GHG forcing and the aerosol forcing, and the combined GHG + aerosol forcing. It is hard to show those on this diagram (and creates more distraction than relevant illumination), which is intended to provide an indication of

the relative contributions of different forcings, but there is an excellent presentation by Myhre et al. (2013), i.e., the radiative forcing chapter of the last IPCC report – so we have added a comment to the figure caption and reference to Myhre et al. (2013).

**2) and 3):** These typos have been corrected.

**4):** the suggestion to find other references for current estimates of Earth's energy imbalance was useful, as there are three 2016 papers published or in press, including up-to-date corrections of instrumental biases in the ocean heat content measurements. These references are now included; their results are consistent with the range that we stated.

**5):** No equations are used in this check on the consistency of observed warming with the climate forcing and remaining planetary energy imbalance. The only thing required is the proportionality constant between forcing and temperature change, i.e., climate sensitivity, which is 0.75°C per W/m$^2$. Nevertheless, it should now be clearer, because of the more explicit discussion of climate sensitivity and fast and slow feedbacks.

**6):** Yes, consistent with related suggestions of R2, we have discussed ice sheets and other slow feedbacks in their own subsection.

**7):** The definition of SRES has been added.

**8):** Although all carbon cycle experts would probably agree with the statement, a definitive statement is not needed for our purposes, and the discussion thereof could be long – so instead we changed "it is not expected to continue" to "it may not continue"

**1. Figure 9:** In the period in question the gas changes are from accurate in situ measurements and the radiation equations are published and in good agreement with other researchers. The fact that the growth rates of CFCs and $CH_4$ declined at the same time is coincidental, as the reasons differed. The $CO_2$ dip seems to be associated with Pinatubo effects, at least a strong case for that has been made in the literature.

**2. Line 483:** I presume that this refers to RCP2.6, not RCP2.5. We have double-checked our graph, and it is correct. The shift to negative growth of the net (all GHGs) forcing is in 25 years. Fossil fuel emissions do not become negative until about 2070 (see Fig. 10), but atmospheric $CO_2$ growth becomes negative long before emissions are zero (today uptake of emissions is almost half of emissions). In any case, the data for that specific line in Figure 9 are not based on our calculations, they are taken from IPCC – the IPCC tables are published.

**3. Line 642:** The REDD+ program has an approach similar to what we are advocating, but it seems to still be in its infancy, and, of more relevance to the question of whether we can use it for quantitative statements about how it relates to our carbon uptake goals, there are major uncertainties regarding how to assess REDD+ carbon uptake potential. We note that Richard Houghton is planning to address this topic at the AAAS meeting in February 2017. As of now, it is still a research matter, which we cannot undertake as part of this paper.

However, in the spirit of R3's question, this topic is addressed by land use (LU) plans outlined in the 189 Intended Nationally Determined Contributions (INDCs) that were provided to the UNFCCC as of 04 April 2016. There is a 2016 Technical Annex to the UNFCCC that gives data for the estimated $CO_2$ removal rate for the proposed INDCs, although it only extends to 2030. We have added reference and quantitative discussion of this. As may have been expected, the expected drawdown from the INDCs is only a fraction of what is needed to achieve our goal of 100 PgC removal, the annual carbon removal being about one/third of what Smith (2016) estimated to be possible and one-fifth of the rate that would be needed to achieve our goal of 100 PgC this century.

**4. Carbon removal section:** we have revised the text for clarity, and we believe that it is now easier to read. We doubt that adding another figure helps.

**5. L695:** R3 suggests that we should include negative impacts of basalt dust, not only benefits. We agree, and thus we have added the following:
Against these benefits, we note the potential negative impacts of increased mining operations including downstream environmental consequences if silicates are washed into rivers and the ocean, causing increased turbidity, sedimentation, and pH, with unknown impacts on biodiversity (Edwards et al., 2016).

**6. L775:** The methane discussion has been revised and reorganized, based in part on suggestions of R1, and we no longer speculate on a specific methane reduction.

**7. L785:** Yes, there are a lot of interesting aspects and uncertainties in global climate and biogeochemical cycles, not just the one noted here by R3 but also many others. We can't get into each of these, but note that this point draws attention to the merits of our approach, as (1) we look at the most up-to-date global data on each of the three main GHGs and compare their ongoing changes to their ranges in the IPCC models, and (2) we emphasize the response of each of these gases to large paleoclimate changes. While not perfect, these empirical approaches do include all processes occurring in the real global systems.

**8. Figure 12a:** Yes, this is an important point. In the real world 6%/year reductions are hard to imagine, but 3%/year is plausible. Rather than just showing this result, we should draw attention to it, which we have done in the revision.

---

## Author Response (AR2)

We thank the editor and referees for their helpful reviews. Given that our revision addresses the Reviewers' concerns, it is disappointing that the paper must go out again for another round of independent reviews. New referees are likely to have their own perspectives, opening the possibility of still further iterations, a questionable state of affairs, considering that the science of our paper does not seem to be at issue.

Our response here follows the order of the issues described in:

**Editor's Decision: Reconsider after major revisions** (16 March 2017) by James Dyke

We agree that the increased length of the paper detracts from its readability. We note that the increased length resulted from a very long list of requested explanations and clarifications by the referees of our original paper (ESDD version, published on 4 October 2016). The best approach now is probably, as suggested by the Editor, to move less essential material to the Appendix or Supplementary Material. Supplementary Material is less convenient to readers, and thus not opened by most readers, so we have chosen to use the Appendix, but it would not be difficult to instead put some of the Appendix into Supplementary Material, if the Editor so instructs us.

**Reviewer #2 Major Comments**

**(1) Paring down, distilling the main message**

Reviewer #2 suggests that our criteria for the dangerous level of warming or $CO_2$ could be made clearer. An outline of our message follows:

(a) T (global temperature) provides the fundamental constraint; we use Holocene T and Eemian T as guides to help define the allowable T.

(b) We show that current T has already reached Eemian T and is far above the Holocene T range.

(c) $CO_2$ is the principal forcing that affects T, and because fossil fuel $CO_2$ remains in the climate system for millennia (a period exceeding the expected response time of ice sheets) $CO_2$ is the crucial climate forcing whose amount needs to be constrained if eventual large changes of ice sheet volume are to be avoided.

(d) Other climate forcings alter the safe range for $CO_2$, but the potential for reducing these other forcings to permit larger $CO_2$ concentrations is very limited.

(e) As a result, the burden on future generations (to somehow extract atmospheric $CO_2$) will be very high unless fossil fuel emissions are reduced rapidly.

As Reviewer #2 concludes at the end of his point f), the Abstract was already close to this outline, so only moderate change of the Abstract is required. In rewording the Abstract, space limitation prevents us from explicitly going into the trade-off between $CO_2$ and other forcings. However, we have edited the Introduction to make these points clearer.

**(2 and 3) Cut tangential material**

Co-authors have reviewed the manuscript with the specific objective of finding material that can be pared and/or moved to the Appendix. This has improved the readability beyond the

proportion of shortening, because the material removed from the main text is less central to the main theme.

We have followed Reviewer #2's suggestion, making the Introduction and Discussion sections clearer statements of the distilled main message, as summarized under (1) above, while avoiding an increase in the length of the Abstract.

**The two other general concerns of Referee #2**

**(1) Are we using an Eemian-based limit or a Holocene-based limit?** We use both, and we now state this more explicitly. Eemian temperature is clearly too high for a long-term target, given the high likelihood that it would lead to multi-meter sea level rise. The appropriate initial target, for the sake of keeping shorelines close to where they have been for the past several millennia, is within or close to the Holocene range. We do not need to define the target more precisely than that, because we show that global temperature has already risen far above the Holocene range and it will take time to get the temperature moving downward. This is the same reason that the target for $CO_2$ ("less than 350 ppm") does not need to be more precise yet.

Why do we need both Eemian and Holocene comparisons? Could we just say that an appropriate target is to stay in or close to the Holocene range? Perhaps, but the Eemian (which is not much warmer than the Holocene maximum) reveals how substantial the consequences could be with only a moderate overshoot of the target. Because Eemian temperature was only approximately $+1°C$ it reveals that the long-range target for global temperature must indeed be very close to preindustrial Holocene temperature.

(2) His/her second concern relates to whether sea level change should be framed as a feedback. Paleoclimate studies show that the two large slow global climate feedbacks (changes in response to global temperature change, which amplify or diminish that global temperature change) are changes of long-lived GHG ($CO_2$, $CH_4$, $N_2O$) amount and changes of ice sheet size. We did not intend to leave an impression that sea level change was the feed, but in tightening up the slow feedback section (we agree that it was too long), we have made clear that the GHGs and ice sheet size are the principal slow feedbacks.

**Reviewer #2 and Editor's comment re carbon fee**

Reviewer #2 suggests cutting lines 1227-1237 and 1246-1249. The editor notes that discussion of a carbon fee is within the scope of the journal, but would need to appropriately framed within the rest of the manuscript and occupy commensurate space.

Given the additional time and space that would be required to develop such a section, and the possibility that reviewers might continue to object to it, we decided to basically follow Reviewer #2's suggestion and remove that material. We still have one sentence following the historical examples of fast emission reductions, simply noting that those examples were not aided by a carbon fee or tax. The conclusion (that rapid emission reduction is conceivable) is strengthened by this notation.

**Reviewer #3 Major Comments**

**L127:** Reviewer #3 notes that it would be interesting to compare our model's TCRE (Transient Climate Response to Cumulative $CO_2$ Emissions) to that of other models. We have now done so, calculating TCRE(t) as specified in Section 10.8.4 of IPCC (2013), specifically TCRE(t) = $TCR(t) \times CAF(t)/C_0$, where $C_0$ = preindustrial atmospheric $CO_2$ mass = 590 PgC and CAF (t) = Catm(t)/Csum(t), Catm(t) = atmospheric $CO_2$ mass at time t minus $C_0$ and Csum(t) = cumulative $CO_2$ emissions at time t.

These calculations yield TCRE = 1.67°C per 1000 PgC at time t = 2100 with constant emissions (which yields cumulative emissions of 1180 PgC at 2100, which is almost precisely the midpoint of the range assessed by IPCC, i.e., 0.8°C to 2.5°C per 1000 PgC (IPCC, 2013). To avoid making the paper more technical and longer, we have added one sentence in the text and defined the calculations briefly in Appendix A2.

**L150-154:** Reviewer #3 suggests that we cite the final Paris Agreement rather than the draft. Yes, that is now possible, so we have done so, and we thank the reviewer for pointing this out.

**L432-440:** Reviewer #3 suggests that we compare our estimates of observed warming with IPCC AR5 WGI estimates. We have added that comparison, which is a useful clarification.

**L508:** Reviewer #3 is right that it is possible to be more precise in comparing the PETM, a doubling of $CO_2$, and burning all fossil fuels. We previously (Hansen et al. 2013b paper in Proc. Roy Soc.) estimated that fossil fuel reserves + resources amount to ~15,000 GtC (which is almost the same as the 15,600 GtC upper limit estimated in the AR5-WG1 report. The PETM release is usually estimated as ~5,000 GtC (though outlier studies estimate closer to 10,000 GtC), which is much greater than known reserves (~1900 GtC) but about 1/3 of all reserves + resources. Without going into all this detail, we have restated the comparisons better.

**L535-538:** Reviewer #3 asks for a clarification of the experimental results of Crowther et al. (2016) for soil carbon release with soil warming. We have expanded that sentence to achieve that clarification.

**L583-584:** Reviewer #3 suggests that it may be useful to clarify that oxidation of $CH_4$ is the atmospheric sink, which we have now done.

**L634:** Reviewer #3 is correct, the reference should be Hansen et al. (2000), not Hansen and Sato (2004). We have made that correction.

**Figure 10:** Reviewer #3 asks what airborne fraction is assumed in arriving at the scale for the right hand axis. Actually, the ppm on the right scale is just the emissions in another unit (1 ppm is ~2.12 GtC); we have clarified that in the Figure 10 caption.

**L786:** Reviewer #3 notes that atmospheric aerosols seem likely to decrease in the future as fossil fuel use declines. Although we assess the probability of this differently than Rao et al. (2017), it is a good point that only strengthens our argument that it will be difficult to obtain a decrease of the net non-$CO_2$ anthropogenic climate forcing. We also thank the Reviewer for the Rao et al (2017) reference, which we now incorporate in the first paragraph of Section 10.2.

**L917:** Reviewer #3 notes that burning of biofuels at power plants is only one option for BECCS. We have reworded the sentence to clarify that.

**L975:** Reviewer #3 notes that costs of $CO_2$ extraction might decline in the future, and it would be helpful to discuss the potential for falling costs. We agree that it is conceivable that costs might decline below the range estimated by current experts, but given the energy requirements for $CO_2$ removal there is reason to believe that the cost will remain substantial. However, we have added a caveat sentence about possible cost reduction.

**Figure 14:** Reviewer #3 asks that we clarify the energy accounting method used to compare fossil fuel and renewable energy amounts. We are uncertain what is meant by that, unless perhaps it refers to the difference between "nameplate" (maximum potential) renewable energy as opposed to the energy actually produced. The Boden et al. and BP energies refer to total energy produced. BP likes to add the adjective "primary" (primary energy consumption), so we have added that for possible clarification.

**Figure 15:** Reviewer #3 is correct that the RCP scenarios do not include natural sources and feedbacks. We now note this in the figure caption and add a brief comment at the end of the relevant paragraph, which is now the second paragraph in Appendix A14.

**L1100-1103**: Reviewer #3 notes that references are needed here. The principal appropriate reference is perhaps Prather et al. (2013), which is Annex II to IPCC (2013). However, this paragraph was removed in distilling the paper.

**L1120:** Reviewer #3 notes that a reference is needed. An appropriate reference is to tables in the same Annex II to IPCC (2013) that we noted just above. Note that these paragraphs, which were in the main paper, are now in the Appendix.

**L1136:** Reviewer #3 asks if it is possible to provide a range for the aerosol negative forcing of the order of 1 $W/m^2$. "Of the order of" was meant to imply only order of magnitude knowledge, but in fact we can provide a good reference, Boucher et al., the aerosol and cloud chapter of AR5, specifically Figure 7.19 is a good summary.

Note that the large uncertainty in the aerosol forcing, and in the history of the anthropogenic aerosol forcing, is not quite as damning for projections of the future as it may appear at first glance. We have good knowledge of the present planetary energy imbalance, so the issue is primarily how the aerosol forcing will change from its present value going forward.

**L1238-1240:** Reviewer #3 notes that the policies that led several countries to rapidly reduce $CO_2$ emissions were not adopted with that objective, but rather for the objective of energy independence from oil. We agree that the earlier period is not a perfect analogue, and have now so noted, but the present situation actually has even more comprehensive incentives.

**L1295:** Reviewer #3 suggests that we may want to update Fig. A1 with data from Le Quere et al. (2016). We do update that figure (and others) using data of BP, which allows our graphs to go one year further than Le Quere et al. (2016), but we also note now that the data we use are in good agreement with Le Quere et al. (2016) for the prior years. The BP data are preliminary for

the most recent year (2016), but our experience from prior years is that the adjustments of final data have been small.

**Reviewer #1 Minor Comments**

**L490-492:** Part of the text is missing.  Yes, we apologize for that typographical error, which has been corrected.

**L768-769:** Role of Montreal Protocol could be made clearer, i.e., that it does not explicitly address emissions but rather production of certain substances.  That is true, and we have reworded the description accordingly.

**L1055:** Referee #1 correctly points out that gases other than $CH_4$ and $CO_2$ in recent years have together been providing as much forcing as $CH_4$.  That is true, but our sentence here refers to the surge in the growth rate of GHG climate forcing.  As Figure 9 shows, the increase in the growth rate of GHG climate forcing in the past decade is due to increased growth rates of $CO_2$ and $CH_4$. The picture (Figure 9) is probably worth more than our words, but we do try to make clear also the potential merit of slowing or stopping the growth of these other GHGs.

**L1117:** Yes, thank you we have added "stratospheric" before ozone for clarity.  (This paragraph is now in the Appendix).

**L114:** Comma added.

**L252:** Comma added

**L256:** for clarification, parentheses have been imbedded

**L265:** missing degree symbol corrected

**L322:** "is" → "are" correction is made

**L410:** "unknown unknowns" is useful terminology, but perhaps it is an Americanism popularized by a perhaps unpopular Secretary of Defense.  Simple unknowns are those factors that we know about, but have not quantified well enough.  Unknown unknowns are additional uncertainties, which we are not presently aware of.  However, we will change it, if the editor thinks it is advisable to do so.

**L595:** Comma added (this subsection is now in the Appendix)

**L778:** Comma added (this reference is now in the Appendix)

**Reviewer #2 Minor Comments**

**L133:** We have added "magnitude and" for clarity, as suggested.

**L193:** Observed global temperature data are now several months further advanced, so that phrase no longer appears in the description of the data.

**L270-277:** The suggestion is that we could move this paragraph to the section on feedbacks. Because the sensitivity of sea level to temperature change is so crucial to our rationale, including our choice of the dangerous level of warming, we feel that it is better to have this brief paragraph near the beginning of the paper.

**L282-284:** Reviewer #2 notes that the empirical temperature data for the past century (Fig. A2a), which we use to compare the magnitude of warming over land and ocean, refers to a transient case in which the ocean has had insufficient time to reach its equilibrium response. However, the land also has not had time to reach its equilibrium response, because, to a substantial degree, the land responds to the ocean temperature (more so than to the direct radiative forcing).

To help quantify the equilibrium ocean and land responses, we have calculated the ocean and land warmings in years 901-1000 (close to equilibrium response) of the GISS ModelE-R for both the coarse resolution version of the model used in our "Ice Melt" paper (Hansen et al., 2016) and the fine resolution used for CMIP5 model comparisons. The ratio of land/ocean warming is 1.75 and 1.82 for these two cases, so indeed it is less than a factor of two, but not enough to qualitatively alter the conclusions. Using 1.8 for the ratio SAT(land)/SST yields 1.24 as the factor by which global temperature change exceeds SST change, as compared to the factor 1.3 obtained from observed temperature change of the past century. To avoid making the text longer, we make a brief statement in the paper and give the model results in Appendix A6.

**L303:** Reviewer #2 asks what (global) temperature would correspond to 350 ppm.

The choice of 350 ppm as an initial target for $CO_2$ amount is based mainly on the fact that (if the net of all other forcing changes is zero) reduction of $CO_2$ to this amount would restore Earth's energy balance. Thus to first order we might expect global temperature to stabilize at about the present (smoothed) value (about +1°C relative to the 1880-1920 mean).* The complete story is not so simple, for example, slow feedbacks may not yet be in equilibrium, but it seems useful to give this approximate correspondence. The best place for this seems to be earlier in the paper: we have added a statement near the end of the second paragraph of the Introduction.

*It also happens that, if climate sensitivity is ¾°C per $W/m^2$ (3°C for doubled $CO_2$), the change from preindustrial $CO_2$ (278 ppm) to 350 ppm yields +1°C. This way of reaching ~1°C requires the assumption that the net of non-$CO_2$ forcings is small. Although that assumption happens to be true for current estimates of the non-$CO_2$ forcings, the uncertainty in aerosol forcing is so large as to make this approach less meaningful.

**L346:** Reviewer #2 suggests rewording this sentence to remove possible confusion about the roles of climate forcing and planetary energy imbalance. We have reworded the first two sentences to accomplish that objective, while using fewer words.

**L476-478:** Reviewer #2 questions the interpretation of the $CO_2$/sea level relationship in paleoclimate data and notes that it is not central to our arguments. Given that modern forcings other than $CO_2$ are not negligible, we are instead relying on the relationship between sea level

and global temperature.  So we agree that reference to and discussion of the Foster and Rohling paper can be cut, so we have done so in the interests of limiting paper length.

**Line 491:** Yes, there was a typographical error upon inserting a late change to that sentence, which has now been corrected.

**Line 634:** Yes, the correct reference is to Hansen et al. (2000).  Thanks for catching that!

**Lines 643-658:** Reviewer #2 suggests that these two paragraphs are tangential.  We have removed them.

**Lines 698-700:** BP reference has been added.  (This topic has been moved to Appendix A9).

**Figure 9:** Reviewer #2 notes that Fe, MPTGs and OTGs are not defined. They are defined earlier in the text and in the caption of Figure 4, so we now add direction to the latter caption in Fig. 9.

**Equation 1:** Reviewer #2 (thankfully) continues to badger us about this equation.  Programming of the two terms was correct, but the representation in equation (1) was not.

The first term is the contribution of the forcing increment of each preceding year multiplied by the appropriate portion of the equilibrium response, $R(t - t')$, with integration up to time t.

The second term is an approximation of the small warming due to recovery from volcanoes that occurred prior to 1850.  Those unknown volcanoes are approximated as a constant forcing of $-0.3$ W/m$^2$.  Resulting warming increases in proportion to $R(t - 1850)$.  This second term is not a constant, it grows in time.  At t = infinity, the second term is 0.3 W/m$^2 \times 0.75$°C per W/m$^2 =$ 0.225°C.  This warming is countered by volcanic cooling of identified volcanoes in the period 1850-present, so that on average volcanoes do not cause a long term warming or cooling.

**L761:** We have incorporated the footnote into the text, as suggested.

**L765-782:** Reviewer #2 suggests that these three paragraphs are tangential and can be removed, thus improving clarity.  We largely adopt his suggestion.  We need to retain the final sentence of the 3$^{rd}$ paragraph to define our simulations, so we add this sentence to the end of the paragraph that preceded these three paragraphs.  We also retain the first of the three paragraphs, but we move it to the appendix on non-$CO_2$ GHGs, using it as the introductory paragraph.

**L881-888:** Reviewer #2 says this paragraph is an example of an unnecessary digression. Because of the importance to our climate stabilization scenario of the $CO_2$ drawdown from "improved agricultural and forestry practices" we would like to keep this clarification about what we mean.  It is set off at the end of a section as a "comment".

**L959-961:** Reviewer #2 suggests that this sentence could be omitted.  There may be some confusion here.  This sentence does not refer to the "alternative scenario," for which, as mentioned above, we decided to drop the discussion (L643-658).  Instead the sentence here refers to the 6%/year emission reduction scenario defined in the 2013 Plos One paper that served as the scientific basis for Alec L versus United States lawsuit filed in 2012, which was finally lost by Alec L in the United States DC District Court (the court just below the Supreme Court from which Supreme Court justices are commonly selected).  That case was lost largely because

a sufficient Constitutional basis for the claims was not demonstrated, and the Alec L decision is now sited by the U.S. government as a reason to dismiss the new case (Juliana et al versus United States). The important point is that while in 2012 it only required extraction of 100 GtC from the air to get back to 350 ppm $CO_2$ by 2100, an amount that seemed feasible with reasonably natural improvements in agricultural and forestry practices including extensive reforestation, continued high emissions qualitatively change the prospects. This is a fairly important conclusion of the paper, which helps illustrate how young people are being impacted by continuing high emissions, and it only requires one sentence here.

**L962-965:** Reviewer #2 suggests that we can save some sentences here, as there is some repetition from an earlier part. We agree and have made this change.

**L962-970:** Reviewer #2 suggests that these details should have been given in the earlier section where extraction scenarios were first defined. Now that we have minimized the size of this paragraph by omitting the first three sentences (see comment above), and considering that what remains relates to the cost discussion to follow immediately, it seems better to keep the remaining sentences here.

**L971:** Reviewer #2 feels that this statement is confusing. We agree that it can be stated more clearly. We have revised this along the lines suggested by Reviewer #2.

**L1048:** misspelling corrected.

**Reviewer #3 Minor Comments**

**L56:** Thanks – typo corrected.

**L85:** Reviewer #3 suggests that we should clarify whether we refer to annual or cumulative emissions from China. We have fixed that in the course of shortening the paragraph.

**L111-112:** Yes, we used a less appropriate reference – we have switched to the one suggested.

**L135:** CMIP acronym spelled out; as noted, we don't need the integrated assessment reference

**L189:** We added an indication of the super El Niños in the figure.

**L200-201:** We added reference to the Huber and Knutti paper.

**L490-492:** Garbled statement has been fixed.

**L654:** That line was eliminated in cutting of tangential material.

**L675:** Negative growth now defined parenthetically.

**L769:** "annually" has been inserted before "added"

**L852:** We added the terminology "carbon dioxide removal (CDR)"

**L932:** These sentences were eliminated in removing tangential discussion.

**L1048:** spelling corrected

**L1065:** Agreed, it was not clear – we have simplified the sentence

**L1096:** Word "to" added for clarity.

**L1100:** This paragraph was removed in distilling the paper.

**L1142:** We added the specific terminology "solar radiation management (SRM)"

**L1201:** Good point – we don't mean to downplay the implications of 0.5-1 meter sea level rise, so we changed "consequences" to "dire consequences"

**L1252-1253:** We have expanded the sentence to expand on the potential for developed countries to contribute to a slowdown of climate forcing growth.

**L1277:** We have expanded that sentence to clarify extraction and reduction requirements.

---

## Author Response (AR3)

We thank the reviewers and the editor for their considerable assistance. We appreciate the efforts required to deal with a long paper, especially one that explicitly addresses implications of the science for the public and governmental policies, we recognize the reviewer's expertise and insights as revealed by their reviews, and we wish to acknowledge the significant improvements in the paper that resulted from these expert reviews.

The first author (JEH) also acknowledges the unusual contributions of his co-authors, who include world-leading experts in several disparate disciplines essential for assessing the broad issues of concern. As discussed in the paper's Introduction, the paper aims to be clear and understandable to a broad audience including the judiciary, which in many nations is better able than other branches of government to help assure that policies are responsible in the long-term. Although assistance of these experts was requested to assure the paper's accuracy in their specific areas of research, in fact they all contributed to the overall content, organization and conclusions of the paper, and I thank them for the extra efforts required by the several iterations of the paper. This paper complements an earlier paper with similar overall objectives [Assessing 'Dangerous Climate Change', PLOS ONE, 8, e81648, 2013], which also included world experts in additional relevant fields (economics, human physical and mental health, glaciology, species extinctions, and coral reefs).

Below we describe the changes made in response to reviewer suggestions, after first noting a few updates in addition to those requested. We are submitting a version of this revised paper that "tracks" all changes from the prior version, as well as a clean version that incorporates all of the changes and locates the figures conveniently.

Updates/Corrections:

1. Global temperature data is updated through April 2017 in several figures and greenhouse gas amounts are updated to the most recently available data.

2. Several references have been updated to their final published version. Recent references on $CH_4$ were added in Appendix A8 and reference to the recent Etminan et al. paper on $CH_4$ climate forcing was added in Appendix A10.

3. On page 2 the quotation from the Paris Agreement, which was from a draft of that Agreement, has been replaced by the quote as it appeared in the final Paris Agreement.

4. In Section 8 the description of concern about the rate of emission reduction in the Alec L v. Jackson (2012) case is modified, based on a transcript of the court proceedings.

5. Andrew Lacis, who developed the radiation formulae we are using for climate forcings and who made useful inputs to the current paper, is added as co-author.

Response to reviewer suggestions:

**Reviewer #2**
The readability of the paper was my main concern on the last round, and this is now much improved. Going through the new draft, I've jotted down a few additional points. I expect these can be easily addressed.

236, 242-245. A perceptive reader may wonder why sea level didn't fall by 3-5m from the early to late Holocene, when temperatures fell by ~0.5C. It would be good to explain why the stability of sea level through the Holocene doesn't undermine the assumed sea level/warming connection.
**Response:** It takes a little space to address this topic, but we have added the following:

Near stability of sea level in the latter half of the Holocene as global temperature fell about 0.5°C, prior to rapid warming of the Modern Era (Fig. 3), is not inconsistent with that global cooling. Hemispheric solar insolation anomalies in the latter half of the Holocene favored ice sheet growth in the Northern Hemisphere and ice sheet decay in Antarctica (Fig. 27a, Hansen et al., 2016), but the Northern Hemisphere did not become cool enough to reestablish ice sheets on North America or Eurasia. There was a small increase of Greenland ice sheet mass (Larsen et al., 2015), but this was presumably at least balanced by Antarctic ice sheet mass loss (Lambeck et al., 2014).

311: This is reworded from the last draft, but the rewording didn't address my concern. The juxtaposition of this sentence and the sentence starting later on line 329 may help bring the problem into clearer focus: "Climate forcings in Figure 4 are the planetary energy imbalance caused by the preindustrial-to-present change of each atmospheric constituent… Earth's energy imbalance is the portion of the forcing that has not yet been responded to." Two possible fixes: (1) Don't try to define forcing at all, and point to a reference that defines it, or (2) Define forcing rigorously as a hypothetical imbalance that would be obtained if climate is not allowed to change. In this case the distinction between actual and hypothetical imbalance needs to be clearly spelled out.
**Response:** o.k., we understand the reviewer's concern. The problem is fixed with the addition of half a sentence and a reference, and in so doing we have actually included both of the reviewer's suggested alternatives. The added half-sentence makes clear that the energy imbalance refers to the hypothetical situation with the climate held fixed at its preindustrial state. The reference includes a full discussion of climate forcing and its relation to planetary energy imbalance. The sentence at former line 311 now reads:

Climate forcings in Fig. 4 are the planetary energy imbalance that would be caused by the preindustrial-to-present change of each atmospheric constituent, if the climate were held fixed at its preindustrial state (Hansen et al., 2005).

Figure 8 caption: Need to insert the symbol DeltaFe parenthetically after "Effective climate forcing".
**Response:** We have made the suggested change.

Figure 9 caption: The main black curve on left is not adequately defined in the legend. Actual emissions until some year ?, then constant until year ?. Okay, reading on, I see that these details are clarified in the text. But the figure should stand better on its own.
**Response:** We have changed the legend in the figure to define the black curve as actual data.

531. Without any explanation, the focus shifts from combined forcings to CO2 forcing alone.

The reader needs a "sign post" for orientation. Are forcings from all agents other than CO2 going to be neglected in scenarios? Is this reasonable? I suggest that the content from lines 617-619 needs to be moved here, along with a bit more elaboration, to make this transition understandable.
**Response:** We agree and have modified the paragraph accordingly.

537. The focus now shifts back to combined forcings (Fig. 8). To avoid the double shift, I suggest moving this paragraph ahead of the previous one.
**Response:** We have switched these two paragraphs, as suggested.

558. Similarly, it's potentially confusing that the term "scenario" now shifts meaning from emissions scenarios to atmospheric CO2 simulations. I suggest: "Figure 10a shows the simulated changes in atmospheric CO2 for the baseline emission cases (Figure 9a). These cases do not include active CO2 removal. Five additional cases that include CO2 removal are shown …"
**Response:** We have adopted essentially the suggested rewording.

606-608. If true, it would help to state that F(t) includes volcanic forcing from a base that assumes zero volcanism.
**Response:** Equation (1) does not imply assumption of a zero volcanism base. Instead it assumes that the volcanic aerosol forcing over the period prior to 1850 was a constant equal to the average volcanic aerosol forcing in the period 1850-2015 ($- 0.3$ W/m$^2$). Volcanoes that occurred prior to 1850 had a constant cooling effect up until 1850 ($0.225°C$, if climate sensitivity is $0.75°C$ per W/m$^2$), but as time goes on the cooling effect of the pre-1850 volcanoes gradually decreases to zero as volcanic cooling is gradually taken over by the time dependent volcanoes after 1850).

For clarification, we have inserted the following sentence at former line 608:
The assumed-constant pre-1850 volcanic aerosols caused a constant cooling up to 1850, which gradually decreases to zero after 1850 and is replaced by post-1850 time-dependent volcanic cooling; note that $T(1850) = 0°C$.

721. Better to avoid expressing extracted amounts in ppm, which may confuse.
**Response:** We deleted the extracted amount in ppm.

957.. why "but"? Context would fit better with a "moreover"
**Response:** We think "but" fits better – probably the sentence was misread by the reviewer.

1080. The year should be 2013.
**Response:** date has been corrected.

1081. TCR(t) is lacking a definition. As defined in AR5, Chapter 10, TCR doesn't depend on time, so the formulation appears to differ from AR5. If so, some further explanation of the method is needed.
**Response:** We have removed the time dependence (t) and defined TCR at doubled CO$_2$ (1%/year CO$_2$ increase) and TCRE at 2100 as in AR5, Chapter 10, so this section now reads:

**A2. Transient Climate Response to cumulative $CO_2$ Emissions (TCRE)**

The transient climate response (TCR), defined as the global warming at year 70 in response to a 1%/year $CO_2$ increase, for our simple Green's function climate model is 1.89°C with energy imbalance of 1.52 $W/m^2$ at that point; this TCR is in the middle of the range reported in the IPCC AR5 report (IPCC, 2013). We calculate the transient climate response to cumulative carbon emissions (TCRE) of our climate plus carbon cycle model as in Section 10.8.4 of IPCC (2013), i.e., TCRE = TCR $\times$ CAF/$C_0$, where $C_0$ = preindustrial atmospheric $CO_2$ mass = 590 PgC and CAF (t) = Catm/Csum, Catm = atmospheric $CO_2$ mass minus $C_0$ and Csum = cumulative $CO_2$ emissions (all evaluated at year 2100).

We find TCRE = 1.54°C per 1000 PgC at 2100 with constant emissions (which yields cumulative emissions of 1180 PgC at 2100, which is near the midpoint of the range assessed by IPCC, i.e., 0.8°C to 2.5°C per 1000 PgC (IPCC, 2013). Our two cases with rapidly declining emissions never achieve 1000 PgC emissions, but TCRE can still be computed using the IPCC formulae, yielding TCRE = 1.31 and 1.25°C per 1000 PgC at 2100 for the cases of −3%/year and −6%/year respective emission reductions. As expected, the rapid emission reductions substantially reduce the temperature rise in 2100.

**Reviewer #3**

Hansen and colleagues have appropriately taken into account my comments and concerns in this revised manuscript. I have no substantial issues to raise.
Below I list a few final comments and clarifications, which could be taken into account by the authors when preparing their final submission. These are so minor that they do not require a further review round from my side.

1) Line 152: Some spaces and a period seemingly without any sense or purpose have crept into this line.
**Response:** We have fixed this typo.

2) Line 801: "out" seems to be missing
**Response:** Yes, we have inserted "out".

3) Line 841: It has equally be shown based on historical data that nuclear deployment in the past has shown "negative learning" with costs increasing over time instead of being reduced. It seems reasonable to state both sides of the story:
http://www.sciencedirect.com/science/article/pii/S0301421510003526
**Response:** Yes, that is a good point. Actually the Lovering et al. (2016) contains the French (rising) cost data (including reference to the Grubler paper) and nuclear construction cost histories for all nations with available data. To include a proper balance we have changed:

" and it has been shown that reactor costs stabilize or decline with repeated construction of the same reactor design (Lovering et al., 2016)." to "Although in some countries reactor costs stabilized or declined with repeated construction of the same reactor design, in others costs have risen for a variety of reasons (Lovering et al., 2016)."

[Note that the main thing the Lovering et al. paper shows is that negative learning (which has occurred in a number of places) is not intrinsic to nuclear energy, but a result of governance, macroeconomic, and other factors. South Korea is a good example of positive learning, and China probably is too, but they have not made adequate data available.  Also note that costs in France actually were quite stable, growing less than the general construction index.]

4) Line 1150: A paper, only published two weeks ago, reconciling controversies about the hiatus might be a good additional reference here:
https://www.nature.com/nature/journal/v545/n7652/full/nature22315.html
**Response:** Yes, it is a very good summary reference – we added it at two points.

5) Comment on Figure 13: With energy accounting method I was hoping to clarify whether the partial substitution method or the physical energy content methods was applied (see: https://www.iea.org/statistics/resources/questionnaires/faq/#one) as this can strongly impact the perceived share of renewable energy in the primary energy mix. It would still be useful to clarify this in the caption. BP seems to use the partial substitution method if I interpret footnote * on page 38 of this document on the BP statistical review correctly:
http://www.bp.com/content/dam/bp/pdf/energy-economics/statistical-review-2016/bp-statistical-review-of-world-energy-2016-renewable-energy.pdf

**Response:** Agreed, BP uses the substitution method for energy accounting – this has been added to the Fig. 13 caption with reference to Macknick (2011) for explanation of energy accounting.